# The distinct metabolic phenotype of lung squamous cell carcinoma defines selective vulnerability to glycolytic inhibition

Justin Goodwin[1,*], Michael L. Neugent[1,*], Shin Yup Lee[1,2,*], Joshua H. Choe[1,3], Hyunsung Choi[1], Dana M.R Jenkins[1], Robin J. Ruthenborg[1], Maddox W. Robinson[1], Ji Yun Jeong[4], Masaki Wake[5], Hajime Abe[5], Norihiko Takeda[5], Hiroko Endo[6], Masahiro Inoue[6], Zhenyu Xuan[1,7], Hyuntae Yoo[1], Min Chen[8], Jung-Mo Ahn[9], John D. Minna[10], Kristi L. Helke[11], Pankaj K. Singh[12], David B. Shackelford[13] & Jung-whan Kim[1]

Adenocarcinoma (ADC) and squamous cell carcinoma (SqCC) are the two predominant subtypes of non-small cell lung cancer (NSCLC) and are distinct in their histological, molecular and clinical presentation. However, metabolic signatures specific to individual NSCLC subtypes remain unknown. Here, we perform an integrative analysis of human NSCLC tumour samples, patient-derived xenografts, murine model of NSCLC, NSCLC cell lines and The Cancer Genome Atlas (TCGA) and reveal a markedly elevated expression of the GLUT1 glucose transporter in lung SqCC, which augments glucose uptake and glycolytic flux. We show that a critical reliance on glycolysis renders lung SqCC vulnerable to glycolytic inhibition, while lung ADC exhibits significant glucose independence. Clinically, elevated GLUT1-mediated glycolysis in lung SqCC strongly correlates with high [18]F-FDG uptake and poor prognosis. This previously undescribed metabolic heterogeneity of NSCLC subtypes implicates significant potential for the development of diagnostic, prognostic and targeted therapeutic strategies for lung SqCC, a cancer for which existing therapeutic options are clinically insufficient.

[1] Department of Biological Sciences, The University of Texas at Dallas, Richardson, Texas 75080, USA. [2] Department of Internal Medicine, School of Medicine, Kyungpook National University, Daegu 41944, Korea. [3] St Mark's School of Texas, Dallas, Texas 75230, USA. [4] Department of Pathology, School of Medicine, Kyungpook National University, Daegu 41944, Korea. [5] Department of Cardiovascular Medicine, The University of Tokyo, Tokyo 113-8655, Japan. [6] Department of Biochemistry, Osaka International Cancer Institute, Osaka 541-8567, Japan. [7] The Center for Systems Biology, The University of Texas at Dallas, Richardson, Texas 75080, USA. [8] Department of Mathematical Sciences, The University of Texas at Dallas, Richardson, Texas 75080, USA. [9] Department of Chemistry and Biochemistry, The University of Texas at Dallas, Richardson, Texas 75080, USA. [10] Department of Medicine and Pharmacology, Hamon Center for Therapeutic Oncology Research, Simmons Comprehensive Cancer Center, The University of Texas Southwestern Medical Center, Dallas, Texas 75390, USA. [11] Department of Comparative Medicine, Pathology and Laboratory Medicine, Medical University of South Carolina, Charleston, South Carolina 29425, USA. [12] The Eppley Institute for Cancer and Allied Diseases, Department of Biochemistry and Molecular Biology, Department of Genetics, Cell Biology and Anatomy, Department of Pathology and Microbiology, University of Nebraska Medical Center, Omaha, Nebraska 68198, USA. [13] Department of Pulmonary and Critical Care Medicine, David Geffen, School of Medicine, University of California, Los Angeles, California 90095, USA. * These authors contributed equally to this work.. Correspondence and requests for materials should be addressed to J.-w.K. (email: jay.kim@utdallas.edu).

Overall, 80–85% of all human lung cancers are non-small cell lung cancer (NSCLC), and the majority of NSCLC comprises two major histological subtypes: adenocarcinoma (ADC) and squamous cell carcinoma (SqCC)[1]. SqCC accounts for 25–30% of all lung cancers. Five-year survival rates among advanced SqCC patients being treated with current chemotherapeutic regimens is less than 5% (ref. 2). Although ADC has benefited the most from molecularly targeted therapies[3], to date, few achievements in the development of a targeted therapy for SqCC have been made, resulting in the use of platinum-based chemotherapy remaining the first-line treatment for decades[4]. The recent FDA approval of Necitumumab in combination with platinum-based chemotherapy as a first-line treatment for metastatic SqCC has generated positive, albeit limited clinical impact[5,6].

Aerobic glycolysis has been implicated in tumour growth and survival, contributing to cellular energy supply, macromolecular biosynthesis and redox homeostasis[7,8]. Despite recent advances in our understanding of the metabolic differences between cancer and normal cells, tumour-type-dependent metabolic heterogeneity is still largely unknown[9]. In particular, the differential usage of metabolic pathways in NSCLC subtypes has not been addressed outside clinical observations[10–15], and detailed functional studies have not been performed in representative preclinical models.

The glucose transporter 1 (GLUT1) is a facilitative membrane glucose transporter[16]. Among 14 GLUT family members, GLUT1 is the most frequently implicated in human cancers and is responsible for augmented glucose uptake and metabolism[17]. Several oncogenic transcription factors, such as c-Myc, have been shown to directly regulate GLUT1 mRNA expression in human cancers[18]. Aberrant activation of growth factor or oncogenic signalling pathways, such as PI3K/AKT, enhances GLUT1 activity via increased membrane trafficking[19,20]. In addition to these cell-autonomous, intrinsic pathways, GLUT1 expression is profoundly regulated by tumour microenvironmental effectors. For example, hypoxia induces GLUT1 expression via the transcription factor, hypoxia-inducible factor-1α (HIF-1α). In addition, the selective acquisition of KRAS or BRAF mutations in response to glucose deprivation has been shown to upregulate GLUT1 expression[21,22]. Elevated GLUT1 expression is clinically relevant to positron emission tomography (PET) scanning with the use of $^{18}$fluro-2-deoxy-glucose ($^{18}$F-FDG) for initial diagnosis as well as prognostic evaluation of NSCLC[23].

In this study, we sought to identify the lung SqCC-specific core metabolic signature by integrating multifactorial experimental approaches. We show that GLUT1 is remarkably and uniquely elevated at both the mRNA and protein levels in SqCC as the principal cellular glucose transporter, but is minimally expressed in ADC. Elevated GLUT1 expression in SqCC is associated with

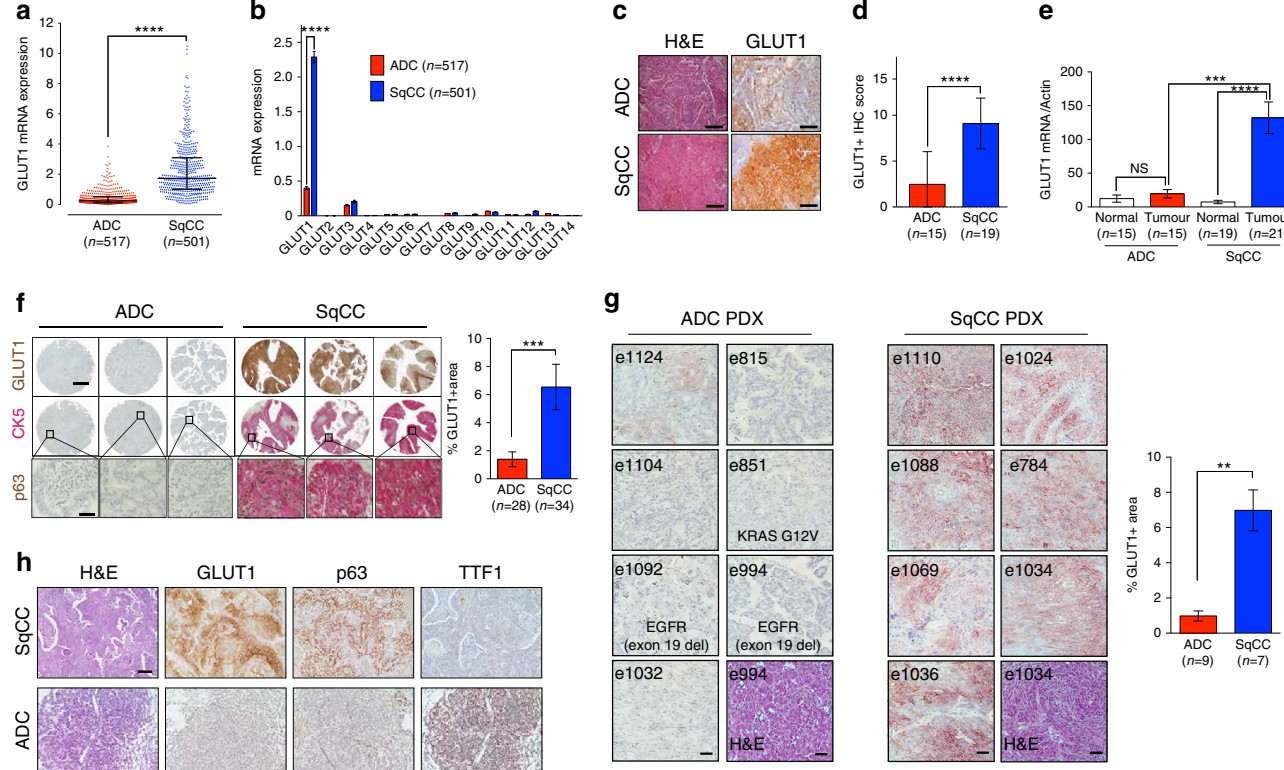

**Figure 1 | GLUT1 is highly expressed in lung SqCC compared to lung ADC.** (**a**) Comparison of GLUT1 expression between 517 ADC and 501 SqCC tumour samples from the TCGA mRNA sequencing data ($10^4$ normalized transcripts per million mappable read (TPM)). Error bars represent the median ± quartile range, Mann–Whitney $u$-test, ****$P < 0.0001$. (**b**) Expression of all GLUT isoform mRNA ($10^4$ normalized transcripts TPM) in TCGA ADC and SqCC tumour samples. Mann–Whitney $u$-test, ****$P < 0.0001$. (**c,d**) Representative IHC images and quantification of GLUT1 expression in human patient clinical samples of ADC and SqCC. ****$P < 0.0001$. (**e**) Comparison of GLUT1 mRNA expression from human NSCLC tumour samples and matched normal lung tissue samples. ****$P < 0.0001$, ***$P < 0.001$. (**f**) Representative IHC images of GLUT1, CK5 and p63 expression in human ADC and SqCC tissue microarray tumour cores (left). Low-magnification scale bar, 300 µm; high-magnification scale bar, 50 µm. Quantification of GLUT1-positive staining in ADC and SqCC microarray samples (right). ***$P < 0.001$. (**g**) Representative IHC images (left) and quantification (right) of GLUT1 expression in SqCC patient-derived xenograft tumours compared to ADC patient-derived xenograft tumours. **$P < 0.01$. (**h**) Representative GLUT1, p63 and TTF1 IHC images of KL ADC and SqCC tumours. Scale bars, 200 µm. All error bars represent the mean ± s.e.m., and two-tailed $t$-test was used unless noted otherwise. All scale bars, 50 µm unless otherwise noted.

enhanced glucose and $^{18}$F-FDG uptake and cellular glucose metabolism, suggesting substantial heterogeneity of glucose dependence and usage between SqCC and ADC. We further demonstrate that SqCC is more susceptible to glucose deprivation than ADC. Notably, pharmacological inhibition of glycolytic flux via non-metabolizable glucose analogue, 2-deoxy-glucose (2-DG) and GLUT1-specific inhibitor, WZB117, selectively suppresses tumour growth in SqCC, whereas ADC is significantly resistant to glycolytic inhibition. These observations suggest that the reliance of SqCC on GLUT1-mediated glucose uptake and metabolism can be exploited for the development of targeted therapeutic strategies for SqCC.

## Results

**TCGA analyses reveal that GLUT1 is elevated in lung SqCC**. To uncover the SqCC-specific gene expression profile among NSCLC, we unbiasedly analysed differential gene expression between SqCC and ADC patient tumour samples utilizing The Cancer Genome Atlas (TCGA) database[24]. Analysis of the mRNA-sequencing gene expression profiles from 501 SqCC and 517 ADC patient tumour samples identified a set of differentially expressed genes (DEGs) between the two NSCLC subtypes (Supplementary Fig. 1). Among enriched DEGs, this analysis identified that the glucose transporter GLUT1 (*SLC2A1*) is among the most significant DEG elevated in SqCC ( > 10-fold median increased expression compared to ADC) along with genes associated with squamous differentiation (for example, *KRTs, PKP1, DSP, SFN* and *JUP*) and extracellular matrix (for example, *COL1A1* and *COL3A1*)[25] (Fig. 1a and Supplementary Fig. 2a). We further examined the mRNA expression of all other 13 GLUT family members and verified that, although other GLUT family member mRNAs (for example, GLUT3) are expressed, GLUT1 is, by far, the most significantly expressed glucose transporter in SqCC patients (Fig. 1b). We also observed a robust correlation between GLUT1 mRNA expression and the mRNA expression of p63, cytokeratin 5 (CK5) and cytokeratin 6A, the most commonly employed clinical biomarkers in distinguishing SqCC[26] (Supplementary Fig. 2b). Notably, no correlation of GLUT1 expression with proliferation markers Ki67 and PCNA in the TCGA cohort of SqCC was found (Supplementary Fig. 2c), suggesting that high GLUT1 expression is not reflective of high cellular proliferation but is phenotypically associated with SqCC. Collectively, these TCGA analyses provide evidence that GLUT1 overexpression is specifically linked to the SqCC subtype.

**Elevated GLUT1 expression in human lung SqCC**. To validate specific upregulation of GLUT1 in human SqCC, we analysed GLUT1 expression in human SqCC (tumour, $n = 21$; normal, $n = 19$) and ADC (tumour, $n = 15$; normal, $n = 15$) tumour tissue samples (Supplementary Table 1). The GLUT1 immunohistochemistry (IHC) score (multiplication of the intensity score by positive area score, Supplementary Fig. 3) was significantly higher in SqCC than in ADC (Fig. 1c,d). Consistently, human SqCC tumours have significantly higher GLUT1 mRNA expression than ADC (Fig. 1e). No significant increase was observed in ADC compared to paired normal lung tissues (Fig. 1e), suggesting that GLUT1 expression is specifically elevated in SqCC.

To further determine whether GLUT1 expression is specifically elevated in human SqCC, we measured GLUT1 levels in tissue microarrays of lung cancer patients. We found that GLUT1 expression is predominantly confined to SqCC patients (79%, 27/34), with little to no expression observed in the various non-squamous subtypes (21%, 6/28; Fig. 1f). This was quantitatively associated with a greater than threefold increase of

GLUT1-positive area in SqCC compared to ADC (Fig. 1f). Consistent with TGCA data analyses (Supplementary Fig. 2b), we found a significant correlation between GLUT1 and p63/CK5 (Supplementary Fig. 4a,b), indicating that elevated GLUT1 expression is specific to SqCC across a wide selection of lung cancer patients as opposed to other NSCLC subtypes.

We further validated differential expression of GLUT1 within human NSCLC using a patient-derived xenograft (PDX) model[27]. Immunohistochemical analysis of GLUT1 expression across an array of SqCC ( $n = 7$ ) and ADC ( $n = 9$ ) PDX samples revealed significantly elevated GLUT1 expression in SqCC PDX tumours compared to ADC PDX tumours (Fig. 1g and Supplementary Table 2). SqCC PDX tumours showed high levels of relatively homogeneous GLUT1 expression ( $P < 0.001$ , two-tailed *t*-test). In contrast, GLUT1 expression was strictly confined adjacent to severely necrotic areas in ADC PDX tumours indicative of hypoxic induction of GLUT1. Notably, despite the fact that most ADC PDX tumours were established from highly advanced metastatic patients (Supplementary Table 2) because of the relatively low PDX success rate of early-stage ADC tumours, GLUT1 expression is negligible in ADC PDX tumours. Taken together, these data provide strong evidence of the heterogeneous expression of GLUT1 in NSCLC and its specific overexpression in SqCC as the principal glucose transporter.

**Lung SqCC-specific GLUT1 expression in NSCLC animal model**. Next, we sought to determine whether GLUT1 expression could differentiate SqCC from other various pulmonary neoplasia in the *Kras*$^{G12D}$; *Lkb1-null* (KL) mouse model of NSCLC. The KL mice, in which oncogenic mutant Kras, Kras$^{G12D}$, is expressed and tumour suppressor *Lkb1* is deleted upon intratracheal inhalation of adenovirus-Cre, exhibit a full spectrum of NSCLC subtypes, allowing the characterization of different tumour types within the same mouse[25,28,29] (Supplementary Fig. 5a). The majority of tumours that develop in KL mice can be histopathologically identified as ADC ( ~45% ) and SqCC ( ~25% ), the two most frequently observed NSCLC phenotypes in humans (Supplementary Fig. 5b). Immunohistochemical staining on serial sections of KL tumours revealed that GLUT1 is exclusively expressed in SqCC tumours determined by co-localization with diagnostic markers such as p63 (Fig. 1h and Supplementary Fig. 5c). In contrast, no detectable levels of GLUT1 were found in TTF-1-positive ADC tumours (Fig. 1h). Accordingly, we found a strong correlation between GLUT1 levels and squamous marker p63 in individual KL tumours (Supplementary Fig. 5d). To exclude the possibility of a distinct subset of KL ADC that may express p63, we further validated SqCC or ADC subtypes in KL tumours by double-staining with two SqCC markers, p63 and CK5, as well as a combination of SqCC and ADC markers, CK5 and TTF-1. In all KL tumours we examined, p63 and CK5 were co-localized in SqCC tumours, whereas TTF-1-expressing tumours (ADC) were negative for SqCC marker CK5 expression and *vice versa* (Supplementary Fig. 5e). Taken together, these data demonstrate that GLUT1 expression is markedly and specifically elevated in murine SqCC tumours.

**Elevated GLUT1 and glucose uptake in lung SqCC cell lines**. We next asked whether NSCLC cell lines retain differential GLUT1 expression to clinical and animal model tissues. Across a panel of NSCLC cell lines, the majority of SqCC cell lines express 7–15-fold higher levels of GLUT1 mRNA as compared to ADC cell lines (Fig. 2a), which closely correlates with GLUT1 protein levels (Fig. 2b). We focused our further *in vitro* and *in vivo* studies to four cell lines, SqCC cell lines HCC95 and HCC1588, and ADC cell lines A549 and H522, after validating their expression of p63 and CK5 (Supplementary Fig. 6a), indicating that these cell lines

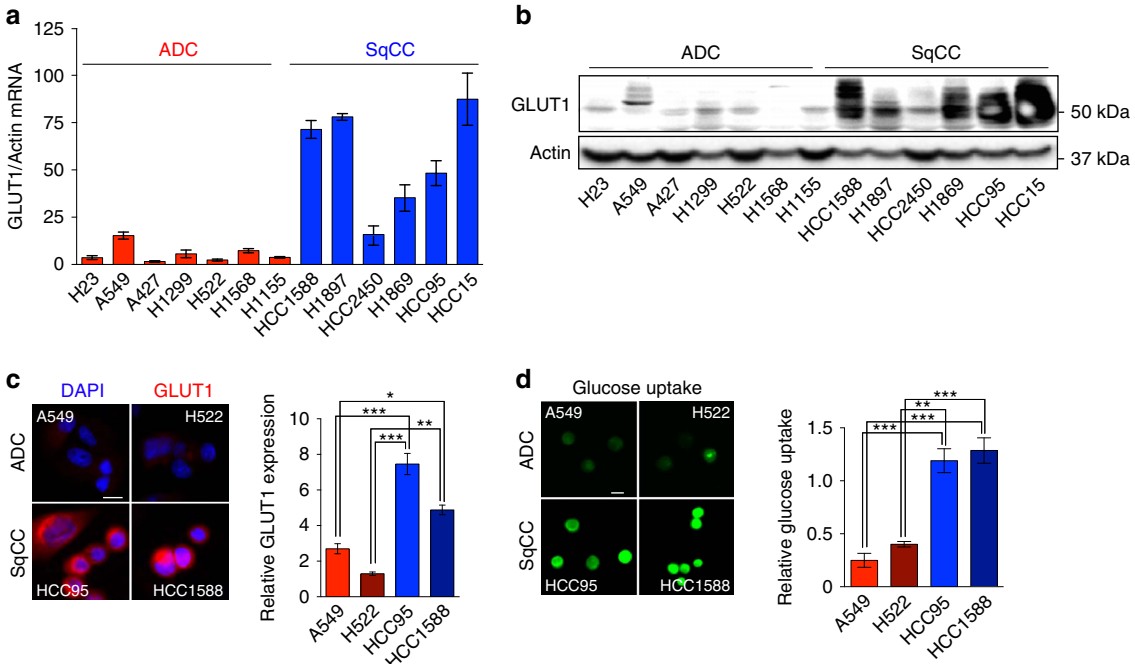

**Figure 2 | GLUT1 expression and glucose uptake in lung SqCC cell lines.** (**a**) qRT–PCR analysis comparing GLUT1 mRNA expression in a panel of ADC and SqCC cell lines ($n = 6$–8 per cell line from three biologically independent experiments). (**b**) Immunoblot analysis of GLUT1 expression in ADC and SqCC cell lines. (**c**) Representative GLUT1 immunocytochemistry images (left) and fluorescent quantification (right) of GLUT1 staining ($n = 3$, 7–10 images from each experiment were captured for quantification). $***P < 0.001$, $**P < 0.01$, $*P < 0.05$. (**d**) Representative fluorescent images (left) and quantification (right) of fluorescent glucose uptake in ADC and SqCC cell lines ($n = 3$, 7–10 images from each experiment were captured for quantification). $***P < 0.001$, $**P < 0.01$. All error bars represent the mean ± s.e.m., and one-way ANOVA was used. All scale bars, 25 m.

conserved their *in situ* specific phenotype. Consistent with TCGA data, GLUT1 is predominantly expressed in SqCC cell lines compared to other GLUT family members (Supplementary Fig. 6b). We further validated membrane localization of GLUT1 in these cells[19,20]. Immunocytochemical staining showed GLUT1 concentrated at the plasma membrane in SqCC cell lines (Fig. 2c). Quantification of GLUT1 fluorescence intensity was three- to fourfold higher in SqCC cell lines as compared to ADC cell lines (Fig. 2c). Fluorescent glucose uptake was measured using flow cytometry (Supplementary Fig. 6c) and fluorescent microscopy (Fig. 2d), revealing a dramatic increase in glucose uptake in SqCC cell lines. Collectively, these data show that not only is GLUT1 transcriptionally upregulated specifically in SqCC, it is localized to the plasma membrane where it significantly contributes to enhanced glucose uptake.

**GLUT1 knockdown impairs SqCC viability and tumour growth.** Given the high expression level and critical role in cellular glucose uptake, we reasoned that GLUT1 may be necessary for cell viability and growth of SqCC. We performed short-hairpin RNA (shRNA)-mediated knockdown of GLUT1 in SqCC HCC95 and HCC1588 cells using two different GLUT1-targeting sequences (Fig. 3a and Supplementary Table 5). GLUT1 knockdown markedly suppressed the proliferative capacity of SqCC cells compared to shGFP control cells even in high-glucose (25 mM) conditions (Fig. 3b). This was accompanied by extensive apoptosis and cell death, which we observed by Annexin-V and 7-aminoactinomycin-D (7-AAD) staining, respectively (Fig. 3c). In contrast, GLUT1 knockdown in ADC cell lines A549 and H522 only moderately suppressed proliferation with no induction of apoptosis or cell death (Supplementary Fig. 7a–c). Fluorescently labelled glucose uptake was significantly decreased in GLUT1-deficient cells, which was correlated with lower

intracellular ATP, NADH and NADPH, suggesting not only disrupted bioenergetics, but an essential role for the GLUT1-dependent flux of glucose intermediates into glycolysis-dependent pathways such as the pentose phosphate pathway (Fig. 3d–f). In contrast to SqCC, reduced glucose uptake did not affect cell viability or intracellular ATP in ADC cell lines (Supplementary Fig. 7d,e). Consistently, transient GLUT1 knockdown using short interfering RNA (siRNA) was more pronounced in SqCC cell lines, with a complete suppression of proliferation in SqCC but not ADC cells despite reduced glucose uptake in both SqCC and ADC cell lines (Supplementary Fig. 8a–c). Intriguingly, A549 was more sensitive to transient GLUT1 knockdown than H522, in accordance with previous reports that loss of LKB1 sensitizes cells to metabolic disruption[30]; however, both ADC cell lines continued to proliferate even after depletion of GLUT1 (Supplementary Fig. 8b). We further employed an additional SqCC cell line, HCC2814, that contains an amplification of PIK3CA and lacks a Kras mutation, which are representative features of human lung SqCC. Consistently, GLUT1 knockdown in HCC2814 markedly suppressed *in vitro* proliferation, which is associated with increased apoptosis and decreased intracellular ATP levels (Supplementary Fig. 8d–h).

To determine whether GLUT1 knockdown inhibits tumour growth *in vivo* in SqCC, we implanted HCC95-expressing stable shGLUT1 or shGFP into nude mice. In accordance with *in vitro* proliferation, GLUT1 knockdown significantly inhibited the growth of HCC95 tumours (Fig. 3g and Supplementary Fig. 7g). In contrast, no significant difference in tumour growth was observed between shGLUT1 and shGFP cells in A549 or H522 tumours (Supplementary Fig. 7f,h). The selective cytotoxicity of GLUT1 knockdown in SqCC strongly suggests glycolytic addiction and a specific reliance on glucose metabolism. Notably, a reduction of glucose uptake in ADC cell lines reveals that, while ADC may rely on GLUT1 as its primary glucose transporter,

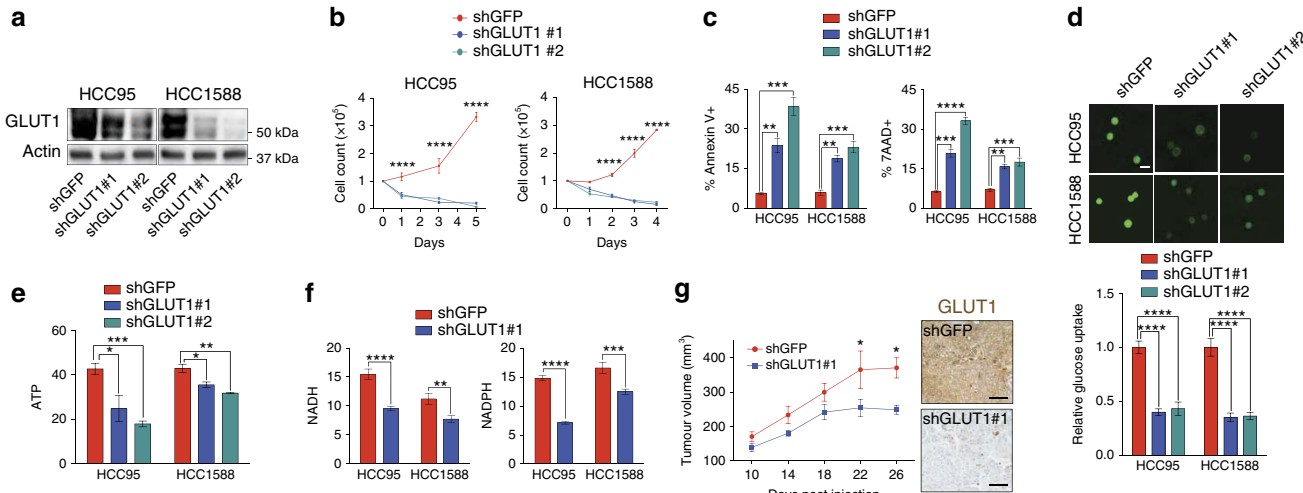

**Figure 3 | GLUT1 knockdown inhibits lung SqCC tumour growth.** (**a**) Immunoblot analysis of GLUT1 expression in control shGFP and shGLUT1 HCC95 and HCC1588 cells. (**b**) *In vitro* proliferation of control shGFP and shGLUT1 HCC95 and HCC1588 cells ($n = 6$ from two biologically independent experiments for each construct). Two-way ANOVA, ****$P < 0.0001$. (**c**) Cell viability was assayed via Annexin-V and 7-AAD staining in control shGFP and shGLUT1 HCC95 and HCC1588 cells ($n = 3$ from three biologically independent experiments for each construct). ANOVA, ****$P < 0.0001$, ***$P < 0.001$, **$P < 0.01$. (**d**) Representative fluorescent images (top), and quantification (bottom) of fluorescent glucose uptake in shGFP and shGLUT1 HCC95 and HCC1588 cells ($n = 3$, six to nine images were captured in each group for quantification). One-way ANOVA, ****$P < 0.0001$. Scale bar, 25 μm. (**e,f**) Comparison of relative intracellular ATP (**e**), NADH and NADPH (**f**) levels between shGFP and shGLUT1 HCC95 and HCC1588 cells ($n = 6$ each group from two to three biologically independent experiments). Two-tailed *t*-test, ****$P < 0.0001$, ***$P < 0.001$, **$P < 0.01$, *$P < 0.05$. (**g**) *In vivo* tumour growth (left) and representative GLUT1 IHC images (right) of shGFP and shGLUT1 HCC95 xenograft tumours ($n = 3$–5 for each group). Scale bar, 100 μm. Two-way ANOVA, *$P < 0.05$. All error bars represent the mean ± s.e.m.

there exist mechanisms within ADC to maintain bioenergetics, cellular viability and proliferative capacity outside of glucose metabolism. Overall, these data show that GLUT1 expression and function are vital for maintaining energy homeostasis and proliferation in SqCC.

**Lung SqCC is reliant on glycolysis.** GLUT1-mediated glucose transport is considered a key regulatory step of glycolysis in several cancers and benign tissues[17]. Thus, we reasoned that enhanced GLUT1 expression and glucose uptake in SqCC can be essentially linked to tumour aerobic glycolysis. To test this, we first performed gene set enrichment analysis (GSEA) on DEGs enriched in TCGA NSCLC tumour subtypes[31]. Notably, glycolysis was identified as one of the most significantly enriched KEGG gene sets in SqCC (Fig. 4a,b and Supplementary Fig. 9a). Moreover, the key genes belonging to metabolic pathways branching off from glycolysis, the pentose phosphate pathway or those providing intermediate metabolites into glycolysis (for example, fructose, mannose, sucrose and pyruvate metabolic pathways) were also highly elevated in the SqCC-enriched DEGs (Fig. 4a and Supplementary Fig. 9a,b). We further analysed the expression levels of individual glycolytic enzyme genes in the TCGA data sets and found that the majority of them are significantly upregulated in SqCC compared to ADC (Fig. 4c). These data suggest that, along with high GLUT1 expression, SqCC exhibits a coordinately elevated glycolytic and branching metabolic flux.

To verify elevated glycolytic metabolism in SqCC, we measured extracellular lactate concentration and oxygen consumption rate (OCR) of representative ADC and SqCC cell lines. We observed significantly increased lactate production in SqCC cell lines cultured in 25 mM glucose compared to the ADC cell lines (Fig. 4d). Interestingly, SqCC cell lines exhibited significantly reduced lactate production at near physiological glucose concentration (5 mM), whereas ADC cell lines showed no change

(Fig. 4d). OCR is also elevated in SqCC cell lines when cultured in 25 mM glucose. Likewise, near physiological glucose (5 mM) resulted in a significant reduction of OCR only in SqCC cell lines (Fig. 4e). We speculated that this reduction in lactate production and OCR was associated with a disruption of cellular bioenergetics. Indeed, SqCC cell lines grown in media containing 5 mM glucose exhibit significantly reduced intracellular ATP, NADH and NADPH levels compared to cells grown in 25 mM glucose (Fig. 4f–h). However, ADC cell lines exhibited no change in ATP, NADH or NADPH levels between cells grown in 5 mM glucose or 25 mM glucose. It is possible that the change in SqCC bioenergetics represents a higher capacity for glucose uptake and utilization, whereas in 25 mM glucose, glucose uptake in ADC becomes saturated. To validate whether this fluctuation in bioenergetics represents a substantial reliance on glucose, but not an artefact of non-physiologically elevated glucose uptake due to high GLUT1 level in SqCC cell lines, we cultured SqCC and ADC cell lines in decreasing concentrations of glucose. At 5 mM glucose concentration, reduced intracellular ATP, NADH and NADPH (Fig. 4f–h) is associated with a significant decrease in cellular viability compared to ADC cell lines (Fig. 4i), indicating that SqCC cell lines are significantly more sensitive to glucose concentrations than ADC cell lines. Lower than physiological glucose concentrations further decreased the cellular viability of SqCC cell lines. Furthermore, metabolic flux analyses in ADC and SqCC cell lines cultured in 5 mM glucose after glucose starvation revealed that cellular glycolytic flux (extracellular acidification rate, ECAR) is significantly more elevated in SqCC cell lines compared to ADC cell lines (Fig. 4j). OCR was markedly decreased in SqCC compared with a divergent increase in ADC (Fig. 4j). In accordance with SqCC-specific susceptibility to GLUT1 knockdown, these distinct metabolic responses strongly suggest a unique reliance on glucose by SqCC but also provide a rationale for decreased cell viability upon glucose deprivation.

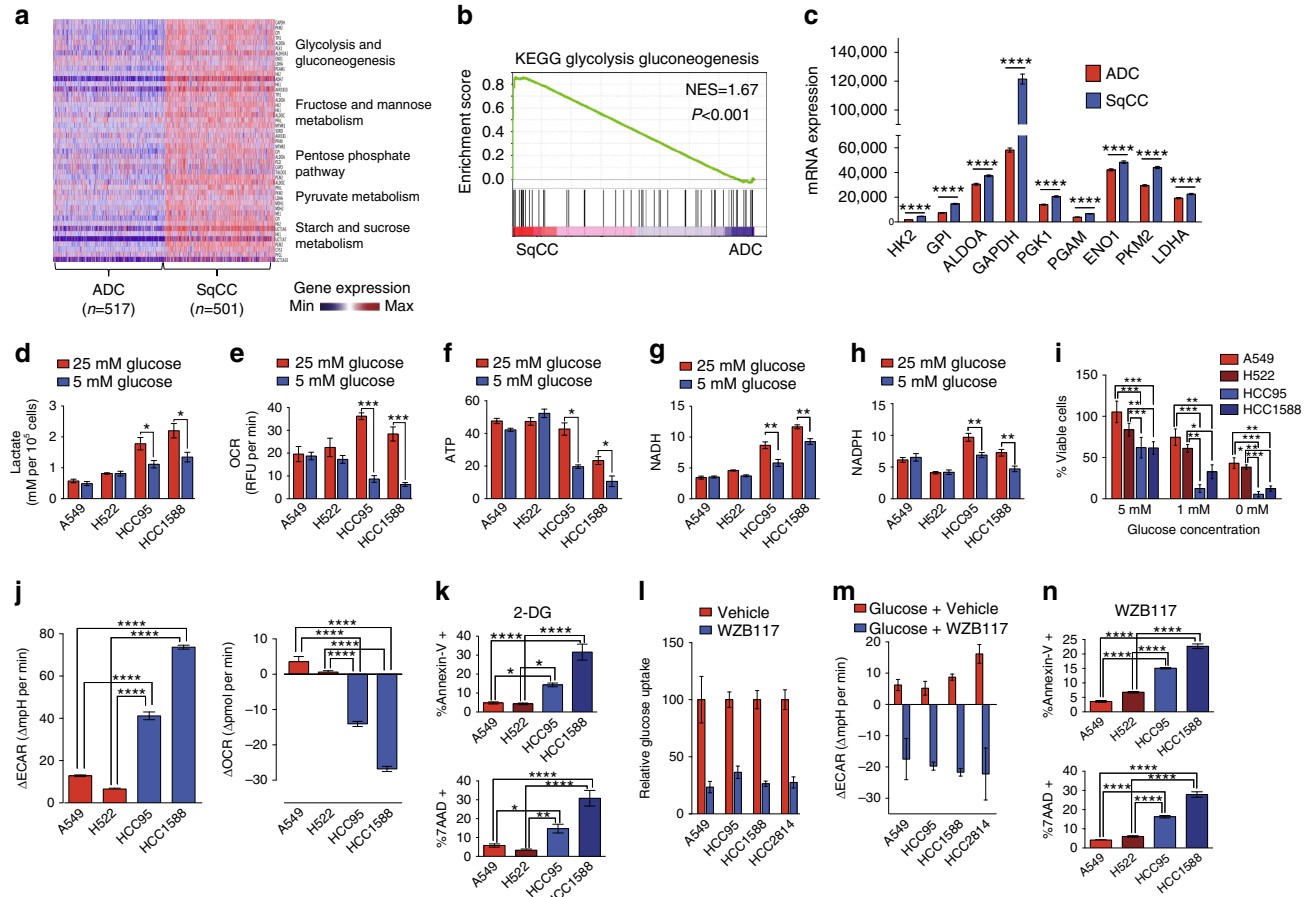

**Figure 4 | Lung SqCC relies on glucose for cellular bioenergetics.** (**a**) Heatmap depicting expression distribution of genes from the highlighted metabolic gene sets enriched in the TCGA cohort of SqCC patients. (**b**) GSEA mountain plot depicting significantly enriched glycolytic gene expression as defined by the KEGG gene set. $P < 0.001$; NES, normalized enrichment score. (**c**) Glycolytic gene expression (normalized TPM) of TCGA SqCC ($n = 501$) and ADC ($n = 517$) patients. Mann–Whitney $u$-test, ****$P < 0.0001$. (**d,e**) Extracellular lactate concentration (**d**) and $O_2$ consumption rate (**e**) in ADC (A549 and H522) and SqCC (HCC95 and HCC1588) cell lines cultured in 25 and 5 mM glucose concentrations for 24 h ($n = 6$–9 each group from two to three biologically independent experiments). Two-tailed $t$-test, ***$P < 0.001$, *$P < 0.05$. (**f–h**) Comparison of relative intracellular ATP (**f**), NADH (**g**) and NADPH (**h**) between cells cultured in 25 and 5 mM glucose concentrations for 24–72 h ($n = 9$ each group from three biologically independent experiments). Two-tailed $t$-test, **$P < 0.01$, *$P < 0.05$. (**i**) Cell viability of ADC and SqCC cell lines cultured in decreasing glucose concentrations for 72–96 h ($n = 3$–4 each group from at least two biologically independent experiments). One-way ANOVA ***$P < 0.001$, **$P < 0.01$, *$P < 0.05$. (**j**) Extracellular acidification rate (left) and $O_2$ consumption rate (right) of ADC (A549 and H522) and SqCC (HCC95 and HCC1588) cell lines in 5 mM glucose after 1-h glucose starvation. ANOVA, ****$P < 0.0001$. (**k**) Cell viability was assayed via Annexin-V and 7-AAD staining in cells treated with glycolytic inhibitor, 2-DG (25 mM) for 72 h ($n = 6$ each group from three to four biologically independent experiments). ANOVA, ****$P < 0.0001$, **$P < 0.01$, *$P < 0.05$. (**l**) Quantification of fluorescent glucose uptake in ADC (A549) and SqCC (HCC95, HCC1588 and HCC2814) cell lines after 1-h treatment with GLUT1 inhibitor WZB117 (50 μM; $n = 2$–3, 10–15 images from each experiment were captured for quantification from at least two biologically independent experiments). (**m**) Extracellular acidification rate of ADC (A549 and H522) and SqCC (HCC95 and HCC1588) cell lines treated with 100 μM WZB117 in 5 mM glucose media ($n = 2$ from two biologically independent experiments). (**n**) Cell viability was assayed via Annexin-V and 7-AAD staining in cells treated with GLUT1 inhibitor WZB117 (50 μM) for 48 h ($n = 6$ each group from three to four biologically independent experiments). ANOVA, ****$P < 0.0001$. All error bars represent the mean ± s.e.m.

If SqCC essentially relies on glycolysis to sustain its survival and proliferation, inhibition of cellular glycolytic flux should be able to selectively exert cytotoxic effects on SqCC (Supplementary Fig. 9c). To test this, we treated SqCC and ADC cells with 25 mM 2-DG, a non-metabolizable glucose analogue that essentially inhibits the cellular glycolytic flux. Cell viability analysis revealed that SqCC cell lines exhibit significantly higher sensitivity to 2-DG treatment compared to ADC cell lines (Fig. 4k; Supplementary Fig. 10a). Notably, A549, although LKB1-null and therefore supposedly more sensitive to metabolic disruption as described previously[30,32], was significantly more viable than SqCC cell lines. Next, we sought to determine whether targeted GLUT1 inhibition selectively affects cellular viability of SqCC cell

lines. We chose WZB117, a selective, small-molecule GLUT1 inhibitor[33], and measured glucose uptake and ECAR in response to WZB117 treatment. Consistent with the previous study demonstrating inhibition of glucose uptake in A549, WZB117 inhibited glucose uptake in A549 by 77% (Fig. 4l). In SqCC cell lines HCC95, HCC1588 and HCC2814, we saw a reduction in glucose uptake of 64%, 74% and 73%, respectively (Fig. 4l). Decreased glucose uptake was associated with a substantial reduction in ECAR in both ADC and SqCC cell lines in a dose-dependent manner after WZB117 treatment (Fig. 4m). Furthermore, cell viability measurements after treatment with WZB117 revealed that SqCC cell lines showed significantly higher susceptibility to GLUT1 inhibition than ADC cell lines (Fig. 4n;

Supplementary Fig. 10b). Considering that WZB117 inhibited glucose uptake and glycolysis in both ADC and SqCC, these results suggest a crucial reliance on GLUT1 for survival and proliferation of SqCC as well as a glycolytic independence in ADC. Although GLUT1 is the predominant glucose transporter expressed in both lung ADC and SqCC cells, a recent study as well as our computational docking analysis (AutoDock Vina, Scripps Research Institute) suggest that WZB117 may exert its effects by binding both GLUT1 and GLUT3 transporters[34]. Despite this additional affinity, GLUT3 is modestly and comparably expressed in the ADC and SqCC cell lines used in our study (Supplementary Fig. 6b), and evaluating the functional role of GLUT3 in ADC using GLUT3 knockdown cells demonstrated no difference in the proliferation of shGLUT3 and shGFP control ADC cells (Supplementary Fig. 11a,b). Collectively, these results suggest a crucial reliance on GLUT1 for survival and proliferation of SqCC as well as a glycolytic independence in ADC.

**Targeting glycolysis or GLUT1 inhibits SqCC tumour growth.** We next investigated whether targeting glycolysis or GLUT1 selectively suppresses *in vivo* tumour growth of SqCC. We treated mice bearing SqCC, HCC1588 or ADC, A549 xenograft tumours with 2-DG (intraperitoneally (i.p.) 500 mg kg$^{-1}$, once daily). Xenograft tumours conserved the distinct histological features and p63 expression patterns of SqCC and ADC upon histopathological evaluation, as well as markedly elevated GLUT1 expression in SqCC tumours (Supplementary Fig. 12). Consistent with *in vitro* results (Fig. 4k), tumour growth of SqCC HCC1588 was significantly inhibited ($\sim$60%) upon 2-DG treatment compared to the vehicle-treated group, with no antitumour activity found in 2-DG-treated ADC A549 tumours (Fig. 5a–c and Supplementary Fig. 13a). Tumour growth inhibition in 2-DG-treated SqCC HCC1588 tumours was associated with increased apoptotic cell death and necrosis, while ADC A549 tumours exhibited no apoptosis or necrotic areas upon 2-DG treatment (Fig. 5d and Supplementary Fig. 13b), indicating that SqCC is selectively susceptible to glycolytic inhibition in accordance with enhanced reliance on glycolytic metabolism.

Next, we reasoned that pharmacological inhibition of GLUT1 should exert similar specific inhibitory effects in SqCC. WZB117 has previously been shown to significantly inhibit ADC A549 tumour growth over a period of 10 weeks. However, ADC A549 showed no difference in tumour growth when treated with WZB117 over the course of 3–4 weeks (Fig. 5e and Supplementary Fig. 13c), in agreement with the previous study[33]. Equivalently, ADC H1299 tumour growth was unaffected by WZB117 treatment (Fig. 5e and Supplementary Fig. 13d). In sharp contrast, SqCC HCC1588 and HCC2814 tumours exhibited up to 40 and 41% reduction in tumour growth upon GLUT1 inhibition, respectively (Fig. 5f,g and Supplementary Fig. 13e,f). Consistent with 2-DG treatment, WZB117-treated SqCC HCC1588 and HCC2814 tumours show increased levels of necrosis and apoptosis; however, cell viability of ADC A549 and H1299 tumours was not affected by WZB117 treatment (Fig. 5h). These results indicate that SqCC relies on GLUT1-mediated glycolysis for *in vivo* tumour growth and is thereby selectively susceptible to glycolytic inhibition.

**Enhanced $^{18}$F-FDG uptake in lung SqCC.** Previous clinical studies have identified increased $^{18}$F-FDG uptake in squamous subtypes[10–15]. We took advantage of the KL NSCLC mouse model, which displays both ADC and SqCC lung tumour heterogeneity (Supplementary Fig. 5a,b), by performing $^{18}$F-FDG-PET/CT imaging of KL mice followed by pathological

evaluation and immunohistochemical staining for GLUT1 (Fig. 6a). $^{18}$F-FDG uptake (SUVmax) was shown to be strongly correlated with SqCC subtypes (Fig. 6a,b), with no correlation found between total tumour burden and SUVmax (Fig. 6c). A comparison of individual SqCC (p63$^+$, SP-C$^-$) and ADC (p63$^-$, SP-C$^+$) tumours within the same mouse confirmed markedly elevated $^{18}$F-FDG uptake in SqCC tumours (Fig. 6d). In addition, analysis of SUVmax of human NSCLC patients (Supplementary Table 1) confirmed that higher SUVmax is evident in SqCC patients compared to ADC patients (Fig. 6e,f). Collectively, these results demonstrate that $^{18}$F-FDG uptake is significantly increased in SqCC tumours, reflecting high GLUT1 expression and glucose uptake compared to ADC tumours.

**PI3K/AKT signalling induces GLUT1 expression through HIF-1α.** We next explored the mechanism underlying the specific upregulation of GLUT1 in SqCC. It has previously been shown that human SqCC tumours and cell lines exhibit high prevalence of genomic copy number gains in chromosome 3q, the genomic region containing the gene of the catalytic subunit of the PI3K complex, PIK3CA[35,36]. Given that PI3K/AKT signalling has been shown to enhance the translation of HIF-1α under non-hypoxic conditions[37,38], we hypothesized that aberrantly activated PI3K/AKT signalling may contribute to GLUT1 expression through oncogenically stabilized HIF-1α signalling. A previous TCGA analysis has identified significant enrichment of PI3K pathway-activating alterations, which include oncogenic or genomic alterations of PIK3CA, PTEN and mTOR in 47% of the TCGA cohort of lung SqCC[35]. We confirmed the frequent amplification of the genomic region containing PIK3CA within lung SqCC (Supplementary Fig. 14a; Broad Institute TCGA Genome Data Analysis Center 2016, SNP6 Copy number analysis, GISTIC2, Broad Institute of MIT and Harvard)[39]. Our TCGA analyses revealed both shallow genomic copy number gains and high-level amplifications of PIK3CA in the majority of patients of the TCGA lung SqCC cohort (46.4% amplification, 44% shallow gain) and a robust linear correlation between PIK3CA genomic copy number and mRNA expression (Fig. 7a and Supplementary Fig. 14c). Notably, GLUT1 mRNA expression positively correlated with PIK3CA genomic copy number (Fig. 7a). Our analysis also revealed that lung SqCC exhibits the highest mRNA expression of PIK3CA among the TCGA cohorts (Supplementary Fig. 15a). Our analysis of the TCGA lung SqCC cohort further revealed frequent shallow deletions in the genomic region containing PTEN (9.8% deep deletion, 44.8% shallow deletion; Supplementary Fig 14b). Similar to PIK3CA, PTEN mRNA expression robustly correlated with genomic copy number (Supplementary Fig 14d). TCGA lung SqCC patients with PTEN genomic loss exhibit the highest expression of GLUT1 mRNA (Fig. 7b). However, no correlation was found between the GLUT1 mRNA level and genomic loss of CDKN2b (p15), the most frequent genomic copy number loss identified within the TCGA lung SqCC cohort (Supplementary Fig. 14b,g), supporting a specific correlation of GLUT1 expression with PIK3CA/PTEN genomic abnormalities. GSEA performed on the ranked set of DEGs between SqCC and ADC TCGA tumour samples identified a significant enrichment of AKT and mTOR oncogenic signalling in SqCC, as defined by the oncogenic signature gene set (Supplementary Fig. 14e,f). Analysis of TCGA reverse-phase protein array (RPPA)[40] data also identified a significant positive correlation between GLUT1 mRNA expression and PIK3CA, phospho-AKT (pT308), phospho-AKT (pS473) and phospho-4E-BP1 (pT37/T46) as well as a negative association between GLUT1 mRNA expression and PTEN protein expression (Supplementary Fig. 15b,c). These results suggest significant input from the PI3K/AKT/mTOR pathway regulating GLUT1 expression in SqCC.

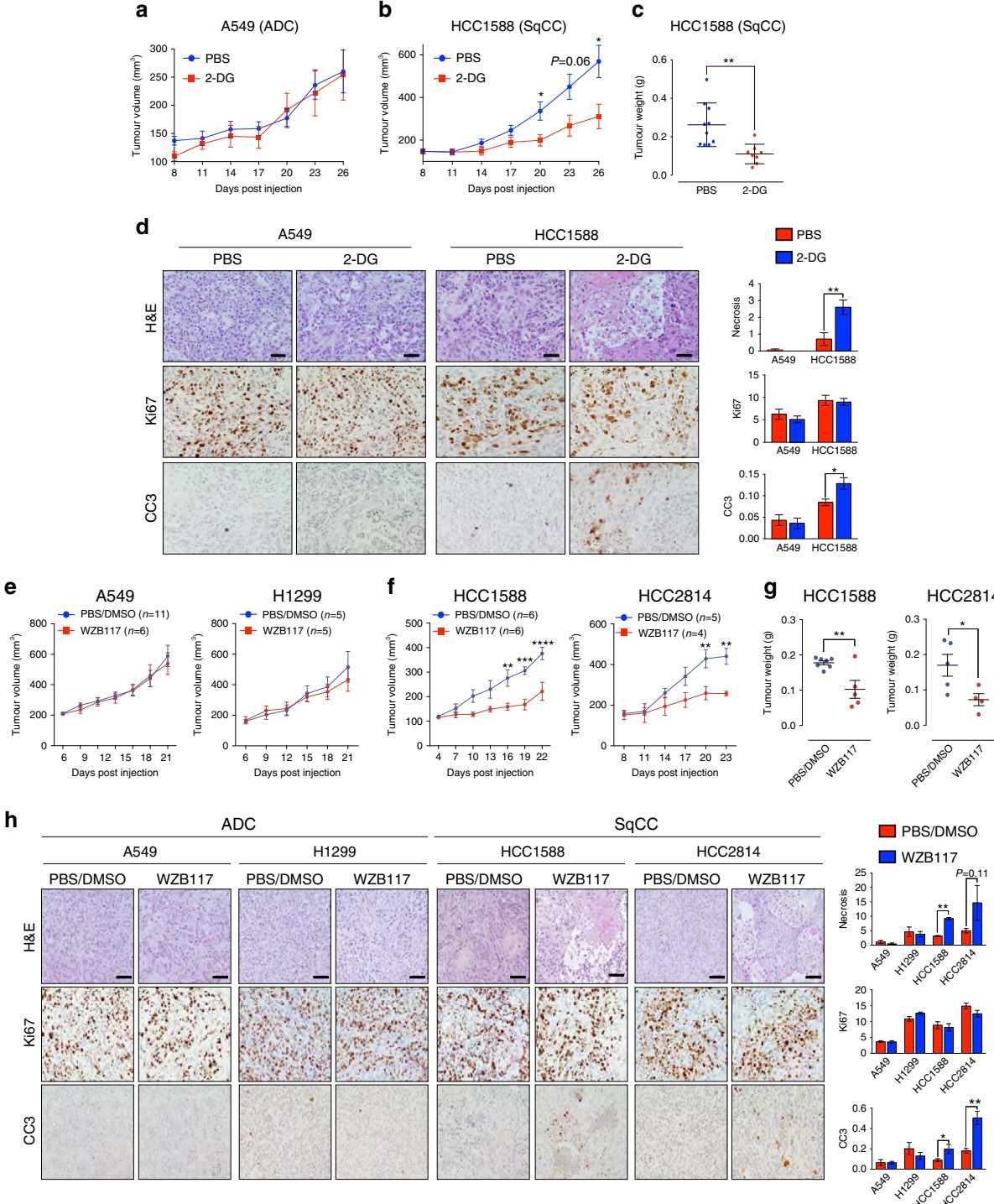

**Figure 5 | Lung SqCC is susceptible to glycolytic inhibition.** (**a,b**) Xenograft tumour growth of A549 ADC (PBS, $n = 9$; 2-DG, $n = 5$; **a**) and HCC1588 SqCC (PBS, $n = 12$; 2-DG, $n = 12$; **b**) treated with PBS as control or glycolytic inhibitor 2-DG (500 mg kg$^{-1}$, once daily). Two-way ANOVA, *$P < 0.05$. (**c**) Tumour weights of PBS- or 2-DG-treated xenograft tumours. Two-tailed $t$-test, **$P < 0.01$. (**d**) IHC analysis (left) and quantification of % area (right) of necrosis, proliferative marker Ki67 and cleaved caspase-3 (CC3) in PBS (A549, $n = 9$; HCC1588, $n = 12$) or 2-DG (A549, $n = 5$; HCC1588, $n = 12$)-treated xenograft tumours. Eight to ten images in each tumour were captured and analysed for quantification. Two-tailed $t$-test, **$P < 0.01$, *$P < 0.05$. (**e,f**) Xenograft tumour growth of ADC A549 (PBS, $n = 11$; WZB117, $n = 6$) and H1299 (PBS, $n = 5$; WZB118, $n = 5$; **e**) and SqCC HCC1588 (PBS, $n = 6$; WZB117, $n = 6$) and HCC2814 (PBS, $n = 5$; WZB117, $n = 4$; **f**) treated with PBS/DMSO as vehicle or GLUT1 inhibitor, WZB117 (10 mg kg$^{-1}$, once daily). Two-way ANOVA, ****$P < 0.0001$, ***$P < 0.001$, *$P < 0.05$. (**g**) Tumour weights of PBS/DMSO or WZB117-treated SqCC HCC1588 (left) and HCC2814 (right) xenograft tumours. Two-tailed $t$-test. **$P < 0.01$. (**h**) IHC analysis (left) and quantification of % area (right) of necrosis, proliferative marker Ki67 and CC3 in PBS/DMSO (A549, $n = 11$; H1299, $n = 5$; HCC1588, $n = 6$; HCC2814, $n = 5$) or WZB117 (A549, $n = 6$; H1299, $n = 5$; HCC1588, $n = 5$; HCC2814, $n = 4$)-treated xenograft tumours. Eight to ten images in each tumour were captured and analysed for quantification. Two-tailed $t$-test. **$P < 0.01$, *$P < 0.05$. All error bars represent the mean ± s.e.m. Scale bars, 100 μm.

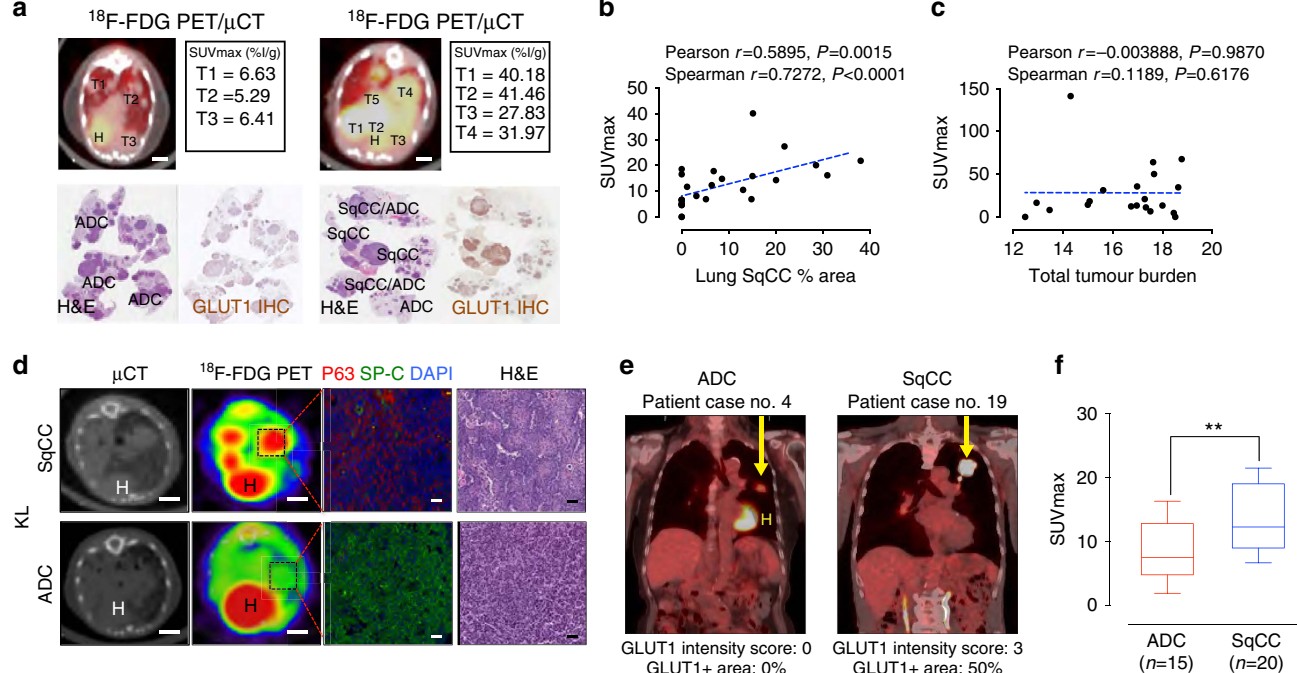

**Figure 6 | Increased $^{18}$F-FDG uptake in lung SqCC.** (**a**) Representative $^{18}$F-FDG-PET/µCT overlays comparing KL animals with only ADC tumours (left) and animals possessing SqCC tumours (right). H, heart. Scale Bar, 4 mm. (**b,c**) Correlative analysis between SUVmax and % area of SqCC tumours (**b**) and total tumour burden (**c**). Pearson and Spearman R-values and probabilities are presented for correlations. (**d**) $^{18}$F-FDG uptake in the areas of SqCC and ADC within the same KL tumour-bearing mouse. Scale bar, 3 mm. Histological phenotype (SqCC or ADC) was determined by IHC analysis of p63 (SqCC) and SP-C (ADC). Scale bar, 50 µm. H, Heart. (**e**) $^{18}$F-FDG uptake of representative human ADC (case 4, Supplementary Table 1) and SqCC (case 19, Supplementary Table 1) patients. H, heart. (**f**) Comparison of SUVmax between ADC and SqCC patients. Each box represents the lower quartile, median and upper quartile, and whiskers represent the 10th and 90th percentiles of the data. Two-tailed t-test. **$P < 0.01$.

To determine whether PI3K/AKT signalling accounts for the HIF-1α-mediated GLUT1 expression in SqCC, we analysed activities of AKT and its downstream signalling pathways as well as HIF-1α expression in KL SqCC and ADC tumours. Consistent with TCGA analysis, SqCC tumours exhibit increased AKT signalling activities compared to ADC tumours, which is strongly associated with HIF-1α expression (Fig. 7c and Supplementary Fig. 16a). Downstream of AKT phosphorylation, 4EBP1 and S6 are known to promote translation of HIF-1α mRNA[37]. Our analysis of SqCC and ADC KL tumours identified that higher p-AKT, p-4EBP1 and p-S6 expression in SqCC is associated with higher HIF-1α and GLUT1 expression in SqCC tumours compared to ADC tumours (Fig. 7c and Supplementary Fig. 16a). Elevated AKT/mTOR activity and HIF-1α expression were also evident in SqCC cells and xenograft tumours (Supplementary Fig. 16b,c).

To validate whether non-hypoxic HIF-1α mediates GLUT1 expression in SqCC, we knocked down HIF-1α in SqCC cells. We found that HIF-1α knockdown resulted in a marked decrease in GLUT1 expression level under non-hypoxic conditions (Fig. 7d). Importantly, SqCC cell lines maintained under non-hypoxic conditions (21% $O_2$) have highly elevated GLUT1 mRNA (Fig. 2a), suggesting hypoxia-independent mechanism(s). Furthermore, GSEA analysis of DEGs between TCGA lung SqCC and ADC cohorts identified a significant upregulation of known HIF-1α target genes[41] (Supplementary Fig. 14h), which include GLUT1 and various glycolytic enzymes within the DEGs enriched in the TCGA lung SqCC cohort (Supplementary Fig. 14i). Collectively, these results suggest that oncogenically activated PI3K/mTOR/HIF-1α signalling contributes to GLUT1 expression in SqCC.

**High GLUT1 correlates with poor prognosis in NSCLC patients.** We next investigated the clinical and prognostic value of GLUT1 expression in the TCGA lung SqCC cohort. We separated the patients into high and low GLUT1 mRNA-expressing groups based on the median GLUT1 mRNA expression. Kaplan–Meier 5-year survival analysis revealed a significantly reduced survival duration in lung SqCC patients with high tumour GLUT1 expression ($P_{\text{log-rank}} = 0.04$) with a hazard ratio of 1.46 (Fig. 8a). Interestingly, we determined high GLUT1 expression to be associated with poor survival in lung ADC patients (Supplementary Fig. 17), suggesting that GLUT1 expression can be employed as a potential prognostic biomarker in NSCLC patients. We further performed clinicopathologic correlation with GLUT1 expression in NSCLC patients (Table 1; Supplementary Table 1). Our analysis confirmed that GLUT1 expression is strongly associated with SqCC (Table 1). High GLUT1 expression is associated with poor tumour cell differentiation (Table 1). Lung SqCC has been primarily linked to smoking[42]. Indeed, we found that GLUT1 expression was strongly associated with smoking ($P = 0.03$, Table 1). High GLUT1 expression was also strongly associated with male patient gender, a reflection of the cultural bias among smokers in the Korean tumour cohort[43]. Consistently, TCGA analysis revealed significantly increased GLUT1 expression in NSCLC patients with smoking history compared to lifelong non-smokers (Fig. 8b). Taken together, these data suggest that lung cancer patients with high GLUT1 expression are significantly associated with SqCC histological phenotype, smoking and poor differentiation.

## Discussion

NSCLC comprises multiple histologically, genetically and presumably metabolically distinct subtypes, given the reciprocal regulation of metabolic pathways and oncogenic signalling[44,45].

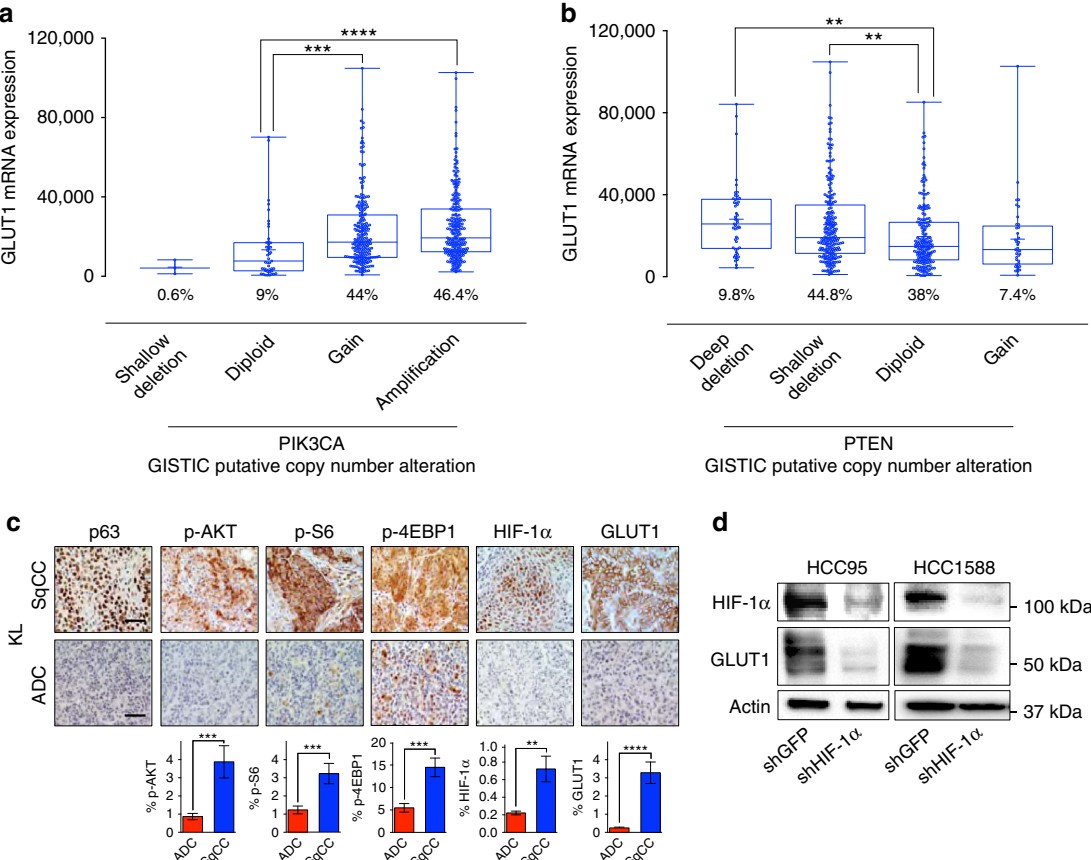

**Figure 7 | Elevated PIK3/AKT/HIF-1α pathways in KL SqCC tumours.** (**a,b**) Analysis of TCGA gene expression (normalized TPM) and PIK3CA (**a**) and PTEN (**b**) genomic copy number alteration profiles. Each dot represents one SqCC patient ($n = 501$). Boxes represent the interquartile range and whiskers are drawn to the minimum and maximum. Kruskal–Wallis non-parametric ANOVA, ****$P < 0.0001$, ***$P < 0.001$, **$P < 0.01$. (**c**) IHC analysis (top) and quantification (bottom) of p63, p-AKT, p-S6, p-4EBP1, HIF-1α and GLUT1 in KL tumours ($n = 6$ each group). Two-tailed $t$-test, ****$P < 0.0001$, ***$P < 0.001$, **$P < 0.01$. Scale bar, 50 μm. (**d**) Immunoblot analysis of HIF-1α and GLUT1 in control shGFP and shHIF-1α knockdown SqCC cell lines, HCC95 and HCC1588. All error bars represent the mean ± s.e.m.

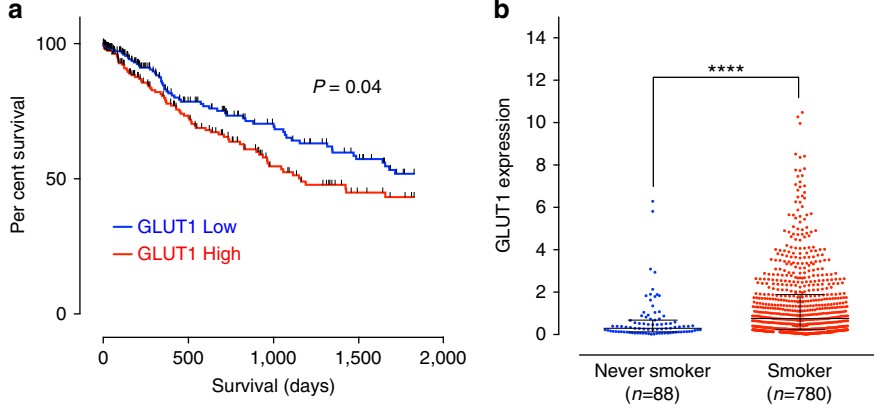

**Figure 8 | High GLUT1 expression is associated with poor prognosis.** (**a**) Kaplan–Meier 5-year survival analysis comparing GLUT1 high and low expressing patients in the TCGA lung SqCC cohort. GLUT1 high and low groups were separated by the median expression. Significance was determined with log-rank test. $P = 0.04$; HR, 1.34. (**b**) Comparison of GLUT1 expression (normalized TPM) in patients with smoking history in TCGA NSCLC cohort. Each dot represents one patient ($n = 1018$). Error bars represent the median ± interquartile range. Mann–Whitney $u$-test. ****$P < 0.0001$.

Increasing evidence suggests that there exist multiple layers of inter- as well as intratumoral metabolic heterogeneity that may essentially affect therapeutic outcomes of metabolic inhibition[9]. In light of this, identification of distinct metabolic alterations that are linked to specific subtypes of NSCLC may lead to the clinical development of effective NSCLC therapeutics with reduced toxicities. In this study, we have discovered distinct metabolic heterogeneity between major tumour subtypes of human NSCLC, ADC and SqCC. Our comprehensive analysis encompassing genetic, metabolic and clinical association approaches demonstrates that GLUT1 expression is markedly and specifically elevated in human SqCC, and is associated with

**Table 1 | Clinical characteristics of patients by GLUT1 expression.**

|  | Total (n = 36) | Low GLUT1* (n = 12) | High GLUT1† (n = 24) | P value |
|---|---|---|---|---|
| Age |  | 65.2 (± 8.7)‡ | 65.3 (± 6.9)‡ | 0.95§ |
| *Sex* |  |  |  |  |
| Male | 27 | 6 (22.2) | 21 (77.8) | 0.04‖ |
| Female | 9 | 6 (66.7) | 3 (33.3) |  |
| *Smoking* |  |  |  |  |
| Ever | 29 | 7 (24.1) | 22 (75.9) | 0.03¶ |
| Never | 7 | 5 (71.4) | 2 (28.6) |  |
| *Histology* |  |  |  |  |
| SqCC | 21 | 1 (4.8) | 20 (95.2) | <0.00001‖ |
| ADC | 15 | 11 (73.3) | 4 (26.7) |  |
| *Stage* |  |  |  |  |
| I | 19 | 9 (47.4) | 10 (52.6) | 0.66# |
| II | 13 | 1 (7.7) | 12 (92.3) |  |
| III | 3 | 1 (33.3) | 2 (66.7) |  |
| IV | 1 | 1 (100.0) | 0 (0.0) |  |
| *Differentiation* |  |  |  |  |
| Well | 4 | 4 (100.0) | 0 (0.0) | 0.003# |
| Moderately | 29 | 8 (27.6) | 21 (72.4) |  |
| Poorly | 3 | 0 (0.0) | 3 (100.0) |  |

ADC, adenocarcinoma; GLU1, glucose transporter 1; IHC, immunohistochemistry; SqCC, squamous cell carcinoma.
*IHC score: 0–4.
†IHC score: 5–12.
‡Mean (± s.d.).
§t-test.
‖χ²-test.
¶Fisher's exact test.
#Cochran–Armitage test for trend.

enhanced glucose uptake and glycolytic dependency. Conversely, GLUT1 and glycolytic enzyme expression remain relatively low in the majority of ADC when compared to SqCC, suggesting that ADC may be significantly less reliant on glucose metabolism. Our study further provides evidence of the selective antitumour effects of glycolytic (2-DG) or GLUT1 (WZB117) inhibition on SqCC. These results have revealed that SqCC is vitally reliant on augmented GLUT1-mediated glucose metabolism for survival and tumour growth. Intriguingly, a recent epidemiological study has demonstrated, in a large cohort, that a diet composed of foods that cause consistently high availability of blood glucose is strongly associated with the development of SqCC among NSCLC patients[46]. This study suggests that high dietary glycaemic index and carbohydrate availability may contribute to the development of SqCC, supporting our finding of the elevated glucose dependence of SqCC.

In sharp contrast, *in vitro* cellular bioenergetic pathways and cell viability as well as *in vivo* tumour growth of ADC was not affected by glucose restriction or glycolytic inhibition. Although previous studies have reported an increase in GLUT1-mediated aerobic glycolysis by EGFR, KRAS, BRAF or other frequent oncogenic mutations in ADC[22,47–49], our analysis found no significant increase in GLUT1 mRNA expression in ADC patients possessing KRAS, EGFR, BRAF, LKB1, PIK3CA or PTEN mutations compared to those who did not (Supplementary Fig. 18a). However, it should be noted that the limited number of TCGA lung ADC patients who possess rare mutations hinder a conclusive analysis. Further study within large cohorts of NSCLC will be required to assess GLUT1 expression with less-frequent oncogenic mutations. LBK1 inactivation has been linked to increased HIF-1α signalling[50,51]; however, a slight but statistically significantly lower expression of GLUT1 was detected in tumours with LKB1 mutations compared to LKB1 wild-type tumours

($P = 0.0076$, Mann–Whitney $u$-test, Supplementary Fig. 18a). Moreover, these studies did not compare GLUT1 expression or FDG uptake among the different subtypes of NSCLC. We demonstrate that LKB1-null A549 cell line exhibits higher GLUT1 expression as compared to other ADC cell lines we tested, yet all SqCC cell lines except HCC2450 express considerably higher levels of GLUT1 than A549 cells (Fig. 2a,b), and SqCC xenograft tumours (HCC95 and HCC1588) exhibit significantly higher HIF-1α induction than A549 xenograft tumours (Supplementary Fig. 16c). Importantly, as in SqCC, GLUT1 mRNA expression was also increased in ADC patients possessing genomic copy number gains of PIK3CA, although occurring at a much lower frequency than in the TCGA lung SqCC cohort (Supplementary Fig. 18b). However, GLUT1 levels in PIK3CA-amplified ADC tumours were significantly lower (3.3-fold) than SqCC tumours (Fig. 7a and Supplementary Fig. 18b), suggesting additional histology-specific regulation of GLUT1 expression in SqCC. These observations suggest a mechanistic involvement of aberrantly activated PIK3 pathway signalling in elevated glucose metabolism in NSCLC. Considering the recently described plasticity of NSCLC and potential for ADC to escape treatment through LKB1 inactivation or dysregulated pentose phosphate pathway-mediated metabolic reprogramming and squamous transdifferentiation in KL mice[52], our characterization of GLUT1-mediated glycolytic metabolism in SqCC may be an important step in identifying and preventing potential mechanisms of acquired resistance in ADC.

Enhanced influx and consumption of glucose in highly proliferative malignant cells can be utilized not only to fulfil cellular bioenergetic demands, but more importantly to provide various macromolecular building blocks and NADPH for cellular biosynthetic reactions and redox homeostasis[53]. Aberrantly elevated serine biosynthesis and pentose phosphate pathway

have been identified as critical glycolysis-diverting pathways conferring crucial metabolic benefits for tumorigenesis by providing essential intermediates for nucleotide biosynthesis and cellular reducing power in the form of NADPH[54,55]. Intriguingly, SqCC cells have elevated levels of NADPH, which was significantly reduced in lower glucose concentrations (5 mM), suggesting that heightened glucose utilization of SqCC cells may contribute to these branched metabolic pathways (Fig. 4h). Accordingly, TCGA cohort of lung SqCC shows significantly higher expression of rate-limiting enzymes for the serine biosynthesis and pentose phosphate pathways compared to ADC (Supplementary Fig. 19a,b). In addition, a recent metabolic tracing study revealed a significant increase in serine biosynthesis among NSCLC cell lines with high Nrf2 activity[56]. Interestingly, our TCGA analysis revealed greater than twofold higher Nrf2 mRNA expression in SqCC than ADC tumour samples (Supplementary Fig. 19c). Further studies to assess metabolites and metabolic enzymes involved in these pathways will be of great importance.

KRAS mutations are infrequently found in human SqCC lung tumours; this underlines an inherent limitation in capturing the distinct genetic alterations of SqCC within mouse models and individual cell lines. The KL mouse model, as well as SqCC cell line HCC1588, features a KRAS mutation whose role in driving SqCC tumour formation is not well understood. However, GLUT1 is uniformly elevated in most SqCC cell lines we tested (Fig. 2a,b), despite diverse molecular alterations present in these cell lines (Supplementary Table 3). The use of PDX models provides an excellent opportunity to explore the differential metabolism of NSCLC subtypes in an even more clinically relevant setting. Not only do PDX models retain the genetic and histopathological characteristics of their donor tumours, but distinct elevation of GLUT1 expression in SqCC is recapitulated across successive mouse-to-mouse passages (Fig. 1g). Further characterization using the PDX tumours will be crucial for validating enhanced glucose reliance and susceptibility to glycolytic inhibition in SqCC.

This study provides evidence that the elevated GLUT1-mediated glycolytic flux has clinical relevance with high potential for translational development. Our findings of increased [18]F-FDG uptake in human and murine SqCC are consistent with previous clinical studies, demonstrating that SqCC has higher [18]F-FDG-PET activity[11–13]. While increased [8]F-FDG uptake in SqCC is clearly evident, recent studies have also demonstrated that certain subtypes of ADC, such as the solid subtype, demonstrate high [8]F-FDG-PET activity as well[57]. Our analysis of [18]F-FDG uptake in NSCLC patients suggests that, while a small proportion of Acinar and Papillary ADC tumours have comparably similar SUVmax, SqCC tumours exhibit significantly increased SUVmax (Supplementary Table 1). While constitutively elevated GLUT1 expression in SqCC shows no correlation with proliferation markers Ki67 or PCNA (Supplementary Fig. 2c), a small subset of ADC tumours displays comparably high GLUT1 mRNA expression (Fig. 1a) that significantly correlates with Ki67 and PCNA mRNA levels (Supplementary Fig. 20). These data suggest that, in contrast to SqCC, GLUT1 expression in ADC may be restricted to advanced tumours, which may play a critical role in malignancy and [18]F-FDG uptake. Our clinical correlation analysis of both human NSCLC tumours and TCGA clinical data revealed a robust association of SqCC-specific high GLUT1 expression with poor tumour cell differentiation status (Table 1), supporting the prognostic potential of high GLUT1 expression in SqCC.

High prevalence of multiple genetic alterations leading to aberrantly activated PI3K/AKT signalling has been reported in SqCC[35,36]. Genetic deletion of PTEN, a phosphatase that negatively regulates PI3K/AKT activity, resulted in SqCC in LKB1-deficient mice[58], suggesting that the PI3K/AKT pathway is critically implicated in SqCC. Our TCGA analyses confirmed that genomic copy number gain, including both shallow and high-level amplifications, of PIK3CA frequently occurs in the TCGA lung SqCC cohort (Fig. 7a and Supplementary Fig. 14a; Broad Institute TCGA Genome Analysis Data Center, 2016, SNP6 Copy number analysis, GISTIC2, Broad Institute of MIT and Harvard), leading to lung SqCC exhibiting exceptionally high mRNA expression of PIK3CA among the TCGA cohorts (Supplementary Fig. 15a). Besides, a frequent genomic copy number loss of PTEN is evident within the TCGA cohort of SqCC samples (Fig. 7b and Supplementary Fig. 14b). Significantly enriched AKT and mTOR oncogenic signalling (Supplementary Fig. 14e,f), as defined by curated oncogenic signalling gene sets[59], further suggests an increased PI3K/AKT/mTOR activity in SqCC. Correlative analysis of PI3K/AKT/mTOR signalling using RPPA data with GLUT1 mRNA expression confirms a strong association in lung SqCC (Supplementary Fig. 15b). Interestingly, we observed a modest negative correlation between GLUT1 mRNA expression and PIK3CA, p-AKT, and p-S6 in lung ADC, suggesting the involvement of further histology-specific regulation of GLUT1 in ADC (Supplementary Fig. 15c). In an agreement with human SqCC, KL and xenograft SqCC tumours exhibit significantly increased AKT activity and downstream signalling directing the stabilization of HIF-1α and GLUT1 expression (Fig. 7c and Supplementary Fig. 16a–c). These results propose potential mechanisms underlying SqCC-specific GLUT1 induction by which oncogenic PI3K/AKT-mediated stabilization of HIF-1α leads to overexpression of GLUT1 and other glycolytic enzymes, which subsequently results in heightened glucose metabolism in SqCC. Importantly, these genomic alterations that lead to an increase in PI3K/AKT/mTOR signalling are among the most strongly correlated with GLUT1 expression in SqCC (Fig. 7a,b). In addition, it has been shown that mTOR inhibitor MLN0128 attenuates HIF-1α and GLUT1 expression in NSCLC cell lines and the KL model of NSCLC, which is associated with decreased [18]F-FDG uptake, further corroborating the regulation of PI3K/AKT/mTOR pathway signalling on GLUT1 expression through HIF-1α (ref. 30). Collectively, these data suggest that PI3K/AKT/mTOR signalling may represent a viable target for future therapeutic development. However, the clinical efficacy of PI3K/AKT/mTOR inhibition as a stand-alone therapy is limited in lung SqCC[60] because of potential multiple kinase signalling adaptation mechanisms[61–64]. Further preclinical and clinical assessment of resistance mechanisms in lung SqCC is needed for the development of combinatorial PI3K/AKT/mTOR-targeted therapies as well as effective screening to determine patient candidacy for treatment.

We cannot exclude the possibility of involvement of extrinsic microenvironmental factors such as hypoxic induction of HIF-1α that contributes to GLUT1 expression especially when tumours become hypoxic at the advanced stages. Although augmented HIF-1α signalling has been well associated with elevated glycolytic flux and FDG uptake in human cancers, it is evident that tissue-specific metabolic regulation may contribute to further heterogeneity in glucose metabolism. For example, Von Hippel-Lindau factor (VHL) deletion and inactivation are among the most common mutations observed in renal cell carcinoma leading to oncogenic stabilization of HIF-1α. Yet, renal cell carcinoma does not exhibit markedly elevated FDG uptake, suggesting that tissue-specific differences may further regulate glucose metabolism among different cancers that harbour similar oncogenes[65]. A recent study demonstrating significant metabolic heterogeneity being crucially regulated by the tumour microenvironment[66,67] argues for the necessity of further studies to determine whether SqCC exhibits a unique hypoxia-prone tumour microenvironment such as poor perfusion

or infiltration of metabolically active inflammatory cells, which consume high levels of oxygen, thereby causing local hypoxia.

Our results demonstrating specific anticancer effects of GLUT1 and glycolysis inhibitors on SqCC suggest enough therapeutic window for GLUT1 inhibition. Although there is potential risk of side effects in employing GLUT1 inhibitors, given that GLUT1 is the primary GLUT family member expressed in red blood cells and endothelial cells of the blood–brain barrier[16], haematologic analysis and histological evaluation of major organs including the brain revealed no apparent toxicities with the administration regimens used in this study; yet, significant anti-SqCC effects were achieved (Supplementary Fig. 21a,b and Fig. 5f,g). Despite the significant advances in lung cancer treatment with targeted therapy, exceedingly few promising options have been made for SqCC. Our study establishing aberrantly elevated GLUT1 expression and glycolytic metabolism as a specific metabolic feature of SqCC may facilitate the development of the targeted therapeutic strategy for the treatment of SqCC as well as novel diagnostic and prognostic parameters defining histological subtype-specific metabolic dependencies within NSCLC.

## Methods

**Mice.** *Lox-Stop-Lox Kras$^{G12D}$* mice[28] and *Lkb1$^{lox/lox}$* mice[68] were obtained from the Jackson Laboratory (Bar Harbor, ME, USA). All mice were backcrossed over 15 generations into the FVB/N. Lung tumours were induced by intranasal inhalation of Adenovirus-Cre[28]. All experimental procedures utilizing mice were approved by The University of Texas at Dallas and The University of California, Los Angeles, Institutional Animal Care and Use Committees.

**Human NSCLC samples.** Human lung tumour, corresponding non-malignant lung tissue specimens and clinical information were provided by the National Biobank of Korea—Kyungpook National University Hospital (NBK-KNUH). A collection of 21 SqCC, 15 ADC and paired non-tumoral frozen tissue samples were obtained for RNA extraction, as well as archived formalin-fixed paraffin-embedded samples from surgically resected lung cancer specimens of the same set of patients. All materials derived from the NBK-KNUH were obtained from patients at the time of surgery under institutional review board-approved protocols. Informed written consent was obtained from all patients and the study protocol was approved by the institutional review board of KNUH.

**Cell culture.** All NSCLC lines used in this study were obtained from the Hamon Cancer Center Collection (University of Texas Southwestern Medical Center, TX)[69] and common mutational statuses have been extensively characterized (Supplementary Table 3). Cells were cultured in high-glucose (4.5 g l$^{-1}$) DMEM (Sigma) supplemented with 10% fetal bovine serum (Sigma), 1% penicillin/ streptomycin and 1% non-essential amino acids at 37 °C in a humidified atmosphere containing 5% $CO_2$ and 95% air. All cell lines have been DNA-fingerprinted using the PowerPlex 1.2 Kit (Promega) and were found mycoplasma-free using the e-Myco kit (Boca Scientific).

**In vivo xenograft tumour experiments.** For tumour implantation, $5 \times 10^6$ cells were suspended in 50% matrigel (Corning Life Sciences) and 50% Hank's Balanced Salt Solution (HBSS, Sigma) and was injected subcutaneously into the hind flank of male nu/nu (*Foxn1$^{nu}$*) mice (Jackson Lab) between 6 and 7 weeks of age. In all, 24 mice were used for shGLUT1 knockdown xenograft experiments and 86 mice were used for the treatment of glycolytic inhibitors 2-DG or WZB117. Treatment was administered via intraperitoneal injection: 2-DG (Santa Cruz Biotechnology) 500 mg kg$^{-1}$ in PBS (Sigma) once daily and WZB117 10 mg kg$^{-1}$ in DMSO/PBS (Calbiochem) once daily. Tumour volume was measured at indicated times using electronic calipers.

**Small animal $^{18}$FDG-PET/CT imaging and analysis.** For animal $^{18}$FDG-PET and CT imaging and analysis, a standardized protocol was used[70]. Mice were made to fast for 4 h before the injection of FDG and kept warm, under gas anaesthesia (2% isoflurane) for the duration of the FDG uptake and imaging procedure. Mice were injected with 200 µCi FDG by i.p. injection. A 1-h interval for uptake was allowed between probe administration and micropositron emission tomography (microPET)/microcomputed tomography (microCT) scanning. Image acquisition and reconstruction were performed[70]. Briefly, mice were placed in a dedicated imaging chamber designed for use for the CT and both PET systems. Data were acquired by using a Siemens Inveon Preclinical Solutions microPET focus 220 for 10 min and a MicroCAT II CT (Imtek Inc.) instrument for 8 min. MicroPET images were reconstructed by using statistical maximum *a posteriori* probability

algorithms (MAP) into multiple frames. MicroCT images were reconstructed using filtered back-projection without scatter or attenuation correction. We chose a ramp filter with a cutoff frequency of 0.5 and a zoom of 5 to give a voxel size of 0.379 mm$^3$. PET and CT images were analysed using the OsiriX Imaging Software (version 3.8; OsiriX). MicroCT and PET images were reviewed blinded to score tumour burden starting at 3 weeks post-tumour induction and continuing every 2 weeks after imaging. Lesions were identified in the upper, mid and lower portions of the right and left lung lobes, then they were measured and their locations recorded. Consecutive two-dimensional regions of interest were drawn on lung lesions in coronal and axial views to detect the maximum FDG uptake. These regions encompassed the entire metabolically active tumour. The tumour diameter from microCT images was measured in the coronal and axial views, and the two values were averaged to give a mean diameter per lesion. Tumour volume was calculated using the equation $4/3\pi(r^3)$.

**Clinical $^{18}$FDG-PET imaging and analysis.** For clinical imaging, all patients fasted for at least 6 h, and their blood glucose levels were determined before the administration of $^{18}$F-FDG. Patients with blood glucose levels higher than 150 mg dl$^{-1}$ were rescheduled for a later examination, and treatment was administered to maintain a blood glucose concentration of <150 mg dl$^{-1}$ in all subjects. Patients received intravenous injections of ~5.2 MBq of FDG per kg of body weight and were advised to rest for 1 h before the acquisition of the $^{18}$F-FDG-PET/ CT image. $^{18}$F-FDG-PET/CT scans were performed using a Discovery 600 (GE Healthcare, Milwaukee, WI, USA). Before the PET scan, for attenuation correction, a low-dose CT scan was obtained without contrast enhancement from the skull base to the thigh when the patient was supine and breathing quietly. PET scans were also obtained from the skull base to the thigh at 2.5 min per bed position. PET images were reconstructed with a $128 \times 128$ matrix, an ordered-subset expectation maximum iterative reconstruction algorithm. The PET/CT images were interpreted by two experienced nuclear medicine physicians, and a final consensus was achieved for all patients. A positive finding was defined as any focus with an increased FDG uptake as compared with the surrounding normal tissue. Foci of FDG uptake due to normal physiology or benign variants, such as muscular exercise or an infectious pulmonary infiltration, were excluded from the analysis. The primary tumours underwent semiquantitative and volumetric analyses using a volume viewer software on a GE Advantage Workstation 4.5 (GE Healthcare). The SUVmax was obtained using the following formula: SUVmax = maximum activity in the region of interest (mBq g$^{-1}$)/(injected dose (mBq)/body weight (g)). Automatic measurements were used to delineate the volume of interest using an isocontour threshold method based on the SUV. Informed written consent was obtained from all patients and the study protocol was approved by the institutional review board of KNUH.

**Immunoblot.** Cells were lysed in CST lysis buffer (Cell Signaling Technology) supplemented with MG132 (40 µM; Calbiochem) and cOmplete Protease Inhibitor Cocktail (Roche). Lysates were separated by SDS–PAGE and immunoblotting performed on polyvinylidene difluoride transfer membranes (Fisher Scientific) with primary antibodies diluted in 5% BSA overnight at 4 °C. The following antibodies were used for immunoblotting: GLUT1 (1:1,000; Alpha Diagnostic; GT11-A), p63 (1:1,000; Biocare Medical; CM163A), CK5 (1:1,000; Abcam; ab52635), HIF-1α (1:1,000; BD Transduction Laboratories; 610959), AKT and Ser473-p-AKT (1:1,000; Cell Signaling; 9,272 and 9,271, respectively), β-actin (1:5,000; Sigma; A5441). Horseradish peroxidase-conjugated secondary antibodies (Santa Cruz) were diluted 1:5,000 in 5% skim milk, followed by detection with SuperSignal West Femto or Pico substrate kits (ThermoFisher). Uncropped images of immunoblots are provided as Supplementary Fig. 22.

**Immunocytochemistry.** Cells were seeded on coverslips and allowed to adhere overnight. Cells were then fixed in 4% paraformaldehyde (PFA) followed by permeabilization with 0.5% Triton X-100. Primary antibodies were diluted in 3% BSA and applied overnight at 4 °C. The following primary antibodies were used: GLUT1 (1:250; Alpha Diagnostic; GT11-A) and p63 (1:200; Biocare Medical; CM163A). Fluorophore-conjugated secondary antibodies were used to visualize primary antibody staining (1:400; Life Technologies; A-11070, T-6390). Cells were counterstained with 4,6-diamidino-2-phenylindole (DAPI) and mounted using Vecta-shield Mounting Medium (Vector Labs) and viewed under a fluorescent microscope (Nikon Eclipse Ni-U).

**IHC and immunofluorescence.** Lung cancer patient tissue microarrays were obtained from Protein Biotechnologies. KL and xenograft mice were perfused with PBS/10 mM EDTA followed by 4% PFA. Tissues were removed and further fixed in 4% PFA overnight followed by paraffin embedding. Tissue sections were stained with haematoxylin and eosin in accordance with standard methods. For IHC, tissue sections (5 µm) were subjected to heat-mediated antigen retrieval (citrate buffer, pH 6). Primary antibodies were applied and incubated at 4 °C overnight. Vectastain ABC (Vector Labs), EnVision kit (Dako) or TSA kit (PerkinElmer) with DAB substrate (Vector Labs) were used to amplify and visualize primary antibody staining according to the manufacturer's protocol. For immunofluorescence, primary antibodies were visualized through Alexa Fluor 488 (AF488)-conjugated

secondary antibodies (1:400; Life Technologies; A-11070) or Texas Red (TxRed)-conjugated streptavidin (15 µg ml$^{-1}$; Vector Labs; SA-5006) followed by counterstaining and mounting with Vectashield Mounting Medium with DAPI (Vector Labs). The following primary antibodies were used: GLUT1 (1:250; Alpha Diagnostic; GT11-A), p63 (1:200; Biocare Medical; CM163A), CK5 (1:200; Abcam; ab52635), TTF1 (1:1,000; Dako; M3575), HIF-1α (1:1,000; Novus Biologicals; NB100-449), Ki67 (1:500; Cell Signaling; 12202), Cleaved Caspase-3 (CC3, 1:200; Cell Signaling; 9664), Ser473-p-AKT (1:200; Cell Signaling; 4060), Ser235/236-p-S6 (1:200; Cell Signaling; 4858) and Thr37/46-p-4EBP1 (1:200; Cell Signaling; 2855). Images were taken with a Nikon Eclipse Ni-U microscope through the NIS Elements imaging software (Nikon) and quantified through ImageJ (NIH).

**Patient-derived xenografts.** Surgical specimen samples from lung cancer patients were obtained from Osaka Medical Center for Cancer and Cardiovascular Diseases, with the patients' informed consent. Animal studies were performed in compliance with the guidelines of the institutional animal study committee of Osaka Medical Center. Primary xenograft tumours were generated by inoculating small pieces of patient tumours into immunodeficient mice[27]. Surgically resected tissues were minced and washed with HBSS (Invitrogen). Specimens were digested in DMEM supplemented with 0.26 U ml$^{-1}$ Liberase DH (Roche) and 1% penicillin/streptomycin (Invitrogen) at 37 °C for 1–2 h. Digested tissue suspensions were passed through 500- and 250-µm metal mesh filters for removal of undigested fragments. Suspensions were further filtered through 100- and 40-µm cell strainers (BD Falcon). Fragments on the cell strainer and cells in the flow-through fractions were collected separately, and each were washed with HBSS and cultured in StemPro hESC medium (Gibco). After processing, the tumour fragments spontaneously form cancer-tissue-originated spheroids. One hundred cancer-tissue-originated spheroids were then suspended in 50 µl of matrigel (Corning Life Sciences) and subcutaneously transplanted into nonobese diabetic severe combined immunodeficient mice.

**Quantitative RT–PCR.** Cells were lysed with TRI reagent (Sigma) and RNA extraction performed with the Direct-zol RNA MiniPrep kit (Zymoresearch) according to the manufacturer's instruction. For quantitative RT–PCR, RNA was mixed with iTaq Universal SYBR Green One-Step Kit (Bio-Rad) supplemented with iScript Reverse Transcription Supermix (Bio-Rad). Quantitative PCR was performed with the CFX-96 real-time PCR System (Bio-Rad). Quantitative real-time PCR primers are provided in Supplementary Table 4.

**shRNA and siRNA knockdown.** pLKO.1-shRNA-HIF-1α (Mission TRC shRNA, TRCN0000003810), shRNA-GLUT1#1 (Mission TRC shRNA, TRCN0000043583) and shRNA-GLUT1#2 (Mission TRC shRNA, TRCN0000423590) were obtained from Sigma. For lentivirus production, HEK 293 T cells were transfected with pLKO.1 shRNA, psPAX2 and pMD2.G using Lipofectamine 3000 (Invitrogen). Cells were incubated with viral supernatant containing 8 µg ml$^{-1}$ polybrene. pLKO.1-shRNA-GFP was used as a control vector. Reverse transfection with a pool of four specific siRNAs targeting GLUT1 (L-007509-02-0005, ONTARGETplus SMARTpool; Dharmacon) was performed using siRNAs complexed with Lipofectamine 3000 (Invitrogen). Targeting sequences for all shRNA and siRNAs are provided in Supplementary Table 5.

**Cellular metabolic analysis.** Glucose uptake was analysed using Glucose Uptake Cell-Based Assay Kit (Cayman) according to the manufacturer's instruction. Cells were incubated in glucose-free DMEM (Gibco) containing a fluorescently labelled deoxy-glucose analogue, 2-NBDG at 37° for 1 h. Cells were prepared as per the manufacturer's protocol and emission at 535 nm was measured using flow cytometry (SH800, Sony) and a fluorescent microscope (Nikon Eclipse Ni-U). Generated histograms were overlaid using FlowJo and fluorescent intensity was quantified via ImageJ. For extracellular lactate measurement, cells were cultured in pyruvate-free DMEM (Gibco) containing 5 or 25 mM glucose. Media was collected after 48 h and lactate concentration was measured using the L-Lactate Assay Kit I (Eton Bioscience) as per the manufacturer's instructions. OCR was measured using the Extracellular Oxygen Consumption Assay (Abcam). Cells were prepared in a 96-well plate and OCR was measured through a fluorescent microplate reader at 650 nm for 2 h (Biotek Cytation 5) as per the manufacturer's protocol. Cellular ATP levels in cells cultured in 25 or 5 mM glucose media (DMEM, Gibco) or treated with 50 µM WZB117, or 25 mM 2-DG were measured using an ATP detection kit (Invitrogen) according to the manufacturer's instructions. Cellular NADH and NADPH levels were assayed using NAD + /NADH-Glo or NADP/NADPH-Glo Assay Kit (Promega). Cells were plated on a 96-well plate in pyruvate-free media containing 25 or 5 mM glucose. Cells were then allowed to proliferate for 24 h, lysed and assayed in 1% dodecyltrimethylammonium bromide. Cellular NADH and NADPH levels were assayed based on the manufacturer's instructions and luminescence was measured using a Biotek Cytation 5 plate reader. Raw luminescence was then normalized to cell count measured by a TC20-automated cell counter (Bio-Rad).

**Extracellular metabolic flux analysis.** ECAR and OCR were measured using a Seahorse XFp Analyzer (Agilent). Cells were plated in appropriate growth medium the day prior to the assay and allowed to adhere overnight. On the day of the assay, the cells were washed and incubated in glucose- and pyruvate-free Seahorse XF Base Medium supplemented with 2 mM glutamine for 1 h in a 37 °C non-CO$_2$ incubator. Standard assay settings were used to measure ECAR and OCR prior to and after the addition of 5 mM glucose. To measure changes in ECAR upon WZB117 treatment, escalating doses of WZB117 (50 then 100 µM) were administered using the standard assay settings. ECAR and OCR were normalized to cell count following the assay.

***In vitro* glycolysis inhibition and flow cytometry analyses.** Cell death was measured using the PE Annexin-V Apoptosis Detection Kit I (BD Pharmingen) according to the manufacturer's instruction. Cells were treated with 25 mM 2-DG or 50 µM WZB117 in 5 mM glucose, pyruvate-free DMEM (Gibco) for 48–72 h. Cells were collected and stained with annexin-V and 7-AAD as per the manufacturer's instructions. Cells were analysed by flow cytometry (SH800, Sony) and FlowJo software.

**TCGA analysis.** Level 3 mRNA-sequencing gene expression data from the TCGA was obtained through the Broad Institute's Firebrowse data portal for the TCGA SqCC and ADC cohorts. In total, 501 lung SqCC and 517 lung ADC tumour samples were included in our analysis. We additionally included 51 lung SqCC and 59 lung ADC tumour-paired normal tissue samples in our analysis. The obtained gene expression profiles were produced on the Illumina HiSeq platform and constructed using the RSEM package to determine gene expression in terms of transcripts per million mappable reads. The expression profiles were further quartile-normalized for comparability between data sets. We employed R and Excel to produce a global gene expression set in which each gene is ranked by the difference in average expression (expressed in transcripts per million mappable read) between lung SqCC samples and lung ADC samples. Visualization of differential gene expression by heatmap was produced using the Broad Institute's GenePattern HeatMapImage module using row normalization on gene sets of interest. We then used GSEA to determine gene set and pathway enrichment within DEGs between lung SqCC and lung ADC. Using the pre-ranked GSEA package, a ranked global gene list, produced from the TCGA level 3 mRNA-sequencing data, was used to identify the core enriched KEGG and oncogenic signature gene sets in lung SqCC. A minimum of 10 and a maximum of 500 genes per set were used as the parameters to define gene set enrichment. Pearson parametric and Spearman nonparametric correlation analyses were performed in Prism GraphPad. Somatic Copy number analysis was carried out using the GISTIC2 algorithm on level 3 SNP6 data obtained from the Broad Institute's Firebrowse portal. For this analysis, 501 lung SqCC patient tumour samples with SNP6 genomic copy number data were used. Copy number variation calls were made from the gene-level threshold GISTIC2 output. The GISTIC algorithm takes into account both high and low thresholds for copy number determination across all the input samples to assign significance to copy number variation. Correlative analysis of SNP6 copy number variation data with mRNA gene expression was performed in Excel and R using linear regression. Level 3 RPPA data were obtained through the Broad Institute's Firebrowse portal.

**Statistical analyses.** Statistical analysis was carried out in Excel, R, Prism GraphPad 6 and SAS. All results were expressed as mean ± s.e.m. or median ± the interquartile range. A P value of 0.05 via two-tailed Student's t-test of unknown variance, Mann–Whitney U-test, $\chi^2$-test, one-way ANOVA or two-way ANOVA was considered statistically significant for hypothesis testing. Multiple comparison testings were accompanied with ANOVA to identify the source of significant variance among the data.

**Data availability.** The mRNA sequencing, genomic copy number analysis data, DNA mutational data, RPPA and clinical cohort information referenced during the study are available in a public repository from the TCGA website (www.cancer-genome.nih.gov). All TCGA data used in the study, including mRNA-sequencing gene expression profiles, Gistic2 genomic copy number analysis, DNA mutational data, RPPA and clinical cohort information were obtained through the Firebrowse data portal (www.firebrowse.org). The authors declare that all the other data supporting the findings of this study are available within the article and its supplementary information files and from the corresponding author upon reasonable request.

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

## Acknowledgements

We thank Jennifer Cao for technical support. This work was supported by the UT Dallas start-up fund and American Lung Association LCD-400239 (J.-w.K.), NIH R01 CA163649 and SPORE 2P50 CA127207 (P.K.S.), NIH K25 AR063761 (M.C.), Project for Development of Innovative Research on Cancer Therapeutics, Japan Agency for Medical Research and Development, JSPS Kakenhi JP26111005 and Takeda Science Foundation (M.I.), JP25461937 (H.E.), Grant-in-Aid for Scientific Research on Promotion of Science, 16K09493, on Innovative Areas, 26111003, JST PRESTO, the Banyu Life Science Foundation International and Takeda Science Foundation (N.T.), Welch Foundation, AT-1595 (J.-M.A.), NIH, NCI P50CA70907 (SPORE), Cancer Prevention and Research Institute of Texas (CPRIT) RP110708 (J.D.M.).

## Author contributions

Conceptualization: J.-w.K., J.G., M.L.N. and S.Y.L.; methodology: J.-w.K., S.Y.L., N.T., M.I. and D.B.S.; software: Z.X. and M.L.M.; formal analysis: Z.X., M.C. and M.L.N.; investigations: J.G., M.L.N., S.Y.L., J.H.C., H.C., D.M.R.J., R.J.R., M.W.R., J.Y.J., M.W., H.A., H.E. and K.L.H.; recourses: J.-M.A., P.K.S., J.D.M. and D.B.S.; writing—original draft: J.-w.K., J.G., M.L.N. and S.Y.L.; writing—review and editing: J.-w.K., J.G., M.L.N., S.Y.L., P.K.S. and J.D.M.; funding acquisition: J.-w.K., P.K.S., M.C., M.I., N.T., J.-M.A. and J.D.M.; supervision: J.K.

## Additional information

**Competing interests:** The authors declare no competing financial interests.

