## [Peer Review File · Nature Communications]

Reviewers' comments:

Reviewer #1(Remarks to the Author):

In this report the authors investigate differences in metabolic signatures between the two most predominant subtypes of non-small cell lung cancer (NSCLC); adenocarcinoma (ADC) and squamous cell carcinoma (SQCC). Specifically they find a unique glucose metabolism signature that differs between these 2 subtypes, which they characterize in the manuscript. The authors find that GLUT1 is elevated both transcriptionally and translationally in SQCC compared to ADC and is associated with enhanced glucose uptake, increased glycolysis and enhanced sensitivity to glycolytic inhibition. A number of their findings are validated in a number of experimental systems, from TCGA analysis to patient sample and mouse models. Mechanistically the authors link enhanced GLUT1 expression to enhanced PI3K/AKT signaling the drives HIF1a expression in SQCC.

General comments

SQCC is a distinct and aggressive subset of NSCLC that has an unmet clinical need. Understanding specific metabolic liabilities of SQCC is required for the design of novel therapies to treat this specific lung cancer subset. Overall this is a technically solid and well-put together manuscript that uses a multifactorial approach (TCGA analysis, PDX models, cell line analysis, patient sampling and mouse modeling) that enhances the quality and robustness of the work. Some additional experiments are required to strengthen the study, however the work presented in this manuscript will be of interest to the Nature Communications viewership.

Major comments

1. Work using RNAi to investigate GLUT1 knockdown is underdeveloped. The authors only use 1 cell line for shRNA knockdown of GLUT1. Findings need to be validated in at least one (preferentially two) additional SQCC cell lines. Use of a second hairpin would also strengthen their findings. Additional work characterizing cytotoxicity of shGLUT1 must also be performed. e.g. is there induction of apoptosis with GLUT1 knockdown?
2. Related to the above point, the authors need to perform shRNA knockdown of GLUT1 in two ADC cell lines to show that these cells are unaffected by GLUT1 knockdown using the same assays.
3. For all mouse experiments the individual growth curves of the mice need to be presented in the supplemental figures.
4. Work using the GLUT1 inhibitors is underdeveloped and the specificity of these GLUT1 inhibitors is questionable. Mechanistically, how do these drugs inhibit GLUT1? Is glucose uptake reduced upon treatment with these drugs in SQCC (analogous to shGLUT1)? Is glycolysis inhibited with these drugs?

Minor comments

1. Additional information on the PDX model is needed. Are these subcutaneous tumors?

Reviewer #2 (Remarks to the Author):

General comments

This is a well written, systematic and highly detailed analysis of glucose metabolism of NSCLC and

specifically the role of Glut1. The results convincingly demonstrate an important role of Glut1 in squamous cell lung cancer.

While the role of Glut1 in cancer metabolism has been studied by many investigators, the results are conflicting and incomplete. Therefore, this comprehensive analysis in a specific tumor type is novel and of significant interest.

The authors establish the role of Glut1 in squamous cell carcinoma by a variety of studies at the cellular level, in patient derived xenografts, analysis of patient derived tumor samples, and clinical imaging data of lung cancer patients.

The data are well presented and analyzed using established statistical techniques. The comprehensive preclinical and clinical data allow the authors to make robust conclusions regarding the importance of Glut1 and glycolysis for the viability and growth of squamous cell carcinomas.

One limitation is that the authors present lung adenocarcinoma as a homogeneous disease. However, there are marked differences in glucose metabolic activity, growth pattern and prognosis for the various subtypes of lung adenocarcinoma (lepidic, acinar, papillary, micropapillary, solid). This should be addressed in the discussion.

Another aspect that should be discussed in more detail is the (potential) importance of other glucose transporters, specifically Glut-3 in lung adenocarcinomas.

Specific comments

Introduction

Page 3, paragraph 1

"to date, no successful achievement in the development of a targeted therapy for lung SqCC has been made"

Since the approval of Necitumumab for treatment of squamous NSCLC this is not entirely correct and should be reworded.

Page 3, paragraph 2

"the differential usage of metabolic pathways in NSCLC subtypes has not been assessed"

While it is correct that detailed studies on this subject are lacking, the lower FDG uptake and glucose metabolic rates of lung adenocarcinomas as compared to squamous cell carcinomas are known since several years, see Brown et al. J Nucl Med 1999.

Page 3/4

In this discussion the authors may want to add that there appear to be tissue specific differences in the regulation of glucose metabolism by the oncogenes discussed here. For instance, Hif-1 activation in renal cancer does not induce marked FDG uptake in this tumor type. Therefore, it is not obvious if the results obtained in other tumor types (e.g. colon cancer in reference 14) can also be applied to NSCLC.

Results

Page 7, paragraph 2

The meaning of the following statement is not clear "we analyzed GLUT1 expression in human SqCC (n=19-21)". Were there 19 or 21 samples?

Page 11, last paragraph

"GLUT1-mediated glucose transport is considered a rate-limiting step for glycolysis in several cancers and benign tissues"

The "rate limiting step" of glycolysis in cancer cells has been discussed for decades with no generally accepted conclusion. Therefore, I would suggest to remove this sentence which does not add much to the presentation of the results. It could be changed to something like, "We next

investigated if GLUT1 is the rate limiting step of glycolysis in squamous NSCLC"

Page 12, last paragraph

"... suggesting that SqCC is essentially reliant on glucose metabolism to support cellular bioenergetics".

I don't quite understand this conclusion. Couldn't the change in glucose uptake for squamous cell carcinoma cells indicate that these cells have a very high capacity for glucose uptake whereas glucose uptake becomes saturated above physiologic concentrations in adenocarcinomas? This would be in line with the markedly higher expression levels of glucose transporters in squamous cell carcinomas.

Page 13, first paragraph

Is 5 mmol/l is the physiologic plasma glucose concentration in human and 25 mmol/l is severely hyperglycemic. Is it appropriate to call a concentration of 5 mmol/l glucose deprivation or would it be more appropriate that supra-physiologic glucose concentrations stimulate growth of squamous cell carcinoma cell lines?

Page 14-15

Do 2-DG, WZB117, and STF-31 block other glucose transporters and specifically GLUT-3 with the same affinity as GLUT-1?

Page 15, paragraph 1

GLUT-1 is the major glucose transporter at the blood brain barrier. Did treatment with GLUT-1 inhibitors cause cerebral side effects?

Page 16, paragraph 1

"... and further suggest that 18 F-FDG-PET imaging can aid in designing more feasible diagnostic strategies in differentiating SqCC from other types of lung cancer"

This has been studied rather extensively clinically, but the overlap in FDG uptake between adeno and squamous cell carcinomas appears too large. The authors should therefore consider to delete this statement.

Discussion

Page 20

"Conversely, GLUT1 expression and glycolytic flux remain relatively minimal in ADC, indicative of glucose independency"

This should be formulated more carefully given the various subtypes of lung adenocarcinomas. Some of the subtypes demonstrate equally high FDG uptake on PET/CT scans as squamous cell carcinomas.

Page 21

The meaning of the following sentence is not entirely clear: "ADC may rewire its metabolism to acquire glucose independence in response to glucose deprivation or glycolytic inhibition"
What is meant by "rewiring"?

Page 21-22

I would suggest to shorten the rather speculative discussion on glucose metabolism of adenocarcinomas.

Page 24

"refined noninvasive diagnostic imaging techniques which specifically exploit elevated glucose influx will substantially improve the clinicopathological identification of SqCC from other types of lung cancer."

Considering the overlap in FDG uptake of squamous and adenocarcinomas this appears doubtful.

Figures

Figure 2a

It would be helpful to add information on expression of Glut-3 in the studied cell lines, because Glut-3 is believed to be a major glucose transporter in adenocarcinomas as also indicated by figure 1b.

Supplemental information

Please add brief descriptions on the methods used for small and animal and clinical FDG PET/CT imaging.

Reviewer #3 (Remarks to the Author):

The manuscript by Goodwin et al, entitled, "Distinct Metabolic Phenotypes within Non-small Cell Lung Cancer Define Selective Vulnerability to Glycolytic Inhibition of Lung Squamous Cell Carcinoma," describes an integrative analysis of lung adenocarcinoma and squamous carcinoma using data from patient samples, xenografts, GEMMs, and cell lines. This analysis highlights glycolytic uptake as core metabolic features of SCC, which renders it sensitive to inhibition of glycolysis by blocking the GLUT1 transporter or glycolysis by other means. The study explores an important topic, as new therapeutic targets are certainly needed for lung squamous cancer. The manuscript is well written, and the experimental methods for the preclinical studies appear sound, although there are significant questions regarding statistical analyses of clinical specimens highlighted below. The identification of differences in metabolic profiles and targeting strategies between lung ADC and SCC is notable, although the novelty of this finding is overstated by the authors a bit as there are multiple prior publications showing differences between ADC and SCC in terms of FDG avidity and GLUT1 levels. The study impressively integrates preclinical, clinical, and imaging data.

There are a number of concerns regarding the study, highlighted below.

1. -page 16: the authors state "We confirmed the genomic amplification of PIK3CA (Supplementary Fig. 11a) and correlatively increased PIK3CA mRNA expression (data not shown) in the vast majority of the TCGA lung SqCC cohort". On page 24 it is stated "In agreement with these studies, our TCGA analyses found copy number gain of PIK3CA with amplification of more than 2 copies in 47% of the SqCC samples. Heterozygous loss of PTEN was observed in over 50% of the SqCC samples. (Fig. 7b; Supplementary Fig. 11a-d).

This statement does not seem to accurately reflect the TCGA results: in the original TCGA paper (Nature, 2012), PTEN and PIK3CA were altered in 15 and 16%, respectively. Furthermore, in Figure 7B, it seems that more than 80% have PIK3CA amplification based on the number of dots present (this may be an underestimate- I couldn't count them all). The authors should clarify why their analysis is markedly different than the TCGA, as well as the discrepancy between figure 7B and the 47 or 50% numbers stated in the text. (Note that >2 copies is not typically considered amplified; a higher cutoff is usually used).

2. The suggestion that GLUT1 is upregulated in LUSC, but not in lung ADC ("We show that GLUT1 is remarkably and uniquely elevated at both the mRNA and protein levels in lung SqCC as the principal cellular glucose transporter, but is minimally expressed in lung ADC", page 4) does not appear to take into account subgroups of lung ADC in which these pathways are known to be upregulated. Indeed, the authors note that KRas and BRAF mutations, which occur in lung ADC but not SCC, upregulate GLUT-1 expression (refs 13 and 14).

Furthermore, prior studies have established that LKB1 loss is associated with HIF1-pathway upregulation (e.g. Ji et al, Nature 2006), a finding which was extended in a study (led by a

coauthor of the current study, DBS) showing that LKB1 loss is associated with HIF1a upregulation, GLUT1 upregulation, and increased FDG uptake (Shackelford et al, PNAS 2009). LKB1 loss occurs predominantly in lung adenocarcinoma.

An analysis should be done comparing levels of GLUT1 and other relevant targets here in lung SCC vs specific subgroups of lung ADC, including, BRAF mutants (although the numbers in the TCGA are modest), LKB1 mutants, PIK3CA/PTEN mutants (or PI3K/AKT activated at the protein level, given the proposed role for this pathway in regulating HIF1a). This can be done using the TCGA as well as other publicly available datasets. Furthermore, in supplemental figure 14, it should be clarified as to whether only Kras G12D was analyzed as indicated; if so, this analysis should be changed to include all KRas mutants, given prior data linking Kras with Glut1.

3. The SCCs observed in the KL mice does not seem to be representative of phenotypes observed in human LKB1-mutant lung cancer, which are typically lung ADC without any mixed component. It is thought this squamous differentiation may be due, at least in part, to redox imbalance (Li et al, Cancer Cell 2015) which may differ in human tumors because of additional mutations that typically accompany KL tumors (e.g. Keap1). This raises the possibility that the squamous differentiation is a murine GEMM-specific phenomenon and is a marker for tumors under greater oxidative stress, which could be accompanied by increased GLUT-1 and sensitivity to glycolysis inhibitors.

4. Figure 3B: the impact of shGLUT1 in HCC95 cells should be tested in at least one (but preferably more) SCC line(s), and compared with additional lung ADC lines, to see if the inhibition in cell and tumor growth is truly SCC specific (3B,F).

5. page 16: the finding that SCC has higher FDG uptake, GLUT1 expression, and other glycolysis markers has been reported previously (e.g. as examples: references 55, 56, and 57; as well as Schuurbiens, J Thorac Oncol. 2014 Glucose metabolism in NSCLC is histology-specific and diverges the prognostic potential of 18FDG-PET for adenocarcinoma and squamous cell carcinoma, which is not cited; or Choi et al, Technol Health Care. 2015, 26410497). These studies all support greater GLUT1 and glycolysis in lung SCC, which diminishes the novelty of a main point of the paper. They overstate the novelty of their finding further on page 25: "This study is, to our knowledge, the first implication of differentially utilized and targetable metabolic pathways in NSCLC histological subtypes" as the aforementioned papers and others have demonstrated histology-based differences in GLUT1 and other metabolic pathways.

6. page 16: the authors state "Collectively, these results demonstrate that 18F-FDG uptake is significantly increased in SqCC tumors, reflecting high levels of GLUT1 expression and glucose uptake, and further suggest that 18F-FDG-PET imaging can aid in designing more feasible diagnostic strategies in differentiating SqCC from other types of lung cancer." Also, on page 24 "...developing more refined noninvasive diagnostic imaging techniques which specifically exploit elevated glucose influx will substantially improve the clinicopathological identification of SqCC from other types of lung cancer." It is unclear how specifically FDG-PET would do this, as the overlap in the SUV between ADC and SCC shown in 6F is substantial; I expect an ROC curve which show little diagnostic utility. In the absence of such an analysis, a statement about its diagnostic utility should be removed. (As a separate note, it is unclear we need more "feasible" diagnostic strategies for distinguishing ADC from SCC- current strategies seem quite feasible. This section could benefit from review by an experienced lung cancer clinician).

Minor corrections:

1. Page 21: glucose "depravation" should be "deprivation" (unless glucose is being linked to moral corruption).

We have greatly expanded our RNAi studies to include additional SqCC as well as ADC cell lines. Using two different hairpins we have separately targeted the GLUT1 3'UTR and CDS, and extensively characterized the proliferation, apoptosis, cell death, glucose uptake, and intracellular ATP, NADH, and NADPH levels. We further supplemented our shRNA study with transient siRNA knockdown of GLUT1. We measured the glucose uptake and proliferation of these siGlut1 knockdown cell lines and observed results consistent with our shRNA studies. Acquired information from additional GLUT1 knockdown experiments reinforces our original finding of SqCC-specific cytotoxic effects of GLUT1 knockdown and we have revised manuscript and figures accordingly.

We have as well offered a reinterpretation of our clinical analysis of differential ¹⁸F-FDG uptake among NSCLC patients to support and highlight the distinct glycolytic heterogeneity observed within NSCLC. We have also clarified our analysis of genomic PIK3CA amplification in lung SqCC, addressing the deviation from previously reported findings by the TCGA, which reported only high level, focal amplifications and homozygous deletions.

We supplemented our Glut1 inhibitor studies by examining glucose uptake and extracellular acidification rate (ECAR, Seahorse analyzer) upon treatment with Glut1 inhibitors WZB117 and STF31. Intriguingly, WZB117 was shown to inhibit both glucose uptake and ECAR as expected, however STF31 treatment failed to suppress either glucose uptake or ECAR. Because we could not demonstrate the direct or indirect inhibition of GLUT1 via STF31, we believe that it would be appropriate to omit any data pertaining to STF31 in our study, however we have included all of the STF31 data as a reviewer only figure.

Response to Reviewer 1

✓1. Work using RNAi to investigate GLUT1 knockdown is underdeveloped. The authors only use 1 cell line for shRNA knockdown of GLUT1. Findings need to be validated in at least one (preferentially two) additional SQCC cell lines. Use of a second hairpin would also strengthen their findings. Additional work characterizing cytotoxicity of shGLUT1 must also be performed. e.g. is there induction of apoptosis with GLUT1 knockdown?

We thank the reviewer 1 (and the reviewer 3, comment No.4) for this comment. We performed shRNA knockdown of GLUT1 in two additional SqCC cell lines, HCC1588 and HCC2814. Furthermore, we used a second hairpin targeting the CDS (shGLUT1 #2) as well as the original hairpin targeting the 3'UTR (shGLUT1 #1) for all three SqCC cell lines. We repeated the key experiments characterizing the proliferation, glucose uptake, and change in intracellular ATP, NADH, and NADPH induced by GLUT1 knockdown. Additionally, we used 7-AAD and Annexin V staining to look at cell viability and apoptotic induction after GLUT1 knockdown. We selected HCC2814 as an additional SqCC cell line because it features an amplification of PIK3CA and lacks a Kras mutation, which are representative features of human SqCC.

Consistent with HCC95 cell line, GLUT1 knockdown in additional SqCC cell lines, HCC1588 and HCC2814 dramatically suppressed the proliferative capacity (Figures 3a, b, Supplementary Figures 8d-e). This was accompanied by extensive apoptosis and cell death, observed by Annexin V and 7-AAD staining, respectively (Figure 3c, Supplementary Figures 8f). The uptake of fluorescently labeled

glucose was significantly decreased in GLUT1 knockdown cells, which correlated with lower intracellular ATP, NADH, and NADPH (Figures 3d-f, Supplementary Figures 8g,h).

GLUT1 knockdown via shRNA was specifically cytotoxic in SqCC cell lines with extensive apoptosis and cell death observed compared to shGFP control cells, so much so that we could not select for a stable SqCC GLUT1 knockdown cell line for *in vivo* xenograft tumor assay, aside from HCC95. We are currently establishing inducible GLUT1 knockdown cell lines, which will be reported in a subsequent publication.

Due to how closely related comment 1 and 2 are, we have revised the manuscript with the combined results of both comments, which are presented under comment 2.

✓2. *Related to the above point, the authors need to perform shRNA knockdown of GLUT1 in two ADC cell lines to show that these cells are unaffected by GLUT1 knockdown using the same assays.*

In addition to GLUT1 knockdown in SqCC cell lines as described above, we performed GLUT1 knockdown in two ADC cell lines, A549 and H522, with the same hairpins used to knockdown GLUT1 in SqCC cell lines. We characterized the effects of GLUT1 knockdown in these ADC cell lines by measuring *in vitro* proliferation, cell viability, glucose uptake and intracellular ATP levels. GLUT1 knockdown moderately suppressed the proliferation of A549 and H522, however, no apparent increase in the population of apoptotic or dead cells was noticed (Supplementary Figures 7a-c). After validating reduced uptake of fluorescently labeled glucose, we measured intracellular ATP and found no significant difference between shGFP control ADC cells and shGLUT1 knockdown cells (Supplementary Figures 7d, e). This suggests that upon reduced glucose uptake ADC cells are able to maintain intracellular ATP levels. We then implanted A549 and H522 shGFP and shGLUT1 knockdown cells into nude mice to determine if GLUT1 deficiency affected ADC tumor growth. We observed no significant difference in tumor growth rate in either ADC tumors (Supplementary Figures 7f,h).

In addition to our shRNA studies, we transiently knocked down GLUT1 using siRNA in SqCC cell lines HCC95 and HCC1588, and in ADC cell lines A549 and H522, and we repeated the *in vitro* proliferation and glucose uptake experiments. Consistently, the effect of transient siGLUT1 knockdown was more pronounced in SqCC cell lines, with a complete suppression of *in vitro* proliferation in SqCC but not ADC cells (Supplementary Figures 8a, b). Intriguingly, A549 was more sensitive to transient GLUT1 knockdown than H522, in accordance with previous reports that loss of LKB1 sensitizes cells to metabolic disruption (*Momcilovic et al Cancer Res 2015*), however both ADC cell lines continued to proliferate even after depletion of GLUT1. We measured fluorescent glucose uptake across all four cell lines transfected with either scrambled or GLUT1 targeting siRNA. Glucose uptake was dramatically higher in SqCC cells compared to ADC cells, and was accordingly reduced upon GLUT1 knockdown in all cell lines (Supplementary Fig 8c).

Collectively, our expanded RNAi data using sh- and siRNA to knockdown GLUT1 in ADC and SqCC cells demonstrates a specific reliance on GLUT1-mediated glucose uptake and bioenergetics in SqCC compared to ADC. We have updated the results section to reflect the results of these experiments as follows:

Page 10, Line 21 – Page 12, Line 15

(Original)

“Genetic GLUT1 inhibition impairs cell viability and in vivo tumor growth of Lung SqCC.

Given the high expression level and critical roles for cellular glucose uptake, we reasoned that GLUT1 may be necessary for cell viability and growth of SqCC. Viability of SqCC HCC95 cell lines was

significantly reduced by lentiviral-mediated GLUT1 knockdown compared to shGFP control cells even in high-glucose (25mM) conditions (Fig. 3a,b). We suspected that GLUT1 knockdown attenuates SqCC proliferation due to decreased glucose uptake. Indeed, we found that glucose uptake of HCC95 cell lines was dramatically decreased by 70% in GLUT1 knockdown cells compared to shGFP control cells (Fig. 3c,d). Furthermore, GLUT1 knockdown reduced intracellular ATP and NADH pools indicating increased energetic stress resulting from decreased glucose availability (Fig. 3e). Decreased intracellular NADPH was also observed upon GLUT1 knockdown, suggesting an essential role of GLUT1 for providing glucose for glycolytic branching pathways such as pentose phosphate pathway in SqCC (Fig. 3e). Overall, this data shows that GLUT1 expression and function is vital for maintaining energy homeostasis and proliferation in SqCC HCC95. To determine whether GLUT1 knockdown exerts SqCC tumor growth inhibition *in vivo*, we implanted shGLUT1 knockdown and shGFP control HCC95 SqCC cells into nude mice. In accordance with *in vitro* proliferation, GLUT1 knockdown significantly inhibited HCC95 xenograft tumor growth (Fig. 3f). Consistently, siRNA-mediated knockdown of GLUT1 expression in HCC1588 SqCC cells resulted in growth inhibition (Supplementary Fig. 7a,b). These results demonstrate the necessity of GLUT1 expression in SqCC cells for bioenergetic homeostasis and tumor growth.”

(Revised)

“Genetic GLUT1 inhibition impairs cell viability and *in vivo* tumor growth of Lung SqCC.

Given the high expression level and critical role in cellular glucose uptake, we reasoned that GLUT1 may be necessary for cell viability and growth of SqCC. We performed shRNA-mediated knockdown of GLUT1 in SqCC HCC95 and HCC1588 cell lines using two different GLUT1 targeting sequences (Fig. 3a). GLUT1 knockdown dramatically suppressed the proliferative capacity of SqCC cell lines compared to shGFP control cells even in high-glucose (25mM) conditions (Fig. 3b). This was accompanied by extensive apoptosis and cell death, which we observed by Annexin V and 7-AAD staining, respectively (Fig. 3c). In contrast, GLUT1 knockdown in ADC cell lines A549 and H522 only moderately suppressed proliferation with no induction of apoptosis or cell death (Supplementary Fig. 7a-c). We suspected that the decline in cell viability in SqCC cell lines was a result of a bioenergetic crisis resulting from GLUT1 inhibition. Fluorescently labeled glucose uptake was significantly decreased in GLUT1 deficient cells, which was correlated with lower intracellular ATP, NADH, and NADPH, suggesting not only disrupted bioenergetics, but an essential role for GLUT1 dependent flux of glucose intermediates into glycolysis dependent pathways such as the pentose phosphate pathway (Fig. 3d-f). In contrast to SqCC, reduced glucose uptake did not affect cell viability or intracellular ATP in ADC cells (Supplementary Fig. 7d,e). Consistently, transient GLUT1 knockdown using siRNA was more pronounced in SqCC cell lines, with a complete suppression of proliferation in SqCC but not ADC cells despite reduced glucose uptake in all four cell lines (Supplementary Fig. 8a-c). Intriguingly, A549 was more sensitive to transient GLUT1 knockdown than H522, in accordance with previous reports that loss of LKB1 sensitizes cells to metabolic disruption (*Momcilovic et al Cancer Res 2015*), however both ADC cell lines continued to proliferate even after depletion of GLUT1 (Supplementary Fig. 8b). We further employed an additional SqCC cell line, HCC2814 that contains an amplification of PIK3CA and lacks a Kras mutation, which are representative features of human SqCC. Consistently, GLUT1 knockdown in HCC2814 dramatically suppressed *in vitro* proliferation, which is associated with increased apoptosis and decreased intracellular ATP levels (Supplementary Fig 8d-h).

To determine whether GLUT1 knockdown inhibits tumor growth *in vivo* in SqCC, we implanted HCC95 expressing stable shGLUT1 or shGFP into nude mice. In accordance with *in vitro* proliferation, GLUT1 knockdown significantly inhibited the growth of HCC95 tumors (Fig 3g; Supplementary Fig. 7g). In

contrast, no significant difference in tumor growth was observed between shGLUT1 and shGFP cells in A549 or H522 tumors (Supplementary Fig. 7f, h). The selective cytotoxicity of GLUT1 knockdown in SqCC strongly suggests glycolytic addiction and a specific reliance on glucose metabolism. Notably, a reduction of glucose uptake in ADC cells reveals that while ADC may rely on GLUT1 as its primary glucose transporter, there exist mechanisms within ADC to maintain bioenergetics, cellular viability, and proliferative capacity outside of glucose metabolism. Overall, this data shows that GLUT1 expression and function is vital for maintaining energy homeostasis and proliferation in SqCC.”

✓3. *For all mouse experiments the individual growth curves of the mice need to be presented in the supplemental figures.*

We have included the individual growth curves for each mouse in the revised manuscript. The individual growth curves of each mouse are presented in Supplementary Figures 7g, h, 13a-f.

✓4. *Work using the GLUT1 inhibitors is underdeveloped and the specificity of these GLUT1 inhibitors is questionable. Mechanistically, how do these drugs inhibit GLUT1? Is glucose uptake reduced upon treatment with these drugs in SQCC (analogous to shGLUT1)? Is glycolysis inhibited with these drugs?*

We thank the reviewer 1 (and the reviewer 2, comment No.8) for pointing out issues with GLUT1 inhibitors. A new study by Ojelabi et al. demonstrated that WZB117 inhibited glucose uptake in GLUT1 specifically by binding to the exofacial sugar binding site in a competitive manner (Ojelabi et al. *JBC* 2016). The study further characterized the specificity of WZB117 and demonstrated that it binds to GLUT1 and GLUT3 with similar affinity. Our computational docking analysis also shows similar, but slightly higher affinity for GLUT1 than GLUT3 (Please see the reviewer 2, comment No. 8). STF31, identified and characterized by Chan et al. was shown to inhibit glucose uptake in VHL-/- RCC by binding to the central channel of GLUT1 by structural and docking studies (Chan et al *Sci Transl Med* 2011). Direct interactions between STF31 and GLUT1, but not GLUT2 or GLUT3 were detected using STF31 immobilized affinity columns. However, as described in detail below, we could not demonstrate the direct or indirect inhibition of GLUT1 via STF31 (Reviewer Only Figure 1).

We performed additional experiments to address the reviewer’s questions regarding the effects of WZB117 and STF31 on glucose uptake and glycolytic inhibition. WZB117 effectively inhibited glucose uptake in ADC cell line, A549 and SqCC cell lines HCC95, HCC1588, and HCC2814 by 77%, 64%, 74%, and 73% respectively (Figure 4l). To determine if WZB117 or STF31 could inhibit glycolysis, we measured the Extracellular Acidification Rate (ECAR) of ADC and SqCC cells after treatment with either GLUT1 inhibitors using an XFp flux analyzer (Seahorse). Upon treatment of WZB117, ECAR dropped in both ADC and SqCC cell lines cultured in 5mM glucose (Figure 4m), indicating that WZB117, through the inhibition of GLUT1 and glucose uptake, universally inhibits glycolysis in NSCLC cells. However, STF31 did not appear to inhibit glycolysis and in some cases resulted in a slight increase in ECAR in our flux analysis (Reviewer Only Figure 1a).

Collectively, this data shows that WZB117 reliably inhibits glucose uptake and glycolysis across both ADC and SqCC cell lines, though the most recent study as well as our computational docking analysis suggests that this may be through binding both the GLUT1 and GLUT3 glucose transporters (Ojelabi et al. *JBC* 2016). To address the possible GLUT3 inhibition by WZB117 and to respond the reviewer 2 (comment No.8 and No.14), we further investigated the functional contribution of GLUT3 in ADC cells by shRNA-mediated knockdown of GLUT3, but we found no effects of GLUT3 knockdown in cellular proliferation or viability in ADC cells (Supplementary Figure 11a, b). Despite this potential

affinity for GLUT3, our study proposes a unique reliance on glycolysis in SqCC, and the impact of WZB117 treatment, which is the inhibition of glucose uptake and subsequent glycolysis, exerts a selectively cytotoxic effect on SqCC cells.

Although STF-31 exerted SqCC-specific cytotoxic effects with decreased intracellular ATP levels (Reviewer Only Figure 1b, c), we were not able to demonstrate a GLUT1 inhibitory effect with STF-31 treatment. Therefore, we have decided to remove the data pertaining to STF-31 treatment from the manuscript. Instead, all experimental data presented in the original manuscript as well as newly generated data pertaining to STF-31 are now presented in the Reviewer Only Figure 1. The following changes have been made to the text to reflect this additional data:

Page 15, Line 7 – Page 16, Line 6

(Original)

“Treatment of cells with STF-31 or WZB117, selective, small-molecule GLUT1 inhibitors^{27, 28}, revealed that SqCC cells showed significantly higher susceptibility to GLUT1 inhibition than ADC cells (Fig. 4k,l; Supplementary Fig. 9b,c). Furthermore, upon GLUT1 inhibition, we observed apparent reduction of intracellular ATP in SqCC cells whereas ADC cells maintained their intracellular ATP (Supplementary Fig. 9d). These results suggest a crucial reliance on GLUT1 for survival and proliferation of SqCC as well as a glycolytic independence in ADC.”

(Revised)

“We chose WZB117, a selective, small-molecule GLUT1 inhibitor (*Liu et al Mol Cancer Ther 2012*) and measured glucose uptake and extracellular acidification rate (ECAR) in response to WZB117 treatment. Consistent with the previous study demonstrating inhibition of glucose uptake in A549, WZB117 inhibited glucose uptake in A549 by 77% (Fig. 4l). In SqCC cell lines HCC95, HCC1588, and HCC2814, we saw a reduction in glucose uptake of 64%, 74%, and 73% respectively (Fig. 4l). Decreased glucose uptake was associated with a substantial reduction in ECAR in both ADC and SqCC cell lines in a dose-dependent manner after WZB117 treatment (Fig. 4m). Furthermore, cell viability measurements after treatment with WZB117 revealed that SqCC cell lines showed significantly higher susceptibility to GLUT1 inhibition than ADC cell lines (Fig. 4n; Supplementary Fig. 10b). Considering that WZB117 inhibited glucose uptake and glycolysis in both ADC and SqCC, these results suggest a crucial reliance on GLUT1 for survival and proliferation of SqCC as well as a glycolytic independence in ADC. Though GLUT1 is the predominant glucose transporter expressed in both lung ADC and SqCC cell lines, a recent study as well as our computational docking analysis (AutoDock Vina, Scripps Research Institute) suggest that WZB117 may exert its effects by binding both GLUT1 and GLUT3 transporters (data not shown) (*Ojelabi et. al. JBC 2016*). Despite this additional affinity, GLUT3 is modestly and comparably expressed in the ADC and SqCC cells used in our study (Supplementary Figure 6b), and evaluating the functional role of GLUT3 in ADC using GLUT3 knockdown cells demonstrated no difference in the proliferation of shGLUT3 and shGFP control ADC cells (Supplementary Figure 11a,b). These results suggest a crucial reliance on GLUT1 for survival and proliferation of SqCC as well as a glycolytic independence in ADC.”

(Removed)

“We chose WZB117 whose specificity has been validated by glucose uptake inhibition in red blood cells, which express GLUT1 as a predominant glucose transporter²⁸. More importantly, STF-31 has been recently identified as a potent inhibitor of nicotinamide phosphoribosyltransferase (NAMPT), an enzyme essential for NAD⁺ salvage pathways, which presumably exert GLUT1-independent metabolic impact in cancer cells²⁹.”

Minor comments

✓1. *Additional information on the PDX model is needed. Are these subcutaneous tumors?*

The NSCLC PDX model used in this study was generated by inoculating small pieces of patient tumors into NOD/Scid mice and then successively passaging them as cancer tissue-originated spheroids (CTOS) by subcutaneous transplantation into the flanks in NOD/Scid mice. We have updated the methods section to remove any ambiguity as follows:

Page 36, Line 21 – Page 37, Line 2

(Original)

“Patient Derived Xenografts

...Fragments on the cell strainer and cells in the flow-through fractions were collected separately, and were each washed with HBSS and cultured in StemPro hESC medium (Gibco).”

(Revised)

“Patient Derived Xenografts

...Fragments on the cell strainer and cells in the flow-through fractions were collected separately, and were each washed with HBSS and cultured in StemPro hESC medium (Gibco). After processing, the tumor fragments spontaneously form cancer-tissue originated spheroids (CTOS). One hundred CTOS were then suspended in 50µL of matrigel (Corning Life Sciences) and subcutaneously transplanted into NOD/Scid mice.”

Response to Reviewer 2

✓1. Page 3, paragraph 1

To date, no successful achievement in the development of a targeted therapy for lung SqCC has been made. Since the approval of Necitumumab for treatment of squamous NSCLC this is not entirely correct and should be reworded.

We thank the reviewer 2 for pointing this out. We have edited the manuscript to reflect the recent FDA approval of Necitumumab as a first line therapy for metastatic lung SqCC, citing *Thatcher et al Lancet Oncology 2015* and *Zugazagoitia et al Translational Lung Cancer Research 2016*.

Page 3, Line 7-12

(original)

“...to date, no successful achievement in the development of a targeted therapy for lung SqCC has been made...”

(Revised)

“... to date, few achievements in the development of a targeted therapy for SqCC have been made, resulting in the use of platinum-based chemotherapy remaining the first-line treatment for decades⁴. The recent FDA approval of Necitumumab in combination with platinum based chemotherapy as a first line treatment for metastatic SqCC has generated positive, albeit limited clinical impact (*Thatcher et al Lancet Oncology 2015, Zugazagoitia et al Translational Lung Cancer Research 2016*).

✓2. Page 3, paragraph 2

“the differential usage of metabolic pathways in NSCLC subtypes has not been assessed”

While it is correct that detailed studies on this subject are lacking, the lower FDG uptake and glucose metabolic rates of lung adenocarcinomas as compared to squamous cell carcinomas are known since several years, see Brown et al. J Nucl Med 1999.

We agree with the reviewer 2 (and the reviewer 3, comment No.5) that clinical observations of higher FDG-PET activity and glucose metabolic rates in lung SqCC have been previously made. Although we have made references to previous studies in the original manuscript, we have rightfully included *Brown et al J Nucl Med 1999* and *Schuurbiens et al J Thorac 2014* in the revised manuscript and apologize for not including these references in the original submission. As the reviewers state, in these previous studies, no detailed, functional study has been performed to assess metabolic heterogeneity among NSCLC lung cancer phenotypes, which we present here. Accordingly, we have modified the manuscript as follows:

Page 3, Line 19-22

(original)

“the differential usage of metabolic pathways in NSCLC subtypes has not been assessed”

(Revised)

“In particular, the differential usage of metabolic pathways in NSCLC subtypes has not been addressed outside clinical observations, (*de Geus et al Lung Cancer 2007, Marom et al Lung Cancer 2001, Meijer et al Lung Cancer 2012, Brown et al J Nucl Med 1999, Schuurbiens et al J Thorac 2014, Choi Technol Health Care 2015*) and detailed functional studies have not been performed in representative pre-clinical models.”

(Removed from Discussion)

“This study is, to our knowledge, the first implication of differentially utilized and targetable metabolic pathways in NSCLC histological subtypes.”

3. Page 3/4

In this discussion the authors may want to add that there appear to be tissue specific differences in the regulation of glucose metabolism by the oncogenes discussed here. For instance, Hif-1 activation in renal cancer does not induce marked FDG uptake in this tumor type. Therefore, it is not obvious if the results obtained in other tumor types (e.g. colon cancer in reference 14) can also be applied to NSCLC.

We appreciate this insightful comment. The tissue-specific differences in metabolic regulation will certainly contribute to metabolic heterogeneity among different cancers, which harbor similar oncogenic drivers. In agreement with the Reviewer’s comments, we have added the following to the discussion:

Page 27, line 3-10

(Added)

“Although augmented HIF-1 α signaling has been well associated with elevated glycolytic flux and FDG uptake in human cancers, it is evident that tissue specific metabolic regulation may contribute to further heterogeneity in glucose metabolism. For example, VHL deletion and inactivation are among the most common mutations observed in renal cell carcinoma leading to oncogenic stabilization of HIF-1 α . Yet, renal cell carcinoma does not exhibit markedly elevated FDG uptake (*Okoro et al J Mol and Gen Medicine, 2014*) suggesting that tissue-specific differences may further regulate glucose metabolism among different cancers that harbor similar oncogenes.”

✓4. Page 7, paragraph 2

The meaning of the following statement is not clear “we analyzed GLUT1 expression in human SqCC (n=19-21)”. Were there 19 or 21 samples?

We apologize for the ambiguity and have modified the n-value reporting to insure interpretation of this analysis is clear. For this analysis, two of the included SqCC patients did not have available normal tissue for mRNA analysis. Therefore, the text should convey Normal n=19; Tumor n=21. We have clarified the text and relevant figures. We have changed the following within the text to address the reviewer’s comments.

Page 7, Line 7-9

(original)

“GLUT1 in human SqCC, we analyzed GLUT1 expression in human SqCC (n=19-12)...”

(Revised)

“we analyzed GLUT1 expression in human SqCC (Tumor, n=21; Normal, n=19) and ADC (Tumor, n=15; Normal, n=15) tumor tissue samples”

✓5. Page 11, last paragraph

“GLUT1-mediated glucose transport is considered a rate-limiting step for glycolysis in several cancers and benign tissues” The “rate limiting step” of glycolysis in cancer cells has been discussed for decades

with no generally accepted conclusion. Therefore, I would suggest to remove this sentence which does not add much to the presentation of the results. It could be changed to something like, "We next investigated if GLUT1 is the rate limiting step of glycolysis in squamous NSCLC"

We thank the reviewer 2 for the comment. In agreement that there is no definitive evidence that GLUT1 is the rate-limiting step of glycolysis, we have modified the questioned statement as follows:

Page 12, Line 18-19

(Original)

"GLUT1-mediated glucose transport is considered a rate-limiting step for glycolysis in several cancers and benign tissues..."

(Revised)

"GLUT1-mediated glucose transport is considered a key regulatory step for glycolysis in several cancers and benign tissues..."

✓6. *Page 12, last paragraph*

"... suggesting that SqCC is essentially reliant on glucose metabolism to support cellular bioenergetics". I don't quite understand this conclusion. Couldn't the change in glucose uptake for squamous cell carcinoma cells indicate that these cells have a very high capacity for glucose uptake whereas glucose uptake becomes saturated above physiologic concentrations in adenocarcinomas? This would be in line with the markedly higher expression levels of glucose transporters in squamous cell carcinomas.

We thank the reviewer 2 for these valid points (No. 6 and No. 7), and we agree that the change in lactate or oxygen consumption rate does not in itself indicate a reliance on glucose metabolism. We have revised the manuscript to include the possibility of glucose uptake saturation in ADC, which would account for a lack of observed metabolic responses in ADC when comparing between 25mM and 5mM glucose concentrations. To address this possibility experimentally, we performed a metabolic flux analysis utilizing Seahorse XFp analyzer and measured Extracellular Acidification Rate (ECAR) and Oxygen Consumption Rate (OCR) between ADC and SqCC cells when introduced to 5mM glucose (close to physiological serum glucose concentration in non-diabetic humans) after a short period of glucose starvation (0mM glucose).

The change in ECAR observed in SqCC cells was much more significant than in ADC cells upon addition of 5mM glucose (Figure 4j). Furthermore, the change in OCR diverged between ADC and SqCC cells, with ADC cells slightly increasing their OCR upon glucose addition, while SqCC cells markedly decreased their OCR (Figure 4j). Together, the differences in ECAR and OCR between ADC and SqCC cells within a physiologically normal range of glucose concentration suggests a much higher capacity for glucose internalization and catabolism in SqCC, as well as distinct propensities for the reduction or further catabolism of pyruvate in the TCA cycle. We amended the manuscript to include the reviewer's consideration as follows:

Page 13, Line 19 – Page 14, Line 19

(Original)

"....Suggesting that SqCC is essentially reliant on glucose metabolism to support cellular bioenergetics....."

(Revised)

“We speculated that this reduction in lactate production and oxygen consumption was associated with a disruption of cellular bioenergetics. Indeed, SqCC cell lines grown in media containing 5mM glucose exhibit significantly reduced intracellular ATP, NADH, and NADPH levels compared to cells grown in 25mM glucose (Fig. 4f-h). However, ADC cell lines exhibited no change in ATP, NADH, or NADPH levels between cells grown in 5mM glucose or 25mM glucose. It is possible that the change in SqCC bioenergetics represents a higher capacity for glucose uptake and utilization, whereas in 25mM glucose, glucose uptake in ADC cell lines becomes saturated. To validate if this fluctuation in bioenergetics represents a substantial reliance on glucose, but not an artifact of non-physiologically elevated glucose uptake due to high GLUT1 level in SqCC cells, we cultured SqCC and ADC cell lines in decreasing concentrations of glucose. At 5mM glucose concentration, reduced intracellular ATP, NADH, and NADPH (Fig. 4f-h) is associated with a significant decrease in cellular viability compared to ADC cell lines (Fig. 4i) indicating that SqCC cell lines are significantly more sensitive to glucose concentrations than lung ADC cell lines. Lower than physiological glucose concentrations further decreased the cellular viability of SqCC cell lines. Furthermore, metabolic flux analyses in ADC and SqCC cell lines cultured in 5mM glucose after glucose starvation (0mM) revealed that cellular glycolytic flux (extracellular acidification rate, ECAR) is significantly more elevated in SqCC cell lines compared to ADC cell lines (Fig. 4j). Oxygen consumption rate (OCR) was markedly decreased in SqCC compared with a divergent increase in ADC (Fig 4j). In accordance with SqCC-specific susceptibility to GLUT1 knockdown, these distinct metabolic responses strongly suggest a unique reliance on glucose by SqCC but also provide a rationale for decreased cell viability upon glucose deprivation.”

✓7. Page 13, first paragraph

Is 5 mmol/l is the physiologic plasma glucose concentration in human and 25 mmol/l is severely hyperglycemic. Is it appropriate to call a concentration of 5 mmol/l glucose deprivation or would it be more appropriate that supra-physiologic glucose concentrations stimulate growth of squamous cell carcinoma cell lines?

We agree with the reviewer 2 that because 5mM glucose approximates physiological blood glucose concentrations, referring to 5mM glucose as “glucose deprivation” is a misnomer, so we revised the results to remove this term and any ambiguity that it may create. Instead, we only refer to < 5mM glucose concentrations as lower than physiological when discussing the significant reduction in cell viability in SqCC cells. As with the previous comment (No.6), we have revised the manuscript to include the possibility of glucose uptake saturation in ADC.

✓8. Page 14-15

Do 2-DG, WZB117, and STF-31 block other glucose transporters and specifically glut-3 with the same affinity as Glut-1?

We performed computational ligand/protein docking simulation analysis with these inhibitors and available outward-open protein structures of human GLUT1 and GLUT3. Briefly, the binding affinity between inhibitors and hGLUT1 and hGLUT3 proteins were computationally calculated by AutoDock Vina package (Trott et al J of Computational Chemistry 2010). The human GLUT3 outward-open crystal structure was obtained from PDB. We utilize online I-TASSER server (Zhang BMC Bioinformatics 2008) to predict the outward-open structure for hGLUT1. For each inhibitor, we used Vina Package to dock the inhibitor in the space close to reported glucose interacting sites of GLUT1 and GLUT3 (Deng et al Nature 2015). 10 replicates of docking were performed for each inhibitor and mean binding affinity

was calculated. As presented in the table below, all three inhibitors have slightly higher (theoretical) affinities for hGLUT1 than hGLUT3.

Protein-ligand binding affinity (Kcal/mol)		
	hGLUT1	hGLUT3
2-DG	-5.6	-5.2
STF-31	-7.5	-6.8
WZB117	-10.0	-9.0

WZB117, which shows highest affinity for both GLUT1 and GLUT3 in our docking analysis, was demonstrated by Ojelabi et al. to inhibit glucose uptake in GLUT1 and GLUT3 with similar affinity (Ojelabi et al. *JBC* 2016). Though GLUT1 is the predominant glucose transporter expressed in both lung ADC and SqCC cells, this study suggests that WZB117 may exert its effects by binding both GLUT1 and GLUT3 transporters. Despite this additional affinity, our study proposes a unique reliance on glycolysis in SqCC, and the impact of WZB117 treatment is much like 2-DG treatment, which is the inhibition of glycolysis that exerts a selectively cytotoxic effect on SqCC cells (Figure 4n; Supplementary Figure 10b). We further explored the expression and functional role of GLUT3 in ADC cells under the reviewer 2, No. 14 comment. We generated GLUT3 knockdown ADC cells and measured in vitro proliferation. No difference in proliferation was noticed between shGLUT3 knockdown and shGFP control ADC cells (Supplementary Figure 11).

STF-31, identified and characterized by Chan et al., was shown to selectively target VHL-/- renal cell carcinoma through GLUT1 inhibition (Chan et al. *Sic Tranl Med* 2011). Direct interactions between STF-31 and GLUT1, but not GLUT2 or GLUT3 were detected using STF-31-immobilized affinity columns. However, despite the fact that STF-31 exerted specific cytotoxic effects on SqCC cell lines associated with decrease ATP levels (presented in Figure 4k and Supplementary Figure 9 in the original manuscript now moved to Reviewer Only Figure 1b, c), we could not demonstrate the direct or indirect inhibition of GLUT1 via STF-31 through metabolic flux assays (Reviewer Only Figure 1a) and have decided to remove this data from the manuscript (Reviewer Only Figure 1). The following updates have been made to the text to reflect these changes:

Page 15, Line 7 – Page 16, Line 6

(Original)

“Treatment of cells with STF-31 or WZB117, selective, small-molecule GLUT1 inhibitors^{27, 28}, revealed that SqCC cells showed significantly higher susceptibility to GLUT1 inhibition than ADC cells (Fig. 4k,l; Supplementary Fig. 9b,c). Furthermore, upon GLUT1 inhibition, we observed apparent reduction of intracellular ATP in SqCC cells whereas ADC cells maintained their intracellular ATP (Supplementary Fig. 9d). These results suggest a crucial reliance on GLUT1 for survival and proliferation of SqCC as well as a glycolytic independence in ADC.”

(Revised)

“We chose WZB117, a selective, small-molecule GLUT1 inhibitor (Liu et al *Mol Cancer Ther* 2012) and measured glucose uptake and extracellular acidification rate (ECAR) in response to WZB117 treatment. Consistent with the previous study demonstrating inhibition of glucose uptake in A549, WZB117 inhibited glucose uptake in A549 by 77% (Fig. 4l). In SqCC cell lines HCC95, HCC1588, and HCC2814, we saw a reduction in glucose uptake of 64%, 74%, and 73% respectively (Fig. 4l). Decreased glucose uptake was associated with a substantial reduction in ECAR in both ADC and SqCC cell lines in a dose-dependent manner after WZB117 treatment (Fig. 4m). Furthermore, cell viability measurements after

treatment with WZB117 revealed that SqCC cell lines showed significantly higher susceptibility to GLUT1 inhibition than ADC cell lines (Fig. 4n; Supplementary Fig. 10b). Considering that WZB117 inhibited glucose uptake and glycolysis in both ADC and SqCC, these results suggest a crucial reliance on GLUT1 for survival and proliferation of SqCC as well as a glycolytic independence in ADC. Though GLUT1 is the predominant glucose transporter expressed in both lung ADC and SqCC cell lines, a recent study as well as our computational docking analysis (AutoDock Vina, Scripps Research Institute) suggest that WZB117 may exert its effects by binding both GLUT1 and GLUT3 transporters (data not shown) (Ojelabi *et. al. JBC 2016*). Despite this additional affinity, GLUT3 is modestly and comparably expressed in the ADC and SqCC cell lines used in our study (Supplementary Figure 6b), and evaluating the functional role of GLUT3 in ADC using GLUT3 knockdown cells demonstrated no difference in the proliferation of shGLUT3 and shGFP control ADC cell lines (Supplementary Figure 11a,b). These results suggest a crucial reliance on GLUT1 for survival and proliferation of SqCC as well as a glycolytic independence in ADC.”

(Removed from Result)

“We chose WZB117 whose specificity has been validated by glucose uptake inhibition in red blood cells, which express GLUT1 as a predominant glucose transporter²⁸. More importantly, STF-31 has been recently identified as a potent inhibitor of nicotinamide phosphoribosyltransferase (NAMTP), an enzyme essential for NAD⁺ salvage pathways, which presumably exert GLUT1-independent metabolic impact in cancer cells²⁹.”

✓9. Page 15, paragraph 1

Glut-1 is the major glucose transporter at the blood brain barrier. Did treatment with Glut-1 inhibitors cause cerebral side effects?

As the reviewer 2 pointed out, we have addressed potential neurological side effects in the manuscript discussion (Page 29-30) citing previous studies demonstrating no obvious cerebral side effects with the use of GLUT1 inhibitors (*Chan et al Sci Tranl Med 2011*). Yet, to better address the reviewer’s concerns, we administered WZB117 (10 mg/kg) or vehicle to mice daily for three weeks and grossly and histologically examined multiple tissues including brain to ascertain any toxicity associated with daily treatment. We noticed no apparent histological differences between vehicle-treated and WZB117 treated mice when examining major organs including brain, liver, lung, spleen, heart, and kidney (Supplementary Figure 21b).

As GLUT1 is the principle glucose transporter in erythrocytes, we also performed a blood panel to measure hematocrit, RBC content, WBC content, platelet count, as well as hemoglobin. Liu et al. reported a difference in lymphocyte and platelet count that was within normal cell count ranges (*Liu et al Mol Cancer Ther 2012*). Consistently, we noticed a modest decrease in WBC count, but no difference in platelets. Importantly, vehicle and WZB117 treated mice exhibited no significant difference in hematocrit, RBC count, and hemoglobin content. Overall, the blood panel revealed that all hematological cell counts were within physiologically normal ranges (Supplementary Figure 21a).

We have made the following changes to include this additional data and to address the specificity and potential side effects of GLUT1 inhibitors:

Page 29, Line 17 – Page 30, Line 1

(Original)

“Our results demonstrating specific anti-cancer effects of GLUT1 and glycolysis inhibitors on SqCC suggest enough therapeutic window for GLUT1 inhibition. Although there is potential risk of side effects in employing GLUT1 inhibitors, given that GLUT1 is the primary GLUT family member expressed in red blood cells and endothelial cells of the blood-brain barrier⁸, no apparent toxicities have been observed in the brain and other normal tissues with administration regimens used in this study, yet significant anti-SqCC effects were achieved (Fig. 5f,g).”

(Revised)

Our results demonstrating specific anti-cancer effects of GLUT1 and glycolysis inhibitors on SqCC suggest enough therapeutic window for glycolytic inhibition. Although there is potential risk of side effects in employing GLUT1 inhibitors, given that GLUT1 is the primary GLUT family member expressed in red blood cells and endothelial cells of the blood-brain barrier⁸, hematologic analysis and histological evaluation of major organs including brain revealed no apparent toxicities with the administration regimens used in this study, yet significant anti-SqCC effects were achieved (Supplementary Fig. 21; Fig. 5f, g).

✓10. Page 16, paragraph 1

“... and further suggest that 18 F-FDG-PET imaging can aid in designing more feasible diagnostic strategies in differentiating SqCC from other types of lung cancer”. This has been studied rather extensively clinically, but the overlap in FDG uptake between adeno and squamous cell carcinomas appears too large. The authors should therefore consider to delete this statement Page 24 “refined noninvasive diagnostic imaging techniques which specifically exploit elevated glucose influx will substantially improve the clinicopathological identification of SqCC from other types of lung cancer.” Considering the overlap in FDG uptake of squamous and adenocarcinomas this appears doubtful.

We thank the reviewer 2 (and the reviewer 3, comment No.6) for the insightful comment. We agree with the reviewers that the SUVmax overlap is possibly too large to offer clinical utility in distinguishing ADC from SqCC. Although significantly higher in lung SqCC, FDG uptake may not be enough to differentiate between ADC and SqCC in clinical diagnosis. However, the observation of higher FDG uptake in lung SqCC highlights a core metabolic difference between lung SqCC and ADC. We have revised the interpretations of these experiments to support the observation of the elevated GLUT1 mediated glucose uptake in lung SqCC which we have expanded on in multiple functional, pre-clinical, and clinical models. We have also proposed that future studies will be performed to determine if FDG-PET imaging can be used to identify tumors most susceptible to GLUT1 or glycolytic inhibition. In this way, we propose that FDG-PET imaging may contribute to non-invasive molecular subtyping of SqCC and ADC tumors based on metabolic phenotypes and may contribute to the development of a metabolic marker for treatment outcome prediction. In response to the comments from the reviewer 2 as well as the reviewer 3, we have modified and removed the following sentences from the manuscript:

Page 17, Line 19-23

(Original)

“Differential expression of GLUT1 and glucose uptake between SqCC and ADC suggests potential indication of ¹⁸F-FDG-PET-based imaging for the differential diagnosis of SqCC. We took advantage of the KL mouse model, which displays mixed ADC and SqCC lung tumor heterogeneity (Supplementary Fig. 5a,b), by performing ¹⁸F-FDG-PET/CT imaging of KL mice followed by pathological evaluation and immunohistochemical staining for GLUT1 (Fig. 6a).”

(Revised)

“Previous clinical studies have identified increased ^{18}F -FDG uptake in squamous subtypes (*de Geus et al Lung Cancer 2007, Marom et al Lung Cancer 2001, Meijer et al Lung Cancer 2012, Brown et al J Nucl Med 1999, Schuurbiens et al J Thorac 2014, Choi Technol Health Care 2015*). We took advantage of the KL mouse model, which displays both ADC and SqCC lung tumor heterogeneity (Supplementary Fig. 5a,b), by performing ^{18}F -FDG-PET/CT imaging of KL mice followed by pathological evaluation and immunohistochemical staining for GLUT1 (Fig. 6a).”

(Removed from Results)

“Differential expression of GLUT1 and glucose uptake between SqCC and ADC suggests potential indication of ^{18}F -FDG-PET-based imaging for the differential diagnosis of SqCC.”

(Removed from Results)

“...further suggest that ^{18}F -FDG-PET imaging can aid in designing more feasible diagnostic strategies in differentiating SqCC from other types of lung cancer.”

(Removed from Discussion)

“Given that differential diagnosis of NSCLC is essentially required for determining a patient’s candidacy for molecularly targeted therapy, developing more refined noninvasive diagnostic imaging techniques which specifically exploit elevated glucose influx will substantially improve the clinicopathological identification of SqCC from other types of lung cancer.”

✓11. Page 20

“Conversely, GLUT1 expression and glycolytic flux remain relatively minimal in ADC, indicative of glucose independency” This should be formulated more carefully given the various subtypes of lung adenocarcinomas. Some of the subtypes demonstrate equally high FDG uptake on PET/CT scans as squamous cell carcinomas.

We thank the reviewer 2 for commenting on this and completely agree with the comment. Recent studies correlating GLUT1 expression and SUVmax with ADC subtypes (*Nakamura et al Lung Cancer 2015 and Maki et al Oncology Reports 2013*) have demonstrated significant heterogeneity within lung ADC according to the histopathological classification criteria. We agree that some ADC tumors, especially those exhibiting the solid subtype of ADC, do exhibit higher SUVmax which can be comparable to SqCC (*Chiu et al Journal of Thoracic Oncology 2011*). To address this comment, we compared FDG uptake, via SUVmax, between SqCC tumors and the various histopathological subtypes of ADC tumors (Ancinar, Papillary, Micropapillary, Lipidic, and Solid), using the same NSCLC patient cohort presented in the original submission (Supplementary Table 1). Although, some patients belonging to ADC Ancinar and Papillary subtypes have similar SUVmax as SqCC tumors, SqCC tumors exhibit higher median SUVmax than any of the ADC subtypes represented in the cohort (Reviewer Only Figure 2). We also analyzed GLUT1 expression using semi-quantitative IHC and qRT-PCR analysis, comparing SqCC tumors to ADC subtypes. Similarly, we observed that GLUT1 expression is much higher in SqCC than any of the ADC subtypes represented in the cohort (Reviewer Only Figure 2). Due to the limited number of patients belonging to each ADC subtype, a statistical analysis is not possible. It should be also stated that the cohort used for this analysis lacked both Solid and Micropapillary subtypes. Therefore, we believe it would be inappropriate to present these analyses in the manuscript, and have instead presented this analysis in Reviewer Only Figure 2. Based on previous studies (*Chiu*

et.al. Journal of Thoracic Oncology 2011), we would expect the Solid subtype of ADC to have higher SUVmax, perhaps more comparable to SqCC. Further study is required to address this. In agreement with the Reviewer's comments, we have made the following modification to the text:

Page 23, Line 14–16

(Original)

“Conversely, GLUT1 expression and glycolytic flux remain relatively minimal in ADC, indicative of glucose independency”

(Revised)

“Conversely, GLUT1 and glycolytic enzyme expression remain relatively low in lung ADC when compared to SqCC, suggesting that lung ADC may be significantly less reliant on glucose metabolism.”

Page 27, Line 5-11

(Added)

“While increased ¹⁸F-FDG uptake in SqCC is clearly evident, recent studies have demonstrated that certain subtypes of ADC, such as the solid subtype, demonstrate high ¹⁸F-FDG-PET activity as well (*Nakamura et al Lung Cancer 2015, Maki et al Oncology Reports 2013, Chiu et.al. Journal of Thoracic Oncology 2011*). Our analysis of ¹⁸F-FDG uptake in NSCLC patients suggests that, while a small proportion of Ancinar and Papillary ADC tumors have comparably similar SUVmax, SqCC tumors exhibit significantly increased SUVmax (Supplementary Table 1).”

✓12. Page 21

The meaning of the following sentence is not entirely clear: “ADC may rewire its metabolism to acquire glucose independence in response to glucose deprivation or glycolytic inhibition”

What is meant by “rewiring”?

The meaning of “rewire its metabolism” is referring to the possibility that lung ADC may adapt to glucose deprivation by altering the metabolic pathways it uses to survive. To clarify our discussion, we have removed this statement from the manuscript.

13. Page 21-22

I would suggest to shorten the rather speculative discussion on glucose metabolism of adenocarcinomas

We apologize for the speculative discussion about glucose metabolism in ADC. We have revised the discussion, as the following examples indicate, to be less speculative about ADC metabolism.

Page 23, Line 14-16

(Original)

“Conversely, GLUT1 expression and glycolytic flux remain relatively minimal in ADC, indicative of glucose independency”

(Revised)

“Conversely, GLUT1 and glycolytic enzyme expression remain relatively low in ADC when compared to SqCC, suggesting that ADC may be significantly less reliant on glucose metabolism.”

(Removed from Discussion)

“We speculate that ADC may utilize metabolic pathways independent of glucose availability, such as glutamine metabolism, to sustain its energetic demands.”

(Removed from Discussion)

“Alternatively, ADC may rewire its metabolism to acquire glucose independence in response to glucose deprivation or glycolytic inhibition.”

(Removed from Discussion)

“Although our IHC analysis of ADC PDX samples revealed negligible GLUT1 protein expression in EGFR or KRAS mutants (Fig. 1g; Supplementary Table 2), we cannot exclude the possibility that EGFR or KRAS mutations may increase GLUT1 membrane localization via PI3K/AKT/mTOR signaling pathways in ADC.”

✓14. Figure 2a

It would be helpful to add information on expression of Glut-3 in the studied cell lines, because Glut-3 is believed to be a major glucose transporter in adenocarcinomas as also indicated by figure 1b.

We thank the reviewer for this comment. We attempted to evaluate the protein expression of GLUT3 in the studied cell lines, however we could not detect a reliable band for GLUT3 after attempting several different antibodies, possibly due to the very low expression level in the aforementioned cell lines (Supplementary Figure 11a). However, our qPCR analysis suggests that GLUT3 is minimally expressed among the ADC cell lines in the present study when compared to the mRNA expression of all the GLUT isoforms and the sodium glucose transporters (Supplementary Figure 6b). To evaluate the functional role of GLUT3 in ADC, we generated GLUT3 knockdown ADC cells and measured *in vitro* proliferation (Supplementary Figure 11a). No difference in proliferation was noticed between shGLUT3 knockdown and shGFP control ADC cells (Supplementary Figure 11b). In reflection of the Reviewer 2 (Comment No. 8 and No. 14) as well as Reviewer 1 (Comment No. 4) we have added this data in the results section concerning the use of WZB117.

Page 15, Line 7 – Page 16, Line 6

(Original)

“Treatment of cells with STF-31 or WZB117, selective, small-molecule GLUT1 inhibitors^{27, 28}, revealed that SqCC cells showed significantly higher susceptibility to GLUT1 inhibition than ADC cells (Fig. 4k,l; Supplementary Fig. 9b,c). Furthermore, upon GLUT1 inhibition, we observed apparent reduction of intracellular ATP in SqCC cells whereas ADC cells maintained their intracellular ATP (Supplementary Fig. 9d). These results suggest a crucial reliance on GLUT1 for survival and proliferation of SqCC as well as a glycolytic independence in ADC.”

(Revised)

“We chose WZB117, a selective, small-molecule GLUT1 inhibitor (*Liu et al Mol Cancer Ther 2012*) and measured glucose uptake and extracellular acidification rate (ECAR) in response to WZB117 treatment. Consistent with the previous study demonstrating inhibition of glucose uptake in A549, WZB117 inhibited glucose uptake in A549 by 77% (Fig. 4l). In SqCC cell lines HCC95, HCC1588, and HCC2814, we saw a reduction in glucose uptake of 64%, 74%, and 73% respectively (Fig. 4l). Decreased glucose uptake was associated with a substantial reduction in ECAR in both ADC and SqCC cell lines in a dose-dependent manner after WZB117 treatment (Fig. 4m). Furthermore, cell viability measurements after treatment with WZB117 revealed that SqCC cell lines showed significantly higher susceptibility to

GLUT1 inhibition than ADC cell lines (Fig. 4n; Supplementary Fig. 10b). Considering that WZB117 inhibited glucose uptake and glycolysis in both ADC and SqCC, these results suggest a crucial reliance on GLUT1 for survival and proliferation of SqCC as well as a glycolytic independence in ADC. Though GLUT1 is the predominant glucose transporter expressed in both lung ADC and SqCC cell lines, a recent study as well as our computational docking analysis (AutoDock Vina, Scripps Research Institute) suggest that WZB117 may exert its effects by binding both GLUT1 and GLUT3 transporters (data not shown) (*Ojelabi et. al. JBC 2016*). Despite this additional affinity, GLUT3 is modestly and comparably expressed in the ADC and SqCC cell lines used in our study (Supplementary Figure 6b), and evaluating the functional role of GLUT3 in ADC using GLUT3 knockdown cells demonstrated no difference in the proliferation of shGLUT3 and shGFP control ADC cell lines (Supplementary Figure 11a,b). These results suggest a crucial reliance on GLUT1 for survival and proliferation of SqCC as well as a glycolytic independence in ADC.”

✓15. *Supplemental Information*

Please add brief descriptions on the methods used for small and animal and clinical FDG PET/CT imaging.

We apologize for omitting the detailed methodology on FDG PET/CT. We now included detailed descriptions on FDG PET/CT in methods section.

Response to Reviewer 3

✓1. -page 16: the authors state “We confirmed the genomic amplification of PIK3CA (Supplementary Fig. 11a) and correlatively increased PIK3CA mRNA expression (data not shown) in the vast majority of the TCGA lung SqCC cohort”. On page 24 it is stated “In agreement with these studies, our TCGA analyses found copy number gain of PIK3CA with amplification of more than 2 copies in 47% of the SqCC samples. Heterozygous loss of PTEN was observed in over 50% of the SqCC samples. (Fig. 7b; Supplementary Fig.11a-d).

This statement does not seem to accurately reflect the TCGA results: in the original TCGA paper (Nature, 2012), PTEN and PIK3CA were altered in 15 and 16%, respectively. Furthermore, in Figure 7B, it seems that more than 80% have PIK3CA amplification based on the number of dots present (this may be an underestimate- I couldn't count them all). The authors should clarify why their analysis is markedly different than the TCGA, as well as the discrepancy between figure 7B and the 47 or 50% numbers stated in the text. (Note that >2 copies is not typically considered amplified; a higher cutoff is usually used).

We appreciate the reviewer 3 for pointing this out. The copy number alteration component to the original TCGA paper analysis was limited to high-level amplifications of PIK3CA and homozygous deletions of PTEN. In our analysis, we expanded to include genomic regions, which exhibit both shallow and high-level alterations, resulting in higher frequency of occurrence (15-16% vs 47-50%). Our rationale for including both shallow and high-level copy number alterations is based on our observation that both PTEN and PIK3CA exhibit a robust, relatively linear correlation between copy number and mRNA expression in the TCGA lung SqCC cohort (Supplementary Fig 14c,d). Consistent with our findings, a recent study has identified the recurrent genomic amplification of chromosome 3q26-28, the genomic region containing PIK3CA, in 94% (54% amplification, 40% shallow gain) of TCGA primary lung SqCC tumors ($n=177$) (Kim et al PLOS Biology 2016). We have further confirmed the frequent genomic amplification of SOX2, as well as TP63, which is also located in the lung SqCC 3q26-28 amplification, in 90.56% (48.19% amplification, 42.67% shallow gain) and 88.15% (40.96% amplification, 47.19% shallow gain) of the TCGA lung SqCC cohort with available SNP6 copy number data ($n=498$), respectively (Reviewer Only Figure 4). The reviewer's comments have highlighted the need to modify the nomenclature we have used to denote copy number alterations. Accordingly, we have modified the calling of copy number alterations to match the GISTIC 2.0 default output for discrete copy number calls (Figure 7a,b) (Mermel et al Genome Biology 2011). We have also added linear copy number correlation with gene expression to highlight the effect of copy number alterations on the expression of PIK3CA and PTEN (Supplementary figure 14c, d). To further validate the significance of PIK3CA genomic gain in lung SqCC, we queried the whole TCGA mRNA-Seq data and found that Lung SqCC has the highest expression of PIK3CA (Supplementary Figure 15). Accordingly, we have edited the relevant figures and manuscript to reflect our analysis more clearly:

Page 18, Line 19 – Page 19, Line 14
(Added/Revised)

A previous TCGA analysis has identified significant enrichment of PI3K pathway activating alterations, which included somatic mutations (PIK3CA, PTEN, and mTOR), high-level focal amplifications (PIK3CA), homozygous deletions (PTEN), and significant mRNA expression alterations (PIK3CA, PTEN, and mTOR) in 47% of the TCGA cohort of lung SqCC (Cancer Genome Atlas Research N. Nature 2012). We confirmed the frequent amplification of the genomic region containing PIK3CA within lung SqCC (Supplementary Fig. 14a) (Broad Institute TCGA Genome Data Analysis Center 2016, SNP6 Copy number analysis, GISTIC2, Broad Institute of MIT and Harvard) (Mermel et al Genome Biology 2011).

Our TCGA analyses revealed both shallow genomic copy number gains and high-level amplifications of PIK3CA in the majority of patients of the TCGA lung SqCC cohort (46.4% amplification, 44% shallow gain) and a robust linear correlation between PIK3CA genomic copy number and mRNA expression (Fig. 7a; Supplementary Fig. 14c). Notably, GLUT1 mRNA expression positively correlated with PIK3CA genomic copy number (Fig. 7a). Our analysis also revealed that lung SqCC exhibits the highest mRNA expression of PIK3CA among the TCGA cohorts, which may lead to heightened PI3K/AKT pathway activation in lung SqCC (Supplementary Fig. 15). Our analysis of the TCGA lung SqCC cohort further revealed frequent shallow deletions in the genomic region containing PTEN within the TCGA cohort of lung SqCC (9.8% deep deletion, 44.8% shallow deletion) (Supplementary Fig 14b). Similar to PIK3CA, PTEN mRNA expression robustly correlated with genomic copy number (Supplementary Fig 14d).

Page 28, Line 4-12

(Original)

“In agreement with these studies, our TCGA analyses found copy number gain of PIK3CA with amplification of more than 2 copies in 47% of the SqCC samples. Heterozygous loss of PTEN was observed in over 50% of the SqCC samples”

(Revised)

“In agreement with these studies, our TCGA analyses confirmed genomic copy number gain, including both shallow and high level amplifications, of PIK3CA frequently occur in the TCGA lung SqCC cohort (Fig. 7a; Supplementary Fig. 14a) (Broad Institute TCGA Genome Analysis Data Center, 2016, SNP6 Copy number analysis, GISTIC2, Broad Institute of MIT and Harvard), leading to lung SqCC exhibiting exceptionally high mRNA expression of PIK3CA among the TCGA cohorts (Supplementary Fig. 15). As well, a frequent genomic copy number loss of PTEN is evident within the TCGA cohort of SqCC samples (Fig. 7b; Supplementary Fig. 14b).”

✓2. *The suggestion that GLUT1 is upregulated in LUSC, but not in lung ADC (“We show that GLUT1 is remarkably and uniquely elevated at both the mRNA and protein levels in lung SqCC as the principal cellular glucose transporter, but is minimally expressed in lung ADC”, page 4) does not appear to take into account subgroups of lung ADC in which these pathways are known to be upregulated. Indeed, the authors note that KRas and BRAF mutations, which occur in lung ADC but not SCC, upregulate GLUT-1 expression (refs 13 and 14).*

Furthermore, prior studies have established that LKB1 loss is associated with HIF1-pathway upregulation (e.g. Ji et al, Nature 2006), a finding which was extended in a study (led by a coauthor of the current study, DBS) showing that LKB1 loss is associated with HIF1a upregulation, GLUT1 upregulation, and increased FDG uptake (Shackelford et al, PNAS 2009). LKB1 loss occurs predominantly in lung adenocarcinoma.

An analysis should be done comparing levels of GLUT1 and other relevant targets here in lung SCC vs specific subgroups of lung ADC, including, BRAF mutants (although the numbers in the TCGA are modest), LKB1 mutants, PIK3CA/PTEN mutants (or PI3K/AKT activated at the protein level, given the proposed role for this pathway in regulating HIF1a). This can be done using the TCGA as well as other publicly available datasets. Furthermore, in supplemental figure 14, it should be clarified as to whether only Kras G12D was analyzed as indicated; if so, this analysis should be changed to include all KRas mutants, given prior data linking Kras with Glut1.

We appreciate the reviewer 3 for this insightful comment. As the reviewer 3 pointed out, we discussed the previously reported studies suggesting that Kras and Braf mutations are associated with upregulated GLUT1 expression. Our TCGA analysis, however, could not reveal the association between Kras or EGRF mutations and GLUT1 expression in ADC (Supplementary Figure 18a). We have further stratified the TCGA cohort of lung ADC into patients exhibiting common mutations such as: LKB1, BRAF, PIK3CA, and PTEN mutants and found no significant difference in GLUT1 mRNA expression between the groups; although the available numbers are modest for some of the rarer mutations (e.g. BRAF, PIK3CA, and PTEN). As in SqCC, we observed a significant trend of increased GLUT1 expression with increased PIK3CA putative copy number in ADC, although the proportion of patients with PIK3CA copy number gains in ADC are significantly less than what we observe in SqCC (Supplementary Figure 18b).

As the reviewer 3 pointed out, LKB1 loss and inactivating mutations have been shown to induce HIF-1 α upregulation and signaling in various tissues. However, we did not observe GLUT1 upregulation in LKB1 mutant lung ADC tumors compared to SqCC tumors in the TCGA (GLUT1 expression in LKB1 mutant ADC appears to be marginally less than LKB1 wild type ADC tumor) (Supplementary Figure 18a). Moreover, the previous studies linking unregulated HIF-1 α signaling to LKB1 inactivation did not compare GLUT1 expression or FDG uptake in different subtypes of NSCLC (e.g. ADC vs SqCC). Indeed, LKB1-null A549 cells exhibit higher GLUT1 expression as compared to other ADC cell lines we tested, yet all SqCC cell lines except HCC2450 express considerably higher levels of GLUT1 than A549 cells in vitro (Figure 2a,b), and SqCC xenograft tumors (HCC95 and HCC1588) exhibit significantly higher non-hypoxic HIF-1 α induction than A549 xenograft tumors (Supplementary Figure 16c). Although additional studies are necessary for better elucidation of the functional contributions of LKB1 loss in HIF-1 α /GLUT1, our observations suggest that SqCC has markedly elevated HIF-1 α and GLUT1 expression, even higher than LKB1-null ADC tumors.

We also analyzed differential FDG uptake between EGFR wild type and EGFR mutant samples in the clinical cohort reported in the present work and found no significant difference in SUVmax (Reviewer Only Figure 3). Although, given the limited sample size of assayed patients in our clinical cohort (EGFR mt n=6, EGFR wt n=3), we are not able at this time to address differential FDG uptake between the various ADC subtypes.

We would like to apologize for the error in labeling Kras mutants in Supplementary Figure 18a. The figure should read as all oncogenic Kras mutants. This criterion excludes 3 lung ADC patients with Kras mutation of unknown effect. Of the included patients, all but one possess KRAS G12 mutations. In agreement with the reviewer's comments, we have added the following to the text:

Page 24, Line 6 – Page 25, Line 1
(Added/Revised)

“Although previous studies have reported an increase in GLUT1-mediated aerobic glycolysis by EGFR, KRAS, BRAF or other frequent oncogenic mutations in ADC (Makinoshima et al JBC 2014, Kerr et al Nature 2016, Ying et al Cell 2012, Yun et al Science 2009, Yun et al Science 2015), our analysis found no significant increase in GLUT1 mRNA expression in ADC patients possessing KRAS, EGFR, BRAF, LKB1, PIK3CA, or PTEN mutations compared to those that did not (Supplementary Fig. 18a). LKB1 inactivation has been linked to increased HIF-1 α signaling (Faubert et al PNAS 2014, Shackelford et al PNAS 2009), yet, a slight, but statistically significantly lower expression of GLUT1 was detected in tumors with LKB1 mutations compared to LKB1 wild type tumors (Supplementary Fig. 18a). Moreover, these studies did not compare GLUT1 expression or FDG uptake among the different subtypes of NSCLC. We demonstrate that LKB1-null A549 exhibits higher GLUT1 expression as compared to other ADC cell lines we tested, yet all SqCC cell lines except HCC2450 express considerably higher levels of GLUT1 than

A549 cells (Figure 2a,b), and SqCC xenograft tumors (HCC95 and HCC1588) exhibit significantly higher HIF-1 α induction than A549 xenograft tumors (Supplementary Fig. 16c). Importantly, as in SqCC, GLUT1 mRNA expression was also increased in ADC patients possessing genomic copy number gains of PIK3CA, although occurring at a much lower frequency than in the TCGA lung SqCC cohort (Supplementary Fig.18b). This observation suggests a mechanistic involvement of aberrantly activated PIK3 pathway signaling in elevated glucose metabolism in NSCLC.”

✓3. The SCCs observed in the KL mice does not seem to be representative of phenotypes observed in human LKB1-mutant lung cancer, which are typically lung ADC without any mixed component. It is thought this squamous differentiation may be due, at least in part, to redox imbalance (Li et al, Cancer Cell 2015) which may differ in human tumors because of additional mutations that typically accompany KL tumors (e.g. Keap1). This raises the possibility that the squamous differentiation is a murine GEMM-specific phenomenon and is a marker for tumors under greater oxidative stress, which could be accompanied by increased GLUT-1 and sensitivity to glycolysis inhibitors.

We thank the reviewer 3 for the comment. We agree with the reviewer 3 that there are inherent limitations of the KL animal model in studying SqCC tumors. As the reviewer 3 pointed out, most human SqCCs do not present with the genetic mutations, which are engineered into the KL model. For example, Kras is rarely mutated in human lung SqCC and Ras signaling has been shown to drive GLUT1 expression. It is also poorly understood how the combination of a Kras mutation and LKB1 loss drives the formation of very distinct tumor types, ADC and SqCC. However, most observation made using the KL model was consistent with further analysis in human derived cell lines, tissues, patient derived xenografts, or within the TCGA. We are currently performing further functional analysis for subsequent publications in patient derived SqCC xenograft models and the recently described KEAP1 $-/-$, p53 $-/-$ GEMM of lung SqCC (Jeong et al Cancer Discovery 2016).

✓4. Figure 3B: the impact of shGLUT1 in HCC95 cells should be tested in at least one (but preferably more) SCC line(s), and compared with additional lung ADC lines, to see if the inhibition in cell and tumor growth is truly SCC specific (3B,F).

We thank the reviewer 3 (and the reviewer 1, comments No.1 and No.2) for this comment. We performed shRNA knockdown of GLUT1 in two additional SqCC cell lines, HCC1588 and HCC2814. Furthermore, we used a second hairpin targeting the CDS (shGLUT1 #2) as well as the original hairpin targeting the 3'UTR (shGLUT1 #1) for all three SqCC cell lines. We repeated the key experiments characterizing the proliferation, glucose uptake, and change in intracellular ATP, NADH, and NADPH induced by GLUT1 knockdown. Additionally, we used 7-AAD and Annexin V staining to look at cell viability and apoptotic induction after GLUT1 knockdown. We selected HCC2814 as an additional SqCC cell line because it features an amplification of PIK3CA and lacks a Kras mutation, which are representative features of human SqCC.

Consistent with HCC95 cell line, GLUT1 knockdown in additional SqCC cell lines, HCC1588 and HCC2814 dramatically suppressed the proliferative capacity (Figures 3a, b, Supplementary Figures 8d-e). This was accompanied by extensive apoptosis and cell death, observed by Annexin V and 7-AAD staining, respectively (Figure 3c, Supplementary Figures 8f). The uptake of fluorescently labeled glucose was significantly decreased in GLUT1 knockdown cells, which correlated with lower intracellular ATP, NADH, and NADPH (Figures 3d-f, Supplementary Figures 8g,h).

GLUT1 knockdown via shRNA was specifically cytotoxic in SqCC cell lines with extensive apoptosis and cell death observed compared to shGFP control cells, so much so that we could not select for a stable SqCC GLUT1 knockdown cell line for *in vivo* xenograft tumor assay, aside from HCC95. We are currently establishing inducible GLUT1 knockdown cell lines, which will be reported in a subsequent publication.

In addition to GLUT1 knockdown in SqCC cell lines as described above, we performed GLUT1 knockdown in two ADC cell lines, A549 and H522, with the same hairpins used to knockdown GLUT1 in SqCC cell lines. We characterized the effects of GLUT1 knockdown in these ADC cell lines by measuring *in vitro* proliferation, cell viability, glucose uptake and intracellular ATP levels. GLUT1 knockdown moderately suppressed the proliferation of A549 and H522, however, no apparent increase in the population of apoptotic or dead cells was noticed (Supplementary Figures 7a-c). After validating reduced uptake of fluorescently labeled glucose, we measured intracellular ATP and found no significant difference between shGFP control ADC cells and shGLUT1 knockdown cells (Supplementary Figures 7d, e). This suggests that upon reduced glucose uptake ADC cells are able to maintain intracellular ATP levels. We then implanted A549 and H522 shGFP and shGLUT1 knockdown cells into nude mice to determine if GLUT1 deficiency affected ADC tumor growth. We observed no significant difference in tumor growth rate in either ADC tumors (Supplementary Figures 7f,h).

In addition to our shRNA studies, we transiently knocked down GLUT1 using siRNA in SqCC cell lines HCC95 and HCC1588, and in ADC cell lines A549 and H522, and we repeated the *in vitro* proliferation and glucose uptake experiments. Consistently, the effect of transient siGLUT1 knockdown was more pronounced in SqCC cell lines, with a complete suppression of *in vitro* proliferation in SqCC but not ADC cells (Supplementary Figures 8a, b). Intriguingly, A549 was more sensitive to transient GLUT1 knockdown than H522, in accordance with previous reports that loss of LKB1 sensitizes cells to metabolic disruption (*Momcilovic et al Cancer Res 2015*), however both ADC cell lines continued to proliferate even after depletion of GLUT1. We measured fluorescent glucose uptake across all four cell lines transfected with either scrambled or GLUT1 targeting siRNA. Glucose uptake was dramatically higher in SqCC cells compared to ADC cells, and was accordingly reduced upon GLUT1 knockdown in all cell lines (Supplementary Fig 8c).

Finally, we added another ADC cell line, H1299 and SqCC cell line, HCC2814, for *in vivo* WZB117 treatment experiment. Consistent with other two ADC lines, A549 and H522, WZB117 showed no anti-tumor effects on H1299 xenograft tumors while markedly reducing HCC2814 xenograft tumor growth (Figure 5e-h; Supplementary Figure 13d-f).

Collectively, our expanded RNAi data using sh- and siRNA to knockdown GLUT1 in ADC and SqCC cells demonstrates a specific reliance on GLUT1-mediated glucose uptake and bioenergetics in SqCC compared to ADC. We have updated the results section to reflect the results of these experiments as follows:

Page 10, Line 21 – Page 12, Line 15

(Original)

“Genetic GLUT1 inhibition impairs cell viability and in vivo tumor growth of Lung SqCC.

Given the high expression level and critical roles for cellular glucose uptake, we reasoned that GLUT1 may be necessary for cell viability and growth of SqCC. Viability of SqCC HCC95 cells was significantly reduced by lentiviral-mediated GLUT1 knockdown compared to shGFP control cells even in high-glucose (25mM) conditions (Fig. 3a,b). We suspected that GLUT1 knockdown attenuates lung SqCC proliferation due to decreased glucose uptake. Indeed, we found that glucose uptake of HCC95 cells was dramatically decreased by 70% in GLUT1 knockdown cells compared to shGFP control cells (Fig.

3c,d). Furthermore, GLUT1 knockdown reduced intracellular ATP and NADH pools indicating increased energetic stress resulting from decreased glucose availability (Fig. 3e). Decreased intracellular NADPH was also observed upon GLUT1 knockdown, suggesting an essential role of GLUT1 for providing glucose for glycolytic branching pathways such as pentose phosphate pathway in SqCC (Fig. 3e). Overall, this data shows that GLUT1 expression and function is vital for maintaining energy homeostasis and proliferation in SqCC HCC95. To determine whether GLUT1 knockdown exerts SqCC tumor growth inhibition *in vivo*, we implanted shGLUT1 knockdown and shGFP control HCC95 SqCC cells into nude mice. In accordance with *in vitro* proliferation, GLUT1 knockdown significantly inhibited HCC95 xenograft tumor growth (Fig. 3f). Consistently, siRNA-mediated knockdown of GLUT1 expression in HCC1588 SqCC cells resulted in growth inhibition (Supplementary Fig. 7a,b). These results demonstrate the necessity of GLUT1 expression in SqCC cells for bioenergetic homeostasis and tumor growth.”

(Revised)

“Genetic GLUT1 inhibition impairs cell viability and *in vivo* tumor growth of Lung SqCC.

Given the high expression level and critical role in cellular glucose uptake, we reasoned that GLUT1 may be necessary for cell viability and growth of SqCC. We performed shRNA-mediated knockdown of GLUT1 in SqCC HCC95 and HCC1588 cell lines using two different GLUT1 targeting sequences (Fig. 3a). GLUT1 knockdown dramatically suppressed the proliferative capacity of SqCC cell lines compared to shGFP control cells even in high-glucose (25mM) conditions (Fig. 3b). This was accompanied by extensive apoptosis and cell death, which we observed by Annexin V and 7-AAD staining, respectively (Fig. 3c). In contrast, GLUT1 knockdown in ADC cell lines A549 and H522 only moderately suppressed proliferation with no induction of apoptosis or cell death (Supplementary Fig. 7a-c). We suspected that the decline in cell viability in SqCC cell lines was a result of a bioenergetic crisis resulting from GLUT1 inhibition. Fluorescently labeled glucose uptake was significantly decreased in GLUT1 deficient cells, which was correlated with lower intracellular ATP, NADH, and NADPH, suggesting not only disrupted bioenergetics, but an essential role for GLUT1 dependent flux of glucose intermediates into glycolysis dependent pathways such as the pentose phosphate pathway (Fig. 3d-f). In contrast to SqCC, reduced glucose uptake did not affect cell viability or intracellular ATP in ADC cells (Supplementary Fig. 7d,e). Consistently, transient GLUT1 knockdown using siRNA was more pronounced in SqCC cell lines, with a complete suppression of proliferation in SqCC but not ADC cells despite reduced glucose uptake in all four cell lines (Supplementary Fig. 8a-c). Intriguingly, A549 was more sensitive to transient GLUT1 knockdown than H522, in accordance with reports that loss of LKB1 sensitizes cells to metabolic disruption (*Momcilovic et al Cancer Res 2015*), however both ADC cell lines continued to proliferate even after depletion of GLUT1 (Supplementary Fig. 8b). We further employed an additional SqCC cell line, HCC2814 that contains an amplification of PIK3CA and lacks a Kras mutation, which are representative features of human SqCC. Consistently, GLUT1 knockdown in HCC2814 dramatically suppressed *in vitro* proliferation, which is associated with increased apoptosis and decreased intracellular ATP levels (Supplementary Fig 8d-h).

To determine whether GLUT1 knockdown inhibits tumor growth *in vivo* in SqCC, we implanted HCC95 expressing stable shGLUT1 or shGFP into nude mice. In accordance with *in vitro* proliferation, GLUT1 knockdown significantly inhibited the growth of HCC95 tumors (Fig 3g; Supplementary Fig. 7g). In contrast, no significant difference in tumor growth was observed between shGLUT1 and shGFP cells in A549 or H522 tumors (Supplementary Fig. 7f, h). The selective cytotoxicity of GLUT1 knockdown in SqCC strongly suggests glycolytic addiction and a specific reliance on glucose metabolism. Notably, a reduction of glucose uptake in ADC cells reveals that while ADC may rely on GLUT1 as its primary glucose transporter, there exist mechanisms within ADC to maintain bioenergetics, cellular viability,

and proliferative capacity outside of glucose metabolism. Overall, this data shows that GLUT1 expression and function is vital for maintaining energy homeostasis and proliferation in SqCC.”

Page 17, Line 4-14

(Original)

WZB117 has previously been shown to significantly inhibit ADC A549 tumor growth over a period of 10 weeks. However, ADC A549 showed no difference in tumor growth when treated with WZB117 over the course of 3-4 weeks (Fig. 5e), in agreement with the previous study (Liu et al Mol Cancer Ther 2012). In sharp contrast, SqCC HCC1588 exhibited up to 40% reduction in tumor growth upon GLUT1 inhibition (Fig. 5f,g). Consistent with 2-DG treatment, WZB117-treated SqCC HCC1588 tumors show increased levels of necrosis and apoptosis, however cell viability of ADC A549 tumors was not affected by WZB117 treatment (Fig. 5h).”

(Revised)

WZB117 has previously been shown to significantly inhibit ADC A549 tumor growth over a period of 10 weeks. However, ADC A549 showed no difference in tumor growth when treated with WZB117 over the course of 3-4 weeks (Fig. 5e; Supplementary Fig. 13c), in agreement with the previous study (Liu et al Mol Cancer Ther 2012). Equivalently, ADC H1299 tumor growth was unaffected by WZB117 treatment (Fig 5e; Supplementary Fig. 13d). In sharp contrast, SqCC HCC1588 and HCC2814 tumors exhibited up to 40% and 41% reduction in tumor growth upon GLUT1 inhibition, respectively (Fig. 5f, g; Supplementary Fig. 13e, f). Consistent with 2-DG treatment, WZB117-treated SqCC tumors show increased levels of necrosis and apoptosis, however cell viability of ADC A549 and H1299 tumors was not affected by WZB117 treatment (Fig. 5h).”

✓5. page 16: the finding that SCC has higher FDG uptake, GLUT1 expression, and other glycolysis markers has been reported previously (e.g. as examples: references 55, 56, and 57; as well as Schuurbiens, J Thorac Oncol. 2014 Glucose metabolism in NSCLC is histology-specific and diverges the prognostic potential of 18FDG-PET for adenocarcinoma and squamous cell carcinoma, which is not cited; or Choi et al, Technol Health Care. 2015, 26410497). These studies all support greater GLUT1 and glycolysis in lung SCC, which diminishes the novelty of a main point of the paper. They overstate the novelty of their finding further on page 25: “This study is, to our knowledge, the first implication of differentially utilized and targetable metabolic pathways in NSCLC histological subtypes” as the aforementioned papers and others have demonstrated histology-based differences in GLUT1 and other metabolic pathways.

We agree with the reviewer 3 (and the reviewer 2, comment No.2) that clinical observations of higher FDG-PET activity and glucose metabolic rates in lung SqCC have been previously made. Although we have made references to previous studies in the original manuscript, we have rightfully included Choi et al Technol Health Care 2015, Brown et al J Nucl Med 1999 and Schuurbiens et al J Thorac 2014 in the revised manuscript and apologize for not including these references in the original submission. As the reviewers state, in these previous studies, no detailed, functional study has been performed to assess metabolic heterogeneity among NSCLC lung cancer phenotypes, which we present here. Accordingly, we have modified the manuscript as follows:

Page 3, Line 19-22

(original)

“the differential usage of metabolic pathways in NSCLC subtypes has not been assessed”

(Revised)

“In particular, the differential usage of metabolic pathways in NSCLC subtypes has not been addressed outside clinical observations (*de Geus et al Lung Cancer 2007, Marom et al Lung Cancer 2001, Meijer et al Lung Cancer 2012, Brown et al J Nucl Med 1999, Schuurbiens et al J Thorac 2014, Choi Technol Health Care 2015*) and detailed functional studies have not been performed in representative pre-clinical models.”

(Removed from Discussion)

“This study is, to our knowledge, the first implication of differentially utilized and targetable metabolic pathways in NSCLC histological subtypes.”

✓6. page 16: the authors state “Collectively, these results demonstrate that 18F-FDG uptake is significantly increased in SqCC tumors, reflecting high levels of GLUT1 expression and glucose uptake, and further suggest that 18F-FDG-PET imaging can aid in designing more feasible diagnostic strategies in differentiating SqCC from other types of lung cancer.” Also, on page 24 “...developing more refined noninvasive diagnostic imaging techniques which specifically exploit elevated glucose influx will substantially improve the clinicopathological identification of SqCC from other types of lung cancer.” It is unclear how specifically FDG-PET would do this, as the overlap in the SUV between ADC and SCC shown in 6F is substantial; I expect an ROC curve which show little diagnostic utility. In the absence of such an analysis, a statement about its diagnostic utility should be removed. (As a separate note, it is unclear we need more “feasible” diagnostic strategies for distinguishing ADC from SCC- current strategies seem quite feasible. This section could benefit from review by an experienced lung cancer clinician).

We thank the reviewer 3 (and the reviewer 2, comment No.10) for the insightful comment. We agree with the reviewers that the SUVmax overlap is possibly too large to offer clinical utility in distinguishing ADC from SqCC. Although significantly higher in lung SqCC, FDG uptake may not be enough to differentiate between ADC and SqCC in clinical diagnosis, the observation of higher FDG uptake in lung SqCC highlights a core metabolic difference between lung SqCC and ADC. We have revised the interpretations of these experiments to support the observation of the elevated GLUT1 mediated glucose uptake in lung SqCC which we have expanded on in multiple functional, pre-clinical, and clinical models. We have also proposed that future studies will be performed to determine if FDG-PET imaging can be used to identify tumors most susceptible to GLUT1 or glycolytic inhibition. In this way, we propose that FDG-PET imaging may contribute to non-invasive molecular subtyping of SqCC and ADC tumors based on metabolic phenotypes and may contribute to the development of a metabolic marker for treatment outcome prediction. In response to the comments from the reviewer 2 as well as the reviewer 3, we have modified and removed the following sentences from the manuscript:

Page 17, Line 19-23

(Original)

“Differential expression of GLUT1 and glucose uptake between SqCC and ADC suggests potential indication of ¹⁸F-FDG-PET-based imaging for the differential diagnosis of SqCC. We took advantage of the KL mouse model, which displays mixed ADC and SqCC lung tumor heterogeneity (Supplementary Fig. 5a,b), by performing ¹⁸F-FDG-PET/CT imaging of KL mice followed by pathological evaluation and immunohistochemical staining for GLUT1 (Fig. 6a).”

(Revised)

“Previous clinical studies have identified increased ^{18}F -FDG uptake in squamous subtypes (*de Geus et al Lung Cancer 2007, Marom et al Lung Cancer 2001, Meijer et al Lung Cancer 2012, Brown et al J Nucl Med 1999, Schuurbiens et al J Thorac 2014, Choi Technol Health Care 2015*). We took advantage of the KL mouse model, which displays both ADC and SqCC lung tumor heterogeneity (Supplementary Fig. 5a,b), by performing ^{18}F -FDG-PET/CT imaging of KL mice followed by pathological evaluation and immunohistochemical staining for GLUT1 (Fig. 6a).”

(Removed from Results)

“Differential expression of GLUT1 and glucose uptake between SqCC and ADC suggests potential indication of ^{18}F -FDG-PET-based imaging for the differential diagnosis of SqCC.”

(Removed from Results)

“...further suggest that ^{18}F -FDG-PET imaging can aid in designing more feasible diagnostic strategies in differentiating SqCC from other types of lung cancer.”

(Removed from Discussion)

“Given that differential diagnosis of NSCLC is essentially required for determining a patient’s candidacy for molecularly targeted therapy, developing more refined noninvasive diagnostic imaging techniques which specifically exploit elevated glucose influx will substantially improve the clinicopathological identification of SqCC from other types of lung cancer.”

Minor corrections:

Page 21: glucose “depravation” should be “deprivation” (unless glucose is being linked to moral corruption).

We apologize for the typo. We have removed the sentence containing this error in response to the reviewer 2 (comment No. 13).

Reviewer Only Figure 1. (a) Extracellular acidification rate of ADC (A549 and H522) and SqCC (HCC95 and HCC1588) cell lines treated with 100 μM STF-31 (n=2 from two biologically independent experiments). (b) Cell viability (left) and flow cytometry (right) of Annexin-V and 7-AAD stained cells treated with GLUT1 inhibitor STF-31 (50 μM) for 48 hours (n=6 each group from three to four biologically independent experiments). ANOVA, ****P<0.0001. (c) Intracellular ATP levels in SqCC and ADC cells treated with 50 μM of STF-31 for 72 hours (n=5 each group from three to four biologically independent experiments). Error bars represent mean ± s.e.m. Two-tailed t-test. *P<0.05.

Reviewer Only Figure 2. Analysis of differential 18F-FDG uptake via SUVmax between lung SqCC (n=21) and ADC subtypes (Acinar, n= 10; Papillary, n=3; Lepidic, n=1) represented in clinical cohort reveals that SqCC patients have higher PET scan activity than ADC. Although, it should be noted that the numbers of patients belonging to the individual ADC subtypes are limited in this cohort. Boxes represent the median the interquartile range, whiskers are drawn from minima to maxima. The mean is denoted with a cross. Due to limited numbers of ADC patients within the various subtypes, we do not believe reporting a statistical analysis is appropriate for this data.

Reviewer Only Figure 3. Analysis of 18F-FDG uptake via SUVmax within ADC patients harboring EGFR mutations (n=6) and ADC patients with known wild type EGFR status (n=3). No significant difference in SUVmax was observed. However, it should be noted that the numbers of ADC patients tested for EGFR mutations in this cohort were limited. Error bars represent the mean the SEM. Two tailed T-test.

Reviewer Only Figure 4. Analysis of genes located in the SqCC ch3q amplification within the TCGA cohort of lung SqCC (n=498). Similar to PIK3CA, SOX2 and p63 exhibit frequent genomic amplifications in lung SqCC, with a frequency of 48.2% amplification/42.4% shallow gain for SOX2 and 41% amplification/47.2% shallow gain for p63. Boxes represent the median the interquartile range, whiskers are drawn from minima to maxima. P<0.0001, Mann-Whitney U-Test.

Reviewers' comments:

Reviewer #1 (Remarks to the Author):

The authors have addressed nearly all of my concerns and the paper is significantly advanced and now appropriate for acceptance for publication in Nature Communications.

Reviewer #2 (Remarks to the Author):

Thank you for your detailed response! I have no further questions or concerns.

Reviewer #3 (Remarks to the Author):

Comments regarding rebuttal by Goodwin et al, for manuscript entitled, "Distinct Metabolic Phenotypes within Non-small Cell Lung Cancer Define Selective Vulnerability to Glycolytic Inhibition of Lung Squamous Cell Carcinoma,"

Overall the revisions have adequately addressed the majority of the comments and improved the manuscript significantly. One area that remains largely correlative is the connection between the PI3K/AKT pathway in SCC, HIF1alpha, and GLUT1. The manuscript would benefit from the following:

1. Is the altered glycolytic dependence a function of histology, or merely activation of the PI3K/AKT pathway via amplification or mutation in PI3KCA, or unknown factors? To address this, it would be useful to assess Glut1 levels in adenocarcinoma tumors vs squamous tumors with comparable PI3KCA amplification assessed in the same manner. The data from supplemental figure 18b suggests that GLUT1 is elevated in amplified PI3KCA lung adenocarcinomas. If GLUT1 levels are comparable in PI3KCA amplified Adeno and SCC, it would be worth discussing this finding and highlighting that PI3KCA amplification may be the primary driver.
2. Increases in PI3KCA mutant tumors was not observed, which presumably have activation of the pathway. Please discuss why these tumors do not seem to have elevated Glut1 based on the limited data shown.
3. Is activation based on levels of phosphoAKT, phosphoS6, or other proteomic markers correlated with Glut1 levels in SCC and Adeno? This can be assessed using the RPPA data from the TCGA set, for example.
4. Does blockade of the PI3K/AKT pathway, or mTOR, reduce Glut1 levels in preclinical models? If so, how can the lack of significant activity for PI3K, AKT, or mTOR inhibitors clinically in SCC be explained?

Reviewer #3

Overall the revisions have adequately addressed the majority of the comments and improved the manuscript significantly. One area that remains largely correlative is the connection between the PI3K/AKT pathway in SCC, HIF1alpha, and GLUT1. The manuscript would benefit from the following:

1. Is the altered glycolytic dependence a function of histology, or merely activation of the PI3K/AKT pathway via amplification or mutation in PI3KCA, or unknown factors? To address this, it would be useful to assess Glut1 levels in adenocarcinoma tumors vs squamous tumors with comparable PI3KCA amplification assessed in the same manner. The data from supplemental figure 18b suggests that GLUT1 is elevated in amplified PIK3CA lung adenocarcinomas. If GLUT1 levels are comparable in PIK3CA amplified Adeno and SCC, it would be worth discussing this finding and highlighting that PI3KCA amplication may be the primary driver.

We appreciate the reviewer's comments. Lung ADC and SqCC patients with similar PIK3CA amplification statuses do not have similar levels of GLUT1 mRNA expression (Fig. 7a, Supplementary Fig. 18b), although in both histological subtypes, genomic amplification PIK3CA is associated with increased GLUT1 mRNA expression within the TCGA data. This suggests that additional histology-specific factors are most likely involved in the regulation of GLUT1 expression in lung SqCC in combination with PI3K/AKT signaling. We have recently reported that the NRF2 target gene, NQO1, stabilizes HIF1 α protein expression (Oh et. al. Nature Communications 2016) and our analysis of the TCGA has also revealed that NRF2 is elevated in lung SqCC patients compared to lung ADC (Supplementary Fig. 19c). Lung SqCC patients also exhibit high frequencies of NRF2 and KEAP1 mutations, stabilizing NRF2 signaling. Currently, we are actively studying the NRF2/NQO1/HIF-1 α axis as a potential candidate for GLUT1 regulation in lung SqCC. To address the Reviewer's comments, we have added the following to the discussion.

Page 25 Line 2 – 5 (Discussion)

"...Importantly, as in SqCC, GLUT1 mRNA expression was also increased in ADC patients possessing genomic copy number gains of PIK3CA, although occurring at a much lower frequency than in the TCGA lung SqCC cohort (Supplementary Fig. 18b). However, GLUT1 levels in PIK3CA amplified lung ADC tumors were significantly lower (3.3 fold) than lung SqCC tumors (Fig. 7a; Supplementary Fig. 18b), suggesting additional histology-specific regulation of GLUT1 expression in lung SqCC." This observation suggests..."

2. Increases in PI3KCA mutant tumors was not observed, which presumably have activation of the pathway. Please discuss why these tumors do not seem to have elevated Glut1 based on the limited data shown.

As the reviewer 3 pointed out, the available numbers of PIK3CA mutant lung ADC patients in the TCGA limit a conclusive analysis. We observed that the 15 patients within the TCGA lung ADC cohort with PIK3CA mutations did not show significantly higher GLUT1 expression than the 503 ADC patients with wild type PIK3CA. However, only 9 of the PIK3CA mutant ADC patients (tcga-44-5645, tcga-80-5607, tcga-75-6214, tcga-64-1677, tcga-05-4249, tcga-95-7948, tcga-93-7348, tcga-50-5044, tcga-55-7724) possess known oncogenic PIK3CA mutations as defined by OncoKB (cbioprotal). The remaining 6 patients possess mutations of unknown oncogenic affect. As well, 2 of the PIK3CA mutant ADC patients (tcga-05-4249 and tcga-95-7948) exhibit genomic loss of PIK3CA. Furthermore, multiple genetic mutations are known to induce GLUT1 expression in human cancers. For example, KRAS activating mutations have been shown to induce GLUT1 up-regulation in colorectal carcinomas (Yun et. al. Science 2009, Kerr et. al. Nature 2016). However, our analysis of the TCGA found no association between increased GLUT1 expression and KRAS activating mutations in lung ADC, again suggesting that histology-specific regulatory mechanisms of GLUT1 expression may exist. To address the reviewer's comment, we have added the following to the discussion.

Page 24 Line 10 – 13 (Discussion)

“our analysis found no significant increase in GLUT1 mRNA expression in ADC patients possessing KRAS, EGFR, BRAF, LKB1, PIK3CA, or PTEN mutations compared to those that did not (Supplementary Fig. 18a). However, it should be noted that the limited number of TCGA lung ADC patients who possess rare mutations hinder a conclusive analysis. Further study within large cohorts of NSCLC will be required to assess GLUT1 expression with less frequent oncogenic mutations. LKB1 inactivation has been linked...”

3. Is activation based on levels of phosphoAKT, phosphoS6, or other proteomic markers correlated with Glut1 levels in SCC and Adeno? This can be assessed using the RPPA data from the TCGA set, for example.

We thank the reviewer for the insightful suggestion. Our analysis of the TCGA RPPA data revealed a significant positive association between PIK3CA, pAKT, and p4E-BP1 protein expression with GLUT1 mRNA expression within lung SqCC, which was not observed in lung ADC (Supplementary Fig. 15b,c). However, we did not find a significant association between GLUT1 mRNA expression and pS6 in lung SqCC. Interestingly, we observed a modest negative correlation between GLUT1 mRNA

expression and PIK3CA, pAKT, and pS6 in lung ADC (Supplementary Fig.15c). These observations are consistent with the observation that EGFR mutant lung ADC tumors, which have increased PI3K/AKT/mTOR signaling (Harber et. al. Cold Spring Harbor Symposia on Quantitative Biology 2005, Camp et. al. Clinical Cancer Res. 2005), exhibit decreased ¹⁸F-FDG uptake when compared to EGFR wild type tumors (Mak et.al. The Oncologist 2011), suggesting the possible involvement of yet unknown histology-specific regulation of GLUT1 and glucose metabolism in lung ADC. Supporting our hypothesis, we also observed a significant negative correlation between GLUT1 mRNA expression and PTEN protein expression in lung SqCC, which was not observed in lung ADC (Supplementary Fig. 15b,c). Together, these results suggest a positive association between PI3K/AKT signaling and GLUT1 expression in Lung SqCC. To address the reviewer's comments, we have added the following to the text.

Page 19 Line 23 – Page 20 Line 6 (Results)

“Furthermore, GSEA performed on the ranked set of differentially expressed genes between SqCC and ADC TCGA tumor samples identified a significant enrichment of AKT and mTOR oncogenic signaling in SqCC, as defined by the oncogenic signatures gene set (Supplementary Fig. 14e,f). Analysis of TCGA reverse phase protein array (RPPA) (Nanjundan et. al. J Thoracic Oncology 2010) data also identified a significant positive correlation between GLUT1 mRNA expression and PIK3CA, phospho-AKT (pT308), phospho-AKT (pS473), and phospho-4E-BP1 (pT37/T46) as well as a negative associate between GLUT1 mRNA expression and PTEN protein expression (Supplementary Fig. 15b,c). These results suggest significant input from the PI3K/AKT/mTOR pathway regulating GLUT1 expression in SqCC.”

Page 29 Line 3 – 8 (Discussion)

“Significantly enriched AKT and mTOR oncogenic signaling (Supplementary Fig. 14e,f) as defined by curated oncogenic signaling gene sets⁶⁴, further suggests increased PI3K/AKT/mTOR activity in SqCC. Correlative analysis of PI3K/AKT/mTOR signaling using RPPA data with GLUT1 mRNA expression confirms a strong association in lung SqCC (Supplementary Fig. 15b). Interestingly, we observed a modest negative correlation between GLUT1 mRNA expression and PIK3CA, pAKT, and pS6 in lung ADC, suggesting the involvement of further histology-specific regulation of GLUT1 in ADC (Supplementary Fig. 15c). In an agreement with human SqCC, KL and Xenograft SqCC tumors exhibit significantly increased AKT activity and downstream signaling directing the stabilization of HIF1-a and GLUT1 expression (Fig. 7c; Supplementary Fig. 16a-c).”

4. Does blockade of the PI3K/AKT pathway, or mTOR, reduce Glut1 levels in preclinical models? If so, how can the lack of significant activity for PI3K, AKT, or mTOR inhibitors clinically in SCC be explained?

We thank the reviewer for this question. Indeed, the Shackelford group has reported that mTOR inhibitor, MLN0128, attenuates HIF-1 α and GLUT1 expression in lung cancer cell lines and the KL murine model of NSCLC. ¹⁸F-FDG uptake of KL tumors was reported to be significantly decreased when treated with MLN0128 (Momcilovic et. al. Cancer Res. 2015). Additionally, Mayer et. al. has shown significant reduction in ¹⁸F-FDG uptake in ER+/HER2- human breast cancers treated with the pan-PIK3 inhibitor, Buparlisib, suggesting the role of PIK3 and subsequent downstream signaling in the regulation of glucose uptake (Mayer et. al. J. Clinical Oncology 2014). Despite promising effects in pre-clinical models, PI3K, AKT, and mTOR inhibition have proven to lack therapeutic efficacy as stand-alone therapies due to possible acquired resistance, which may arise from multiple signaling kinase network adaptation mechanisms. For example, Rodrik-Outmezguine et. al. and Chandarlapaty et. al. have shown that mTOR kinase inhibitors can abolish AKT phosphorylation at serine 473 but only transiently reduce phosphorylation at threonine 308, while at the same time, reducing feedback inhibition of receptor tyrosine kinases, leading to subsequent AKT phosphorylation and pathway propagation (Rodrik-Outmezguine et. al. Cancer Discovery 2011; Chandarlapaty et. al. Cancer Cell 2011). Despite significant reduction of GLUT1 and HIF1 α expression with the use of mTOR inhibitors in preclinical models, the coexistence of multiple genetic aberrations, observed in NSCLC patients, perturbing numerous signaling pathways can reduce the efficacy of the inhibition of a single pathway. In particular, the PI3K/AKT/mTOR pathway is extensively interconnected with the RAS/MAPK pathway. Inhibition of mTORC1 has been shown to activate the MAPK pathway via feedback activation from the PIK3 pathway (Carracedo et. al. JCI 2008, Mendoza et. al. Trends Biochem Sci. 2011). Future pre-clinical and clinical assessment is needed to assess the effectiveness of stand-alone and combinatorial PI3K/AKT/mTOR inhibition as an effective therapeutic option for lung SqCC. Many of the current clinical studies investigating inhibition of this pathway have not been designed to enroll patients with activated PI3K/AKT/mTOR pathways, and have not applied predictive biomarkers for the determination of therapeutic effectiveness. Optimal patient selection will be important for future clinical investigation of PI3K/AKT/mTOR pathway inhibition as a therapeutic option. To address the reviewer's comments, we have added the following to the discussion.

Page 29 Line 10 – 20 (Discussion)

Importantly, these genomic alterations that lead to an increase in PI3K/AKT/mTOR signaling are among the most strongly correlated with GLUT1 expression in SqCC (Fig.

7a,b). Additionally, it has been shown that mTOR inhibitor, MLN0128, attenuates HIF-1 α and GLUT1 expression in NSCLC cell lines and the KL model of NSCLC, which is associated with decreased ¹⁸F-FDG uptake further corroborating the regulation of PI3K/AKT/mTOR pathway signaling on GLUT1 expression through HIF-1 α (Momcilovic et. al. Cancer Res. 2015). Collectively, these data suggest that PI3K/AKT/mTOR signaling may represent a viable target for future therapeutic development. However, the clinical efficacy of PI3K/AKT/mTOR inhibition as a stand-alone therapy is limited in lung SqCC (Vansteenkiste et. al. J. Thoracic Oncology 2015, Tan et. al. J. Thoracic Oncology 2015) due to potential multiple kinase signaling adaptation mechanisms (Rodrik-Outmezguine et. al. Cancer Discovery 2011; Chandarlapaty et. al. Cancer Cell 2011, O'Reilly et. al. Cancer Research 2006, Vallejo-Díaz Oncotarget 2016, Carracedo et. al. JCI 2008, Mendoza et. al. Trends Biochem Sci. 2011). Further pre-clinical and clinical assessment of resistance mechanisms in lung SqCC is needed for the development of combinatorial PIK3/AKT/mTOR targeted therapies as well as effective screening to determine patient candidacy for treatment.”

REVIEWERS' COMMENTS:

Reviewer #3 (Remarks to the Author):

The authors have adequately addressed the majority of concerns.

In response #3, the authors state "Analysis of TCGA reverse phase protein array (RPPA) (Nanjundan et. al. J Thoracic Oncology 2010) data...".

The Nanjundan paper is not from the TCGA, which was published after this paper. Please correct.

Reviewer #3

The authors have adequately addressed the majority of concerns. In response #3, the authors state “Analysis of TCGA reverse phase protein array (RPPA) (Nanjundan et. al. J Thoracic Oncology 2010) data...”.

The Nanjundan paper is not from the TCGA, which was published after this paper. Please correct.

We would like to thank the reviewer for the valuable feedback given to this manuscript and we apologize for not citing the TCGA RPPA data correctly. We have modified the manuscript by citing *Li J, et al. Nat Methods 2013*.